# Tiled Flash Linear Attention:
# More Efficient Linear RNN and xLSTM Kernels

**Maximilian Beck**[1,2]    **Korbinian Pöppel**[1,2]    **Phillip Lippe**[2*]    **Sepp Hochreiter**[1,2]

[1] ELLIS Unit, LIT AI Lab, Institute for Machine Learning, JKU Linz, Austria
[2] NXAI GmbH, Linz, Austria

## Abstract

Linear RNNs with gating recently demonstrated competitive performance compared to Transformers in language modeling. Although their linear compute scaling in sequence length offers theoretical runtime advantages over Transformers, realizing these benefits in practice requires optimized custom kernels, as Transformers rely on the highly efficient Flash Attention kernels (Dao, 2024). Leveraging the chunkwise-parallel formulation of linear RNNs, Flash Linear Attention (FLA) (Yang & Zhang, 2024) shows that linear RNN kernels are faster than Flash Attention, by parallelizing over chunks of the input sequence. However, since the chunk size of FLA is limited, many intermediate states must be materialized in GPU memory. This leads to low arithmetic intensity and causes high memory consumption and IO cost, especially for long-context pre-training. In this work, we present *Tiled Flash Linear Attention* (TFLA), a novel kernel algorithm for linear RNNs, that enables arbitrary large chunk sizes and high arithmetic intensity by introducing an additional level of sequence parallelization within each chunk. First, we apply TFLA to the xLSTM with matrix memory, the mLSTM (Beck et al., 2024). Second, we propose an mLSTM variant with sigmoid input gate and reduced computation for even faster kernel runtimes at equal language modeling performance. In our speed benchmarks, we show that our new mLSTM kernels based on TFLA outperform highly optimized Flash Attention, Linear Attention and Mamba kernels, setting a new state of the art for efficient long-context sequence modeling primitives. Our code is available at: `https://github.com/NX-AI/mlstm_kernels`

## 1 Introduction

With the trend of training models of ever increasing size with large datasets on thousands of GPUs, it becomes increasingly important to optimize the model architecture as well as its low-level implementations for modern hardware. Transformers (Vaswani et al., 2017), which are the core architecture of nowadays state-of-the-art models are highly optimized, but the computational requirements of self-attention scale quadratically with sequence length. This creates significant challenges for both training and inference on long context.

Recently, recurrent alternatives with linear scaling in sequence length (Beck et al., 2024; Sun et al., 2023; Dao & Gu, 2024; Yang et al., 2024b) promise efficiency gains, especially on long sequences and during inference while providing competitive performance. The success of these emerging recurrent architectures is based on two main pillars: (1) A parallel or chunkwise-parallel formulation (Sun et al., 2023; Hua et al., 2022), which, like Attention, calculates all outputs in parallel during training, and (2) kernel implementations that are close to or exceed training speeds of FlashAttention (Dao, 2024).

Yang et al. (2024b) show that their custom Flash Linear Attention (FLA) kernels, based on the chunkwise-parallel formulation of linear RNNs, achieve faster runtimes than FlashAttention.

---

*Now at Google Deepmind.

39th Conference on Neural Information Processing Systems (NeurIPS 2025).

They accomplish this by dividing the sequence into chunks and recurrently materializing only the initial RNN state of each chunk in GPU memory. Subsequently, in the parallel part they employ one level of sequence parallelism and compute the outputs for each chunk in parallel. For a small chunk size and long sequences, this leads to a large amount of intermediate states to be stored and loaded from GPU memory, which increases memory consumptionand decreases arithmetic intensity. Since modern GPUs see a faster increase in computation throughput than memory bandwidth (Gholami et al., 2024), it is essential to minimize large memory IO and increase arithmetic intensity. A simple approach would be to increase the chunk size. However, the chunk size of FLA is limited by the physical SRAM available on the GPU.

To solve this problem, we introduce *Tiled Flash Linear Attention* (TFLA) which enables unlimited chunk sizes by introducing a second level of sequence parallelism via tiling of the matrix computations in sequence dimension within each chunk. This increases the arithmetic intensity of the kernels and allows us to efficiently balance memory consumption and IO vs. computation.

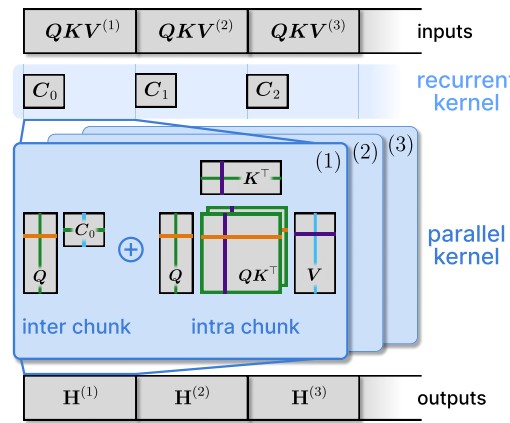

Figure 1: **Tiled Flash Linear Attention (TFLA)** consists of a recurrent kernel and a parallel kernel, which process the input sequence in chunks $QKV^{(k)}$ (1st level of sequence parallelism). The recurrent kernel materializes the memory state $C_{k-1}$ for each chunk. The parallel kernel computes the output states $H^{(k)}$ for all chunks. TFLA uses tiling for the 3 matrix-multiplications in the parallel kernel (2nd level of sequence parallelism) to fully utilize the hardware and to prevent materialization of many memory states.

In this paper, we implement our Tiled Flash Linear Attention algorithm for the xLSTM with matrix memory – the mLSTM Beck et al. (2024). The mLSTM is a linear RNN that uses exponential gating with scalar gates per head, along with an additional normalizer state for output normalization. This gating mechanism has demonstrated competitive performance compared to Transformers and Mamba on language modeling tasks at moderate scales. However, for comparisons at even larger scales, efficient kernels that leverage the chunkwise-parallel formulation for the mLSTM were still missing. In our speed benchmarks, we show that our new mLSTM kernels based on TFLA outperform highly optimized Attention, Linear Attention and Mamba kernels.

After optimizing our kernels for the existing mLSTM computation, we seek ways to reduce kernel runtime by targeted modifications to the mLSTM. Towards this end, we propose *mLSTMsig*, an mLSTM with sigmoid input gate and reduced computation, that enables even faster kernel implementations at no performance drops on language modeling up to 1.4B parameter scale.

Finally, motivated by the equal performance of both mLSTM variants, we perform an empirical study inspired by transfer function analysis from control theory (Ogata, 2010) to understand their differences and characteristics. We find that both mLSTM variants exhibit the same transfer behavior and, moreover, our analysis suggests that the input gate biases should be initialized at larger negative values. In extensive experiments on language modeling, we confirm that this initialization improves training stability as well as the overall performance of mLSTM models.

To summarize, in this work, we make the following contributions: (1) We introduce *Tiled Flash Linear Attention*, a new chunkwise-parallel kernel algorithm for Linear RNNs with two levels of sequence parallelism, that enables arbitrary large chunk sizes and apply it to the mLSTM (Beck et al., 2024). (2) We introduce *mLSTMsig*, a faster mLSTM variant with sigmoid input gate with no performance losses up to 1.4B parameter scales. (3) We improve the training stability and performance of the mLSTM through careful gate initialization guided by our empirical transfer behavior analysis.

## 2 mLSTM Formulations

The mLSTM cell is the fully parallelizable part of the xLSTM (Beck et al., 2024). It has a matrix memory and exponential gating.

## 2.1 Recurrent Formulation

In its recurrent formulation, the mLSTM cell processes the series of input vectors $\boldsymbol{x}_t \in \mathbb{R}^d$ for time steps $t \in \{1, \ldots, T\}$ mapping a state $(\boldsymbol{h}_{t-1}, \boldsymbol{C}_{t-1}, \boldsymbol{n}_{t-1}, m_{t-1})$ to a successor state $(\boldsymbol{h}_t, \boldsymbol{C}_t, \boldsymbol{n}_t, m_t)$ given an input $\boldsymbol{x}_t$. Here, $\boldsymbol{h}_t \in \mathbb{R}^{d_{hv}}$ denotes the hidden state, $\boldsymbol{C}_t \in \mathbb{R}^{d_{qk} \times d_{hv}}$ denotes the cell state responsible for long-term memory, $\boldsymbol{n}_t \in \mathbb{R}^{d_{qk}}$ denotes the normalizer state, and $m_t \in \mathbb{R}$ denotes the max state. Together normalizer and max state control the magnitude of the exponential input gate and ensure stability (see Appendix D.1). The recurrent mLSTM formulation is given by the following state update equations:

$$m_t = \max\left\{\log \sigma(\tilde{\mathrm{f}}_t) + m_{t-1}, \ \tilde{\mathrm{i}}_t\right\} \tag{1}$$

$$\boldsymbol{C}_t = \mathrm{f}_t \, \boldsymbol{C}_{t-1} + \mathrm{i}_t \, \boldsymbol{k}_t \, \boldsymbol{v}_t^\top \tag{2}$$

$$\boldsymbol{n}_t = \mathrm{f}_t \, \boldsymbol{n}_{t-1} + \mathrm{i}_t \, \boldsymbol{k}_t \tag{3}$$

$$\widetilde{\boldsymbol{h}}_t = \frac{\boldsymbol{C}_t^\top \left(\boldsymbol{q}_t / \sqrt{d_{qk}}\right)}{\max\left\{\left|\boldsymbol{n}_t^\top \left(\boldsymbol{q}_t / \sqrt{d_{qk}}\right)\right|, \exp(-m_t)\right\}} \tag{4}$$

$$\boldsymbol{h}_t = \boldsymbol{o}_t \odot \mathrm{NORM}(\widetilde{\boldsymbol{h}}_t) \tag{5}$$

The scalar forget and input gates $\mathrm{i}_t, \mathrm{f}_t \in \mathbb{R}$ are computed as $\mathrm{f}_t = \exp\left(\log \sigma(\tilde{\mathrm{f}}_t) + m_{t-1} - m_t\right)$ and $\mathrm{i}_t = \exp(\tilde{\mathrm{i}}_t - m_t)$ with the pre-activations $\{\tilde{\mathrm{i}}_t, \tilde{\mathrm{f}}_t\} = \boldsymbol{w}_{\{\mathrm{i,f}\}}^\top \boldsymbol{x}_t + b_{\{\mathrm{i,f}\}}$, respectively. The vector output gate $\boldsymbol{o}_t \in \mathbb{R}^{d_{hv}}$ is given by $\boldsymbol{o}_t = \sigma(\tilde{\boldsymbol{o}}_t)$ with the pre-activations $\tilde{\boldsymbol{o}}_t = \boldsymbol{W}_{\mathbf{o}} \boldsymbol{x}_t + \boldsymbol{b}_{\mathbf{o}}$ and the sigmoid function $\sigma$. The norm layer NORM in (5) can be either RMS norm (Zhang & Sennrich, 2019) or LayerNorm (Ba et al., 2016). Typically, multiple of these cells operate simultaneously as parallel heads, similar to Transformers (Vaswani et al., 2017).

## 2.2 Chunkwise-Parallel Formulation

The chunkwise-parallel formulation is a trade-off between the parallel and the fully recurrent formulation. It has a recurrent part and a (quadratic) parallel part, with an overall sub-quadratic scaling in sequence length. Similar to the fully parallel formulation (see Appendix B.1), we assume that all inputs are available at once. We then split the sequence of length $T$ into $N_c = \lceil T/L \rceil$ chunks of length $L$ and use $k \in \{1, \ldots, N_c\}$ for the chunk index. We rearrange the input and forget gates, as well as the queries, keys, and values into chunkwise matrices, where the chunk index becomes the first dimension. For example, the forget gate pre-activations $\tilde{\mathbf{f}} \in \mathbb{R}^T$ are rearranged into a matrix $\tilde{\mathbf{f}} = (\tilde{\mathbf{f}}^{(1)}, \tilde{\mathbf{f}}^{(2)}, \ldots, \tilde{\mathbf{f}}^{(N_c)}) \in \mathbb{R}^{N_c \times L}$, where each row $\tilde{\mathbf{f}}^{(k)} = (\mathrm{f}_{(k-1)N_c+1}, \mathrm{f}_{(k-1)N_c+2}, \ldots, \mathrm{f}_{kN_c}) \in \mathbb{R}^L$ contains the pre-activations of the chunk $k$. The input gate pre-activations follow analogously. Similarly, the queries, keys and values are rearranged into chunkwise tensors $\boldsymbol{Q}, \boldsymbol{K} \in \mathbb{R}^{N_c \times L \times d_{qk}}$ and $\boldsymbol{V} \in \mathbb{R}^{N_c \times L \times d_{hv}}$. Here, the query matrix $\boldsymbol{Q}^{(k)} = (\boldsymbol{q}_{(k-1)N_c+1}, \ldots, \boldsymbol{q}_{kN_c}) \in \mathbb{R}^{L \times d_{qk}}$ contains the query vectors of chunk $k$. Keys, and values follow analogously. For notational simplicity we drop the leading $N_c$ dimension and omit normalization layer and the output gate, i.e. consider $\widetilde{\boldsymbol{h}}_t$ as hidden state outputs.

**Chunkwise Gates.** Given the logarithmic forget gates $\bar{\mathbf{f}}^{(k)} = \log \sigma(\tilde{\mathbf{f}}^{(k)}) \in \mathbb{R}^L$ and input gates $\bar{\mathbf{i}}^{(k)} = \log \exp(\tilde{\mathbf{i}}^{(k)}) \in \mathbb{R}^L$, we can compute the logarithmic chunkwise gates $g_k \in \mathbb{R}$, and $\mathbf{b}_k, \mathbf{a}_k \in \mathbb{R}^L$ as $g_k = \mathrm{sum}\left(\bar{\mathbf{f}}^{(k)}\right)$, $\mathbf{b}_k = \mathrm{cumsum}\left(\bar{\mathbf{f}}^{(k)}\right)$, and $\mathbf{a}_k = \mathrm{rev\_cumsum}\left(\bar{\mathbf{f}}^{(k)}\right) + \bar{\mathbf{i}}^{(k)}$. We refer to Appendix B.2 for more details on the chunkwise gates. In Figure 2, we show the summed forget gates $g_k$ contain the forget gate contribution of all forget gates within a chunk. The cumulative forget gate vectors $\mathbf{b}_k$ contain the forget gate contributions *from the beginning of the chunk up to the current time step* within the current chunk. The cumulative input gate vectors $\mathbf{a}_k$ contain the input gates for every timestep as well as the forget gate contributions *from the current time step to the end of the chunk*.

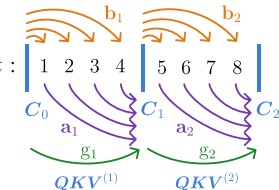

Figure 2: Illustration of the chunkwise gates $\mathbf{a}_k$, $\mathbf{b}_k$ and $g_k$ with chunk size $L = 4$. Each arrow denotes an element in the gate vectors. See Figure 9 in Appendix B.2 for more details.

**Inter-chunk Recurrent Contribution.** The inter-chunk recurrence is given by

$$\boldsymbol{C}_k = \bar{\mathrm{g}}_k \boldsymbol{C}_{k-1} + \left( \bar{\mathbf{a}}_k \odot \boldsymbol{K}^{(k)} \right)^\top \boldsymbol{V}^{(k)} \tag{6}$$

$$\boldsymbol{n}_k = \bar{\mathrm{g}}_k \boldsymbol{n}_{k-1} + \left( \bar{\mathbf{a}}_k \odot \boldsymbol{K}^{(k)} \right)^\top \mathbf{1}, \tag{7}$$

where $\bar{\mathrm{g}}_k$ and $\bar{\mathbf{a}}_k$ are the stabilized chunkwise gates. This recurrent part resembles the fully recurrent formulation in Section 2.1, but instead of computing the intermediate states for every timestep $t$, we compute them directly for every $L$ time steps without materializing the states in between.

**Intra-chunk Parallel Contribution.** The recurrent part is followed by the intra-chunk parallel contribution:

$$\widetilde{\mathbf{D}}^{(k)} = \begin{cases} -\infty & \text{for } i < j \\ \mathbf{b}_k - \mathbf{b}_k^\top + \bar{\mathbf{i}}^{(k)^\top} & \text{for } i \geqslant j \end{cases} \tag{8}$$

$$\overline{\mathbf{S}}^{(k)} = \left( \frac{1}{\sqrt{d_{qk}}} \boldsymbol{Q}^{(k)} \boldsymbol{K}^{(k)^\top} \right) \odot \mathbf{D}^{(k)} = \mathbf{S}^{(k)} \odot \mathbf{D}^{(k)}, \tag{9}$$

where $\mathbf{D}^{(k)} \in \mathbb{R}^{L \times L}$ is the stabilized gate matrix. Compared to the fully parallel part from Appendix B.1, the quadratic cost of the matrices $\mathbf{D}^{(k)}, \mathbf{S}^{(k)} \in \mathbb{R}^{L \times L}$ is greatly reduced, since the chunk size $L$ is typically small compared to the sequence length $T$.

**Output Computation.** Finally, the contributions from the intra-chunk parallel part $\mathbf{H}_{\text{intra}}^{(k)}$ are combined with the inter-chunk recurrent part $\mathbf{H}_{\text{inter}}^{(k)}$ to obtain the hidden states $\mathbf{H}^{(k)} \in \mathbb{R}^{L \times d_{hv}}$ for each chunk $k$ (see Figure 1):

$$\mathbf{H}_{\text{inter}}^{(k)} = \left( \overline{\mathbf{b}}_k \odot \frac{\boldsymbol{Q}^{(k)}}{\sqrt{d_{qk}}} \right) \boldsymbol{C}_{k-1} = \overline{\boldsymbol{Q}}^{(k)} \boldsymbol{C}_{k-1}, \qquad \mathbf{H}_{\text{intra}}^{(k)} = \overline{\mathbf{S}}^{(k)} \boldsymbol{V}^{(k)}, \tag{10}$$

$$\mathbf{H}^{(k)} = \left( \mathbf{H}_{\text{inter}}^{(k)} + \mathbf{H}_{\text{intra}}^{(k)} \right) / \mathbf{h}_{\text{denom}}^{(k)}, \tag{11}$$

where $\mathbf{h}_{\text{denom}}^{(k)} \in \mathbb{R}^L$ is a normalization factor. Appendix B.2 and B.3 provide a detailed description of the chunkwise-parallel forward and backward pass. Appendix F provides the FLOP and memory operation counts for all formulations.

## 3 Tiled Flash Linear Attention

Flash Linear Attention (Yang et al., 2024b) introduces a fast kernel algorithm for the chunkwise formulation for Linear Attention (cf. Section 2.2 without gates) and shows that their implementation is faster than optimized FlashAttention (Dao, 2024). This speedup is achieved by single level sequence parallelism, where the states $\boldsymbol{C}_k$ are first materialized in GPU memory and then the outputs $\mathbf{H}^{(k)}$ are computed in parallel. However, since in Flash Linear Attention the chunk size parameter determines the tile sizes in SRAM, the maximum chunk size is limited (typically $L = 64$) by the physical SRAM size of the GPU. Therefore, we have to materialize many states in HBM, where the number of states is $N_c = \lceil T/L \rceil$. This leads to low arithmetic intensity and high GPU memory consumption, which poses challenges especially for long-context pre-training.

We begin with a brief review of fundamentals on GPUs for writing efficient kernels in Appendix C.1.

**More Efficient Kernels via Two Level Sequence Parallelism.** To address the issue of limited chunk sizes, Tiled Flash Linear Attention (TFLA) introduces two levels of sequence parallelism, which enables fast kernels and a trade-off between memory consumption and computational efficiency (see Figure 6). The first level is the parallelization over the chunks of the sequence, which requires to compute and materialize intermediate states $\boldsymbol{C}_k$ in GPU High Bandwidth memory (HBM). For this we use a recurrent kernel similar to previous work (Yang et al., 2024b). The second level is the parallelization within each chunk, which is achieved by tiling the intra chunk attention matrix along the chunk dimension. This second level of parallelism enables large chunk sizes and hence reduces the memory consumption for the intermediate states as we have to store and load $N_c = \lceil T/L \rceil$ intermediate states in HBM on each kernel call, where $T$ is the sequence length and $L$ is the chunk size. In addition to the two levels of sequence parallelism and the naive parallelization over the batch

and head dimensions, TFLA also parallelizes over the embedding dimension. This enables arbitrary large head dimensions and results in a massive parallelization over five dimensions, which is crucial for achieving high performance on modern GPUs. We analyze the theoretical runtime of our TFLA kernels in Appendix G.

**Forward Pass.** We review the matrix multiplication operations of the intra-chunk parallel part of the mLSTM in order to show how we efficiently parallelize these operations. For simplicity we omit the the gate computations and normalization, as these do not influence the work partitioning. We also omit the leading batch, head and chunk dimension, over which we can parallelize naively as they do not interact with the matrix multiplication (see Table 1). In simplified form, the intra-chunk parallel forward pass of the mLSTM (and other linear RNNs) for a chunk $k$ can be written as three matrix multiplications, which we fuse into a single kernel:

$$\underset{(L_{hq} \times d_{hv})}{\mathbf{H}^{(k)}} = \underbrace{\left( \underset{(L_{hq} \times d_{qk})}{\boldsymbol{Q}^{(k)}} \underset{(d_{qk} \times L_{kv})}{\boldsymbol{K}^{(k)\top}} \right) \underset{(L_{kv} \times d_{hv})}{\boldsymbol{V}^{(k)}}}_{\mathbf{H}_{\text{intra}}^{(k)}} + \underbrace{\underset{(L_{hq} \times d_{qk})}{\boldsymbol{Q}^{(k)}} \underset{(d_{qk} \times d_{hv})}{\boldsymbol{C}_{k-1}}}_{\mathbf{H}_{\text{inter}}^{(k)}} \tag{12}$$

In Appendix A.2, we show that TFLA can be applied to any linear RNN that either follows or can be reformulated into this form. In order to parallelize the computation in (12), we introduce the block sizes $B_{Lhq}$, $B_{Lkv}$, $B_{dqk}$ and $B_{dhv}$ for the attention matrix, query, key, value and hidden state dimensions $L_{hq}$, $L_{kv}$, $d_{qk}$ and $d_{hv}$, along which we either parallelize or accumulate by using a loop inside the kernel.

In Figure 3, we show our TFLA tiling strategy for the forward pass $\mathbf{H}^{(k)}$ kernel. We parallelize across the outer sequence dimension $L_{hq}$ with $N_{Lhq} = L_{hq}/B_{Lhq}$ programs, and across the outer embedding dimension $d_{hv}$ with $N_{dhv} = d_{hv}/B_{dhv}$ programs. We loop over the inner dimensions $L_{kv}$ and $d_{qk}$, which are tiled by the block sizes $B_{Lkv}$ and $B_{dqk}$ respectively.

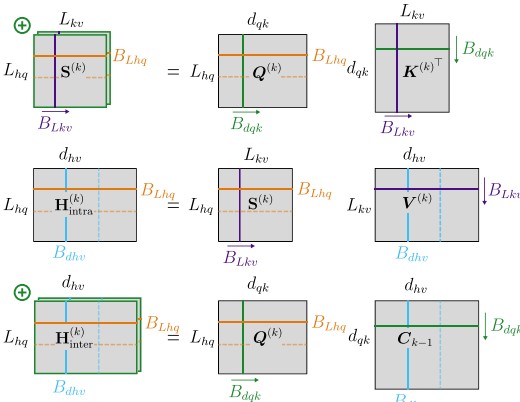

Figure 3: TFLA Intra-Chunk Tiling. We loop over $B_{Lkv}$ and $B_{dqk}$ (indicated by arrows) and parallelize over $B_{Lhq}$ and $B_{dhv}$ (indicated by dashed lines) blocks. $\oplus$ denotes block-wise accumulation.

**Tiled Computation.** For the mLSTM we cannot simply accumulate the results of the matrix multiplications $\mathbf{H}_{\text{intra}}^{(k)}$ along the $L_{kv}$ dimension and $\mathbf{H}_{\text{inter}}^{(k)}$ due to the stabilization of the exponential input gate with the max state $m_t$. The max state tracks the maximum of the forget and input gates over time and is used to stabilize the exponential input gate similar to the safe softmax computation (Milakov & Gimelshein, 2018). Since we compute the hidden state output $\mathbf{H}^{(k)}$ in blocks along the chunk size (i.e. time) dimension $L_{kv}$, we need to rescale during accumulation of the block results for $\mathbf{H}_{\text{intra}}^{(k)}$ and the overall results into $\mathbf{H}^{(k)}$ in the same way as FlashAttention (Dao, 2024). We provide details on the rescaling in Section B.2. For the backward pass there is no rescaling necessary as we store the max states in the forward pass and reuse them in the backward pass. The pseudocode for the forward pass of TFLA for the mLSTM is listed in Algorithm 1.

**Backward Pass.** The parallelization strategy for the backward pass of TFLA is more complex than for the forward pass, since we need to compute three output tensors — the gradients for the queries, keys and values, of which each has an intra-chunk and inter-chunk part. However, in Section C.4 we show that the individual gradients can be mapped to three matrix multiplications similar to the forward pass. In TFLA, we then implement a separate kernel for each gradient and use the same work partitioning as in the forward pass but swap the loop and parallelization dimensions, accordingly. Table 1 summarizes the work partitioning of our TFLA kernels.

Table 1: TFLA kernel parallelization and loop dimensions. Parallelization dimensions are indicated by P and loop dimensions by L. The last column shows the first two dimensions of the 3D kernel launch grid. The last dimension of all kernels is $N_{\text{chunk}} \cdot N_{\text{head}} \cdot N_{\text{batch}}$.

| Kernel | $L_{hq}$ | $L_{kv}$ | $d_{qk}$ | $d_{hv}$ | Thread Block Grid |
|---|---|---|---|---|---|
| $\mathbf{H}^{(k)}$ | P | L | L | P | $\left( \frac{d_{hv}}{B_{dhv}}, \frac{L_{hq}}{B_{Lhq}}, \dots \right)$ |
| $\delta\boldsymbol{Q}^{(k)}$ | P | L | P | L | $\left( \frac{d_{qk}}{B_{dqk}}, \frac{L_{hq}}{B_{Lhq}}, \dots \right)$ |
| $\delta\boldsymbol{K}^{(k)}$ | L | P | P | L | $\left( \frac{d_{qk}}{B_{dqk}}, \frac{L_{kv}}{B_{Lkv}}, \dots \right)$ |
| $\delta\boldsymbol{V}^{(k)}$ | L | P | L | P | $\left( \frac{d_{hv}}{B_{dhv}}, \frac{L_{kv}}{B_{Lkv}}, \dots \right)$ |

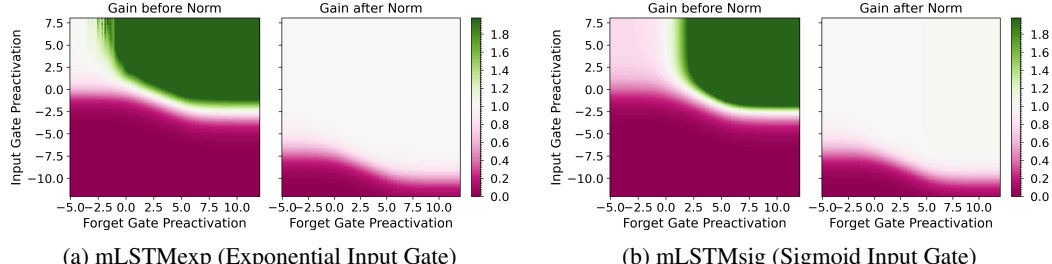

(a) mLSTMexp (Exponential Input Gate)  (b) mLSTMsig (Sigmoid Input Gate)

Figure 4: Transfer behavior of the mLSTM before and after the RMS-norm layer ($\epsilon$ =1e-6) for different input and forget gate values. The color shows the gain of the mLSTM defined in (16). After the norm layer mLSTMexp and mLSTMsig exhibit the same transfer behavior.

## 4 Faster mLSTM with Sigmoid Input Gate

The mLSTM with exponential gating (i.e. exponential input gate) introduced by Beck et al. (2024) requires to compute and keep track of two additional states, the normalizer state $\boldsymbol{n}_t$ and max state $m_t$, as we show in Appendix D.1. Both will increase kernel runtime: The normalizer must be computed through summations, and tracking the max state throughout the tiled computation in TFLA (see Section 3 and C.2) prevents efficient fusing of loops within the kernel (see Appendix C.3).

Additionally, our analysis in Section 4.2 suggests to initialize the input gate biases at larger negative values (e.g. -10), such that the input gate pre-activations can grow slowly during training. We observe that most of these values stay below 0 during training (see Figure 15 in Appendix E). Therefore, we seek an alternative activation function which is similar to the exponential function in the negative range, but bounded in the positive range. This suggests to use the sigmoid function $\sigma(x) = \frac{1}{1+\exp(-x)} = \frac{\exp(x)}{\exp(x)+1}$, which converges to $\exp(x)$ for $x \to -\infty$ and 1 for $x \to \infty$.

### 4.1 mLSTM with Sigmoid Input Gate

The sigmoid function can be computed in two ways as given above.Depending on the sign of $x$ it can be ensured that the argument of exp is always smaller than 0 to avoid numerical overflow. Therefore, we do not need to control the magnitude of $x$ externally with a max state and as a consequence also drop the normalizer state (see Appendix D.1). This yields the mLSTM with sigmoid input gate (henceforth referred to as *mLSTMsig*) in its recurrent formulation as

$$\boldsymbol{C}_t = \sigma(\tilde{\mathrm{f}}_t)\, \boldsymbol{C}_{t-1} + \sigma(\tilde{\mathrm{i}}_t)\, \boldsymbol{k}_t\, \boldsymbol{v}_t^\top \tag{13}$$

$$\widetilde{\boldsymbol{h}}_t = \boldsymbol{C}_t^\top\, (\boldsymbol{q}_t/\sqrt{d_{qk}}) \tag{14}$$

$$\boldsymbol{h}_t = \sigma(\tilde{\mathbf{o}}_t) \odot \mathrm{NORM}\left(\widetilde{\boldsymbol{h}}_t\right) \tag{15}$$

where the query, key, and value vectors $\boldsymbol{q}_t, \boldsymbol{k}_t, \boldsymbol{v}_t$, and the gate preactivations $\tilde{\tilde{\mathrm{i}}}_t, \tilde{\mathrm{f}}_t, \tilde{\mathbf{o}}_t$ remain the same as for the mLSTM with exponential input gate (from now on referred to as *mLSTMexp*) in Section 2.1. We confirm that our TFLA mLSTMsig forward kernel is over 30% faster than the mLSTMexp forward (see Section 5.2), and show that mLSTMsig performs equally well compared to mLSTMexp in our language modeling experiments up to 1.4B parameters (see Section 5.1).

### 4.2 Normalization of mLSTM and Linear RNNs

Motivated by the performance of mLSTMsig, we seek to understand the differences between mLSTMsig and mLSTMexp empirically. To approach this, we draw inspiration from the concept of frequency response and transfer function analysis for control systems design, where typically the amplitude ratio or gain of output and input signals for different frequencies is considered (Ogata, 2010, Ch. 7). In our case, we analyze the transfer behavior of mLSTMsig and mLSTMexp for random inputs $\boldsymbol{q}_t, \boldsymbol{k}_t$ and $\boldsymbol{v}_t$ and different input gate and forget gate preactivations $\tilde{\mathrm{i}}_t$ and $\tilde{\mathrm{f}}_t$.

We will see that the normalization layer $\boldsymbol{y} = \mathrm{NORM}(\boldsymbol{x})$, will play a crucial role in our analysis. The default norm layer in language modeling, the RMS norm (Zhang & Sennrich, 2019) with input vector input vector $\boldsymbol{x} \in \mathbb{R}^d$ and output vector $\boldsymbol{y} \in \mathbb{R}^d$ is defined as $\boldsymbol{y} = \frac{\boldsymbol{x}}{\mathrm{RMS}(\boldsymbol{x})} \odot \boldsymbol{\gamma}$, where $\mathrm{RMS}(\boldsymbol{x}) = \sqrt{\frac{1}{d}\sum_{i=1}^{d} x_i^2 + \epsilon}$, with with $\boldsymbol{\gamma} \in \mathbb{R}^d$ being a learnable scale parameter. The epsilon parameter $\epsilon \in \mathbb{R}$ is a small constant typically set to 1e-6 to avoid division by zero.

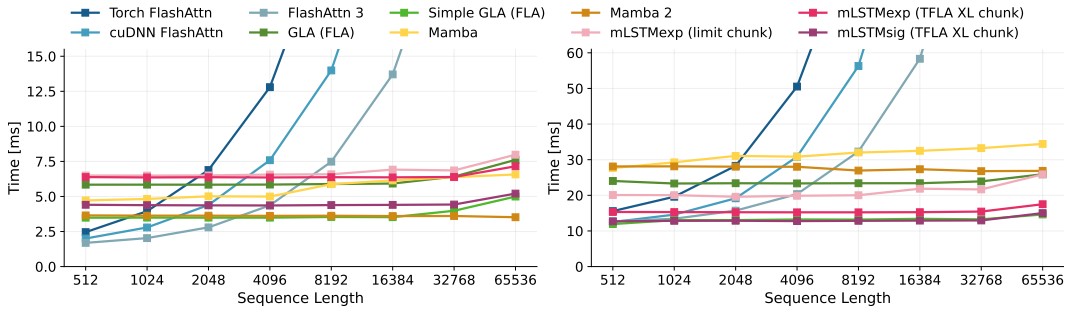

(a) Inference (Forward pass)  (b) Training (Forward and Backward pass)

Figure 5: TFLA Kernel Runtime Benchmark for embedding dimension 4096 and 65,536 tokens on NVIDIA H100 GPUs. In training, our TFLA kernels are faster than FlashAttention 3 for longer sequences and over 2x faster than Mamba 2 kernels for all sequence lengths.

**Transfer Behavior of the mLSTM.** We analyze the transfer behavior by computing the gain of the mLSTM cells from random inputs sampled from $\mathcal{N}(0, 1)$ to hidden states before and after the norm layer for varying input and forget gate values. More specifically, we compute the gains $G_{\text{before}}$ and $G_{\text{after}}$ as

$$G_{\text{before}} = \frac{\|\widetilde{\boldsymbol{h}}_t\|_{\max}}{\|\boldsymbol{v}_t\|_{\max}} \quad \text{and} \quad G_{\text{after}} = \frac{\|\text{NORM}(\widetilde{\boldsymbol{h}}_t)\|_{\max}}{\|\boldsymbol{v}_t\|_{\max}}, \qquad (16)$$

where $\|\boldsymbol{x}\|_{\max} := \max(|\boldsymbol{x}_1|, \ldots, |\boldsymbol{x}_d|)$ and we average over the time dimension. For more details see App. D.2. In Figure 4 we observe that the transfer behavior of mLSTMsig without normalizer is identical to mLSTMexp with normalizer and max state. Both exhibit a transition from suppressing ($G = 0$) to passing ($G = 1$) the signal at larger negative input gate preactivation values, which could partly explain the matching performance in our language modeling experiments.

**Normalization Layers in other Gated Linear RNNs.** Interestingly, almost all other gated linear RNN variants also place a normalization layer after the RNN cell (Sun et al., 2023; Dao & Gu, 2024; Qin et al., 2024b; Yang et al., 2024b). Often this is justified with improved training stability, but a more thorough discussion is missing (Lieber et al., 2024). Qin et al. (2022) analyze the effect of the norm layer after a non-gated, kernel-based linear attention layer (Katharopoulos et al., 2020) and show that this effectively prevents unbounded gradients. We also confirm that the norm layer has a significant impact on training stability and the gradient norm during training. In Section 5.1 we show that initializing the input gate bias at larger negative values, as suggested by our transfer behavior analysis in Figure 4, prevents large gradient norm variance and spikes during training. Relatedly, the general effect of layer normalization in the Transformer architecture has been investigated in several studies (Xiong et al., 2020; Zhu et al., 2025).

**Effect of Normalization on Gating in Linear RNNs.** We hypothesize that at this point the normalization layer does not only have a stabilizing effect by controlling the magnitude of the layer activations through rescaling, but also actively participates in the information routing or gating mechanism of the linear RNN. For example, if the squared norm of $\boldsymbol{C}_t^\top \boldsymbol{q}$, which is controlled by input and forget gates through $\boldsymbol{C}_t^\top$, is smaller than the epsilon, the denominator in the $\text{NORM}(\boldsymbol{x})$ layer is dominated by $\epsilon$ and the output moves towards zero (indicated by the purple area in Fig. 4). Hence, by moving through the x-y plane in Fig. 4, the gates could learn to suppress or amplify any input in the sequence. In Section D.2 we show additional experiments on the effect of varying the normalization layer epsilons and different modifications of the normalizers for the mLSTM.

## 5 Experiments

In this section, we examine the performance of the two mLSTM variants mLSTMexp (mLSTM with exponential input gate) and mLSTMsig (mLSTM with sigmoid input gate). We compare two kernel algorithms: (1) `limit_chunk`: A kernel that is limited in chunk size $L$. (2) `xl_chunk`: Our Tiled Flash Linear Attention (TFLA) kernels with unlimited chunk size. For details see Section 3. We assess the performance of mLSTMsig compared to mLSTMexp in Section 5.1 and benchmark the runtime of our kernels against other baselines in Section 5.2. In App. E.1 we verify the numerical correctness of our kernels.

## 5.1 Language Modeling with mLSTM

We train three different model sizes (160M, 400M, 1.4B parameters) with context lengths 4096 and 8192 on the DCLM dataset (Li et al., 2024). We include Llama2 style Transformer models (Touvron et al., 2023b) as reference in our comparison and describe our experiment setup, model architecture and training recipe in Appendix E.2.

**Performance in Language Modeling.** We compare mLSTMsig and mLSTMexp models on next-token prediction with different number of heads or head dimensions. Table 2 and Table 6 show the results for context length 4096 and 8192, respectively. We find that our `limit_chunk` and `xl_chunk` kernels yield the same loss (up to small numerical deviations) for almost all head dimensions. For some head dimensions, we observe gradient norm or loss spikes for the `xl_chunk` kernels, which affect the final loss. As a main result we find that mLSTMsig performs equally well compared to mLSTMexp.

Table 2: Validation Perplexity at context length 4096. EXP and SIG denote mLSTMexp and mLSTMsig. LIMIT and XL correspond to `limit_chunk` and `xl_chunk` kernels.

| Size | Tokens | Heads | Llama | EXP LIMIT | EXP XL | SIG XL |
|------|--------|-------|-------|-----------|--------|--------|
| 160M | 19B | 6 | | 21.03 | 21.18 | 21.03 |
| | | 12 | 20.89 | 21.03 | 21.06 | 21.05 |
| 400M | 24B | 4 | | 16.66 | 16.66 | 16.67 |
| | | 8 | | 16.55 | 16.80 | 16.67 |
| | | 16 | 16.85 | 16.60 | 16.61 | 16.61 |
| 1.4B | 33B | 4 | | 13.31 | 13.35 | 13.34 |
| | | 8 | | 13.20 | 13.22 | 13.21 |
| | | 16 | 13.64 | 13.20 | 13.87* | 13.22 |

**Effect of Input Gate Bias Initialization.** We analyze the effect of the input gate bias initialization on training stability and performance of our mLSTM models in Appendix E.2. We observe in Figure 12 and 13, that initializing the input gate biases to -10 effectively mitigates large gradient norm spikes and variance during training for both mLSTMexp and mLSTMsig. We therefore conclude that the additional input gate not only improves performance (see Table 7), but also improves training stability, if initialized correctly.

**Effect of Norm Layer Epsilon.** In Appendix E.2, we investigate the effect of the norm layer epsilon on language modeling performance for mLSTMexp. Our transfer behavior analysis in Figure 4 suggests, that there exists an interplay between norm layer epsilon and input gate bias initialization. We confirm this in our grid search in Figure 14 and find that the best performing configuration is the default epsilon $\epsilon = $1e-6 with input gate biases initialized to -10.

## 5.2 Kernel Benchmark

We compare the runtime of our mLSTM `limit_chunk` and TFLA `xl_chunk` kernels with kernel implementations of the state-of-the-art sequence modeling primitives FlashAttention (Dao, 2024; Shah et al., 2024), Mamba (Gu & Dao, 2024; Dao & Gu, 2024) and GLA Yang et al. (2024b). In Appendix E.3 we compare with other kernels from the Flash Linear Attention library (Yang & Zhang, 2024). We run our benchmarks on NVIDIA H100 GPUs.

**Runtime Benchmark.** We use the standard embedding dimension of 4096 for 7B parameter models and adapt the head dimensions per kernel accordingly. For example for FlashAttention we use 32 heads with head dim 128 and for the mLSTM we use 16 heads with head dim 256. Following the practice of Shah et al. (2024), we keep the number of tokens constant at 65,536 and vary sequence length and batch size accordingly. For further details see Appendix E.3. Figure 5 shows the runtime benchmark results for inference, i.e. forward pass only, (left) and for training, i.e. forward-backward pass (right). Our mLSTMexp TFLA `xl_chunk` kernels with two level sequence parallelism is about 25% faster than our `limit_chunk` kernels. Through targeted modifications of the input gate of the mLSTM we save computation and enable more efficient kernel implementations for the forward pass of mLSTMsig (see Sec. 4). This yields another speedup of over 30% for the forward pass of the mLSTMsig TFLA kernel over the mLSTMexp TFLA kernel.

In training, our TFLA kernels are faster than FlashAttention 3 for longer sequences and more than 2x faster than Mamba 2 kernels for all sequence lengths. We perform additional runtime benchmarks for varying head dimensions and a more in-depth comparison to the FLA (Yang et al., 2024b) and LightningAttention2 (Qin et al., 2024a) kernels in Appendix E.3.

**Runtime vs. Memory Trade-off.** The chunk size parameter $L$ balances the computation between the two levels of sequence parallelism (see Sec. 3). Smaller chunk sizes increase memory consumption, because more chunks are materialized in memory, but they reduce the quadratic compute FLOPs in the parallel part. Larger chunk sizes have the opposite effect. They decrease memory consumption, but increase quadratic compute FLOPs. In Figure 6, we measure this trade-off for our mLSTMsig TFLA xl_chunk kernels.

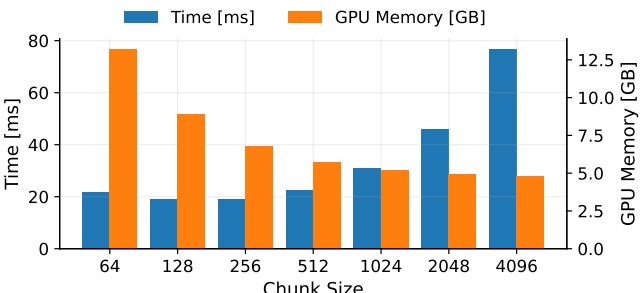

Figure 6: Memory vs. Runtime Trade-off of TFLA Forward-Backward Pass. We show the mLSTMsig for embedding dimension 4096 (8 heads with dim 512), sequence length 8192 and batch size 8. By varying the chunk size parameter, TFLA kernels can effectively balance memory vs. runtime.

## 5.3 Theoretical Runtime and Arithmetic Intensity

In Figure 6, we empirically observe that there exists an optimal chunk size (between 128 and 256) at which the runtime is minimized. In Appendix G, we compute the theoretical runtime optimal chunksize and the arithmetic intensity of TFLA depending on the chunk size by taking the FLOPs, memory operations and hardware accelerator specification into account. We find that the theoretical runtime optimum exceeds the empirically observed one (see Figure 21 in Appendix G.1), suggesting that our current kernel implementations may not yet fully exploit the available performance potential. We analyze the effect of the chunk size parameter $L$ on runtime, FLOPs, and arithmetic intensity in detail in Appendix F and G summarize our findings as follows:

(1) The chunk size $L$ mediates a trade-off between runtime and GPU memory usage [Figure 6]. (2) $L$ determines the total compute in FLOPs: $L = 1$ matches the recurrent formulation, while $L = T$ matches the parallel one [Figure 19]. (3) There exists an optimal chunk size $L \in [1, T]$ that minimized the total FLOP count [Equation (103), Figure 19, Figure 20]. (4) Increasing $L$ raises the arithmetic intensity of TFLA kernels [Equation (109), Figure 22]. (5) The chunk size determines whether the kernel is memory-bound or compute-bound on a given hardware [Figure 23, Figure 22]. (6) FLOPs/s alone can be misleading; the optimal chunk size should be chosen based on total runtime [Figure 24, Figure 21]. (7) The runtime-optimal chunk size scales proportionally with the square root of the head dimension and the accelerator's computational intensity [Figure 25, Figure 26]. (8) Newer hardware generations require larger chunk sizes to approach peak performance. [Figure 26, Figure 21].

## 6 Related Work

Tiled Flash Linear Attention (TFLA) integrates the concept of tiling along one sequence dimension of the attention matrix for improved work partitioning (Dao, 2024) with the strategy of dividing the sequence into chunks (Yang et al., 2024b), yielding two levels of sequence parallelism (see Figure 7).

**Flash Attention.** Flash Attention (Dao et al., 2022) is an IO-aware implementation of softmax attention introduced by (Vaswani et al., 2017). It uses the idea of tiling to reduce the number of memory reads/writes between GPU high bandwidth memory (HBM) and GPU on-chip SRAM. In this way the quadratic attention matrix $QK^\top$ is never materialized in HBM, which reduces the memory requirement from quadratic with sequence length to linear, and significantly speeds up the kernel due to reduced memory IO cost. However, the computation still remains quadratic with sequence length. Flash Attention 2 (Dao, 2024) improves the work partitioning by parallelizing the attention computation over the sequence dimension in addition to the naive parallelization over batch and head dimension. Flash Attention 3 (Shah et al., 2024) leverages new hardware features of recent GPU generations (e.g. NVIDIA Hopper GPUs) such as FP8 precision or exploiting asynchrony of Tensor cores and Tensor Memory Accelerators (TMA) to speed up Flash Attention.

TFLA is IO-aware and parallelizes over one sequence dimension of the intra-chunk $QK^\top$ matrix as the second level of sequence parallelism. New hardware features will also speed up future TFLA implementations.

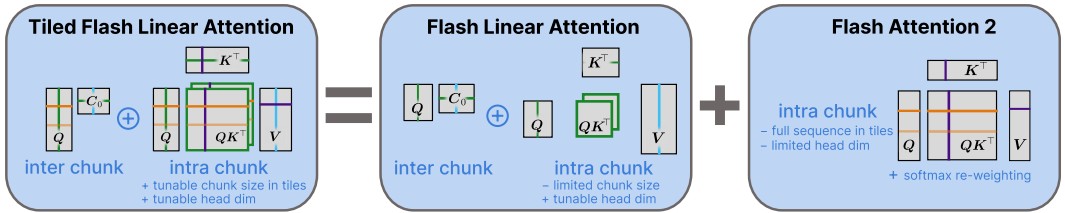

Figure 7: Tiled Flash Linear Attention (TFLA) combines Flash Linear Attention and Flash Attention 2. Flash Attention (middle) employs tiling across the head dimensions (green & light blue), while Flash Attention (right) uses tiling in the sequence dimension (orange & purple). Tiled Flash Linear Attention (left) combines both tiling strategies.

**Flash Linear Attention.** Flash Linear Attention (FLA) (Yang et al., 2024b; Yang & Zhang, 2024) makes use of the fact that linear attention can be interpreted as linear RNN (Katharopoulos et al., 2020). It then leverages the chunkwise-parallel formulation of linear RNNs (Hua et al., 2022; Sun et al., 2023) for efficient kernel implementations, that process the sequence in chunks. More specifically, Yang et al. (2024b) propose two FLA variants: A version that materializes intermediate states in HBM and a non-materialization version. The materialization version consists of two kernels: The first is a recurrent kernel that materializes the first intermediate states of every chunk. The second kernel then processes all chunks in parallel and computes the outputs within the chunks. The non-materialization version was proposed concurrently by Qin et al. (2024a) and does not employ parallelism over the sequence dimension, but processes the inputs sequentially in chunks.

TFLA uses the idea of chunking of the sequence for the first level of sequence parallelism.

**Application of TFLA to other Linear RNNs.** While TFLA is designed for efficient mLSTM kernels (Beck et al., 2024), its formulation also extends naturally to other linear RNNs such as RetNet, Mamba 2, and DeltaNet (Sun et al., 2023, 2024; Dao & Gu, 2024; Yang et al., 2024a). A more detailed discussion of these extensions is provided in Appendix A.

# 7    Conclusion and Future Work

With Tiled Flash Linear Attention (TFLA) we introduce an algorithm for linear RNN and mLSTM kernels with two levels of sequence parallelism. Our TFLA kernels for the mLSTM with exponential input gate (mLSTMexp) achieve state-of-the art kernel execution speeds, while remaining flexible to trade off GPU memory consumption and runtime. To further improve kernel runtimes, we propose mLSTMsig, a mLSTM variant with sigmoid input gate, that reduces computation and increases speed. Our experiments show that both mLSTM variants perform equally well on language modeling.

Although we enhance training stability through careful gate initialization informed by our empirical transfer behavior analysis, future work could explore instabilities arising from numerical errors in kernel implementations in greater depth. Finally, the programming techniques and hardware features used to optimize Flash Attention (Shah et al., 2024) could also be applied to our TFLA algorithm to approach peak performance on next-generation hardware, as suggested by our theoretical runtime analysis. This establishes TFLA as a strong candidate for a foundational primitive in future long-context language models.

## Acknowledgements

We thank Sebastian Böck, Richard Kurle, Patrick Blies, Antonio Orvieto, Maximilian Stadler, Thomas Schmied, Kajetan Schweighofer and Sebastian Lehner for helpful discussions and feedback.

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

# Appendix

## TABLE OF CONTENTS

# A    Related Work

## A.1    Other Related Work

**Other Hardware-Aware Optimizations.**    Optimized, hardware-aware implementations enable the exploration of new primitives or new model architectures. FlashRNN Pöppel et al. (2025) introduces a framework of IO-aware optimized CUDA kernels in order to simplify research on traditional, non-parallelizable RNNs. Mamba (Gu & Dao, 2024) enables large scale language modeling experiments (Waleffe et al., 2024) with an efficient parallel scan algorithm in their optimized CUDA kernels. FlashFFTConv (Fu et al., 2024) provides efficient implementations for FFT convolutions for modern hardware by reducing IO and leveraging specialized matrix multiply units. DeltaNet Yang et al. (2024c,a) introduces an efficient algorithm for training linear Transformers with the delta rule (DeltaNet) (Schlag et al., 2021), which enables to scale up DeltaNet to standard language modeling settings.

Our TFLA kernel algorithm provides an effective method to balance the runtime and memory for linear RNN kernels based on their chunkwise-parallel formulation, paving the way to even larger model training setups, e.g. for multi-billion parameter xLSTM models (Beck et al., 2025).

**Gating mechanisms for Linear RNNs.**    Many different gating techniques for linear RNNs have been explored (Sun et al., 2023; Beck et al., 2024; Yang et al., 2024b; Gu & Dao, 2024; Dao & Gu, 2024; Sun et al., 2024; Qin et al., 2024b; Orvieto et al., 2023; Katsch, 2023; Peng et al., 2024). We propose mLSTMsig, a variant of mLSTM with a sigmoid input gate similar to the original LSTM (Hochreiter, 1991; Hochreiter & Schmidhuber, 1997) and empirically analyze the transfer behavior.

**mLSTM Applications.**    The mLSTM, which is the xLSTM with matrix memory, has already been adopted in several domains beyond language modeling. For example, Alkin et al. (2025) find that the mLSTM can serve as a generic backbone for computer vision architectures. In the field of robotics, the mLSTM architecture has been explored as a large recurring action model, which uses the efficient linear-time inference complexity of the mLSTM (Schmied et al., 2025), and as the backbone for imitation learning policies (Jia et al., 2025). In the domain of biological and chemical sequences, the mLSTM has been studied for generative modeling, representation, and in-context learning (Schmidinger et al., 2025). Finally, mLSTM has also been examined in the domain time series forecasting (Alharthi & Mahmood, 2024), where sLSTM (Beck et al., 2024) has also been applied (Kraus et al., 2024; Kong et al., 2025).

These and other applications will benefit from our TFLA kernels, which provide efficient and scalable implementations of the mLSTM, that can be easily integrated into existing models. We believe that our TFLA mLSTM kernels will increase the adoption of mLSTM in other application areas as well.

## A.2    Application of TFLA to other Linear RNNs

We have applied TFLA to linear RNNs with scalar headwise gates such as mLSTMexp and mLSTMsig. In this section, we show how TFLA could be applied to other Linear RNNs, but leave the implementation to future work.

**Linear RNNs with Scalar Headwise Gates.**    With minimal modifications, TFLA can be applied to other scalar headwise gated linear RNNs, such as, for example, Retention (Sun et al., 2023), Gated Retention (Sun et al., 2024), Simple GLA (Yang et al., 2024b) or Mamba 2 (Dao & Gu, 2024). In general, TFLA implementations of these linear RNNs can be obtained by modifying the forget and input gate parameterizations (of the TFLA mLSTMsig). Since neither of the aforementioned models has an input gate in the mLSTM sense, we fix the input gate of TFLA mLSTMsig to one (i.e. $\sigma(\tilde{i}_t) = 1$). Then, for Simple GLA or Retention we keep the headwise scalar sigmoid forget gate $\sigma(\tilde{f}_t)$ or set it to a constant decay parameter $\sigma(\tilde{f}_t) = \gamma$.

To implement Gated Retention — which introduces a sigmoid forget gate modulated by a temperature parameter $\tau$ — we modify the activation function of the forget gate to incorporate this temperature term. Beyond coupling the forget gate parameter $A$ with the keys $k_t$ (represented by the parameter $B$) through the step size parameter $\Delta$, Mamba 2 also adopts the linear attention structure from

Equation (12). In this formulation, Mamba's notation $\boldsymbol{C}$, $\boldsymbol{B}$, and $\boldsymbol{X}$ corresponds to our queries $\boldsymbol{Q}$, keys $\boldsymbol{K}$, and values $\boldsymbol{V}$, respectively.

**Linear RNNs with Delta Rule.** Recently, Linear RNNs with non-diagonal state transition matrices have become popular due to their increased expressivity that enable them to solve synthetic state tracking tasks (Grazzi et al., 2025; Siems et al., 2025; Peng et al., 2025; Movahedi et al., 2025). One method to implement such non-diagonal linear RNNs efficiently is (Gated) DeltaNet (Yang et al., 2024c,a), which introduces a hardware efficient algorithm for linear transformers with the delta rule. The core of DeltaNet's efficient implementation is the chunkwise-parallel formulation of the delta-rule, which is implemented using primitives from Flash Linear Attention.

In our notation, the chunkwise-parallel formulation of DeltaNet is given by the inter-chunk recurrence

$$
\underset{(d_{qk} \times d_{hv})}{\boldsymbol{C}_k} = \underset{(d_{qk} \times d_{hv})}{\boldsymbol{C}_{k-1}} + \underset{(d_{qk} \times L_{kv})}{\boldsymbol{K}^{(k)\top}} \underbrace{\left( \underset{(L_{kv} \times d_{hv})}{\boldsymbol{U}^{(k)}} - \underset{(L_{kv} \times d_{qk})}{\boldsymbol{W}^{(k)}} \underset{(d_{qk} \times d_{hv})}{\boldsymbol{C}_{k-1}} \right)}_{\boldsymbol{V}^{*(k)}}, \tag{17}
$$

and the combination between inter-chunk and intra-chunk contribution

$$
\underset{(L_{hq} \times d_{hv})}{\mathbf{H}^{(k)}} = \underset{(L_{hq} \times d_{qk})}{\boldsymbol{Q}^{(k)}} \underset{(d_{qk} \times d_{hv})}{\boldsymbol{C}_{k-1}} + \left( \underset{(L_{hq} \times d_{qk})}{\boldsymbol{Q}^{(k)}} \underset{(d_{qk} \times L_{kv})}{\boldsymbol{K}^{(k)\top}} \right) \underbrace{\left( \underset{(L_{kv} \times d_{hv})}{\boldsymbol{U}^{(k)}} - \underset{(L_{kv} \times d_{qk})}{\boldsymbol{W}^{(k)}} \underset{(d_{qk} \times d_{hv})}{\boldsymbol{C}_{k-1}} \right)}_{\boldsymbol{V}^{*(k)}}, \tag{18}
$$

where we omit the mask $\mathbf{M}$ applied to $\boldsymbol{Q}\boldsymbol{K}^\top$ of DeltaNet for clarity. We refer to Yang et al. (2024c) for the definition of the matrices $\boldsymbol{U}^{(k)}$ and $\boldsymbol{W}^{(k)}$.

By defining the new values $\boldsymbol{V}^{*(k)} = \boldsymbol{U}^{(k)} - \boldsymbol{W}^{(k)}\boldsymbol{C}_{k-1}$ we can recover the core formula of TFLA for the intra-chunk parallel forward pass (see Equation (12)) to which we can apply our TFLA tiling strategy from Section 3. For computing the matrices $\boldsymbol{U}^{(k)}$, $\boldsymbol{W}^{(k)}$ and the new values $\boldsymbol{V}^{*(k)}$ one could follow the same strategy as in DeltaNet, where $\boldsymbol{U}^{(k)}$ and $\boldsymbol{W}^{(k)}$ are computed in a separate kernel before the recurrent kernel. Then, the new values $\boldsymbol{V}^{*(k)}$ are computed and stored in HBM in the recurrent kernel together with the memory states $\boldsymbol{C}_k$.

We conclude that TFLA can be applied to any linear RNN that either follows or can be reformulated into the simplified, chunkwise-parallel form of Equation (12).

# B    Extended mLSTM Formulations

## B.1    Fully Parallel Formulation

For the parallel formulation it is assumed that all inputs are available at once. Then, the queries, keys and values $\boldsymbol{q}_t, \boldsymbol{k}_t, \boldsymbol{v}_t$ can be stacked into the matrices $\boldsymbol{Q}, \boldsymbol{K} \in \mathbb{R}^{T \times d_{qk}}, \boldsymbol{V} \in \mathbb{R}^{T \times d_{hv}}$ in order to compute all hidden states $\mathbf{H} \in \mathbb{R}^{T \times d_{hv}}$ in parallel using the following equations:

$$
\widetilde{\mathbf{D}} = \log \mathbf{F} + \tilde{\mathbf{I}} \tag{19}
$$

$$
\boldsymbol{m} = \max_j \widetilde{\mathbf{D}}_{ij}, \tag{20}
$$

$$
\mathbf{D} = \exp(\widetilde{\mathbf{D}} - \boldsymbol{m}) \tag{21}
$$

$$
\mathbf{S} = \frac{1}{\sqrt{d_{qk}}} \boldsymbol{Q}\boldsymbol{K}^\top \tag{22}
$$

$$
\overline{\mathbf{S}} = \mathbf{S} \odot \mathbf{D} \tag{23}
$$

$$
\boldsymbol{n} = \max\left( |\overline{\mathbf{S}}\,\mathbf{1}|, \exp(-\boldsymbol{m}) \right) \tag{24}
$$

$$
\mathbf{H} = \left( \overline{\mathbf{S}} \odot \left( \boldsymbol{n}^{-1} \right) \right) \boldsymbol{V}, \tag{25}
$$

where $\mathbf{1} \in \mathbb{R}^T$ is a vector of ones. The logarithmic forget gate activation matrix $\log \mathbf{F} \in \mathbb{R}^{T \times T}$ is computed by

$$
\log \mathbf{F}_{ij} = \begin{cases} -\infty & \text{for } i < j \\ 0 & \text{for } i = j \\ \log \left( \prod_{k=j+1}^{i} \sigma \left( \tilde{\mathrm{f}}_k \right) \right) = \sum_{k=j+1}^{i} \log \sigma \left( \tilde{\mathrm{f}}_k \right) & \text{for } i > j \end{cases}. \tag{26}
$$

Similarly, the input gate pre-activation matrix $\tilde{\mathbf{I}} \in \mathbb{R}^{T \times T}$ is given by

$$
\tilde{\mathbf{I}}_{ij} = \begin{cases} 0 & \text{for } i < j \\ \tilde{\mathrm{i}}_j & \text{for } i \geqslant j \end{cases}. \tag{27}
$$

Note that in contrast to the recurrent formulation, in the parallel formulation the states $C_t$ are not materialized, i.e. computed explicitly. This comes at the cost of computing the quadratic matrices $\mathbf{D}, \mathbf{S} \in \mathbb{R}^{T \times T}$, with an overall quadratic scaling in sequence length $T$.

## B.2 Detailed Chunkwise-Parallel Formulation

In this section, we provide more detailed formulas for the chunkwise-parallel formulation of the mLSTM from Section 2.2.

**Chunkwise Gates.** Given the logarithmic forget gates $\bar{\mathbf{f}}^{(k)} = \log \sigma(\tilde{\mathbf{f}}^{(k)}) \in \mathbb{R}^L$ and input gates $\bar{\mathbf{i}}^{(k)} = \log \exp(\tilde{\mathbf{i}}^{(k)}) \in \mathbb{R}^L$, we can compute the logarithmic chunkwise gates as

$$
\mathrm{g}_k = \mathrm{sum}\left( \bar{\mathbf{f}}^{(k)} \right) = \sum_{i=1}^{L} \bar{\mathrm{f}}_i^{(k)} \quad \in \mathbb{R}, \tag{28}
$$

$$
\mathbf{b}_k = \mathrm{cumsum}\left( \bar{\mathbf{f}}^{(k)} \right) \in \mathbb{R}^L, \quad \text{with} \quad \mathrm{b}_{k,j} = \sum_{i=1}^{j} \bar{\mathrm{f}}_i^{(k)} \text{ for } j = 1, 2, \ldots, L \tag{29}
$$

$$
\mathbf{a}_k = \mathrm{rev\_cumsum}\left( \bar{\mathbf{f}}^{(k)}[1{:}] \right) + \bar{\mathbf{i}}^{(k)} \in \mathbb{R}^L, \quad \text{with} \quad \mathrm{a}_{k,j} = \sum_{i=j+1}^{L} \bar{\mathrm{f}}_i^{(k)} + \bar{\mathrm{i}}_j^{(k)} \text{ for } j = 1, 2, \ldots, L, \tag{30}
$$

where $[1{:}]$ denotes (in numpy notation) that the first index is excluded as it is done in the sum notation of Equation (30). Additionally, in Figure 9 we illustrate the chunkwise gate computation and show a PyTorch code snippet for computing the chunkwise gates in Figure 8.

```python
def compute_chunkwise_log_gates_vecB_vecA(
    vecI: torch.Tensor,  # (B, NH, S)
    vecF: torch.Tensor,  # (B, NH, S)
    chunk_size: int,
):
    B, NH, S = vecI.shape
    assert S % chunk_size == 0, f"S={S} is not divisible by chunk_size={
        chunk_size}"
    _device = vecI.device
    NC = S // chunk_size
    L = chunk_size

    # compute vecB
    vecF_logsig = logsigmoid(vecF.to(dtype=torch.float32))
    vecF_logsig_chunked = rearrange(vecF_logsig, "b nh (nc l) -> b nh nc
        l", nc=NC, l=L)
    vecB = vecF_logsig_chunked.cumsum(dim=-1)

    # compute vecA
    vecI_chunked = rearrange(vecI, "b nh (nc l) -> b nh nc l", nc=NC, l=L
        )
    # unstable vecA computation:
    # vecA = (vecB[..., -1, None] - vecB) + vecI  # (B, NH, NC, L)
    # stable vecA computation:
    vecA = (
        torch.cat(
            [
                vecF_logsig_chunked[..., 1:].flip(-1).cumsum(-1).flip(-1)
                ,
                torch.zeros((B, NH, NC, 1), device=_device, dtype=torch.
                    float32),
            ],
            dim=-1,
        )
        + vecI_chunked
    )  # (B, NH, NC, L)
    return vecB, vecA
```

Figure 8: PyTorch function to compute the logarithmic chunkwise gates for mLSTMexp.

**Inter-chunk Recurrent Contribution.** The inter-chunk recurrence is given by

$$m_k^{(\text{inter})} = \max\left\{\mathrm{g}_k + m_{k-1}^{(\text{inter})}, \ \max \mathbf{a}_k\right\} \tag{31}$$

$$\boldsymbol{C}_k = \exp\left(\mathrm{g}_k + m_{k-1}^{(\text{inter})} - m_k^{(\text{inter})}\right) \boldsymbol{C}_{k-1} + \left(\exp\left(\mathbf{a}_k - m_k^{(\text{inter})}\right) \odot \boldsymbol{K}^{(k)}\right)^\top \boldsymbol{V}^{(k)} \tag{32}$$

$$\boldsymbol{n}_k = \exp\left(\mathrm{g}_k + m_{k-1}^{(\text{inter})} - m_k^{(\text{inter})}\right) \boldsymbol{n}_{k-1} + \left(\exp\left(\mathbf{a}_k - m_k^{(\text{inter})}\right) \odot \boldsymbol{K}^{(k)}\right)^\top \mathbf{1}. \tag{33}$$

In simplified form we can write the inter-chunk recurrence as

$$\boldsymbol{C}_k = \bar{\mathrm{g}}_k \boldsymbol{C}_{k-1} + \left(\overline{\mathbf{a}}_k \odot \boldsymbol{K}^{(k)}\right)^\top \boldsymbol{V}^{(k)} \qquad\qquad = \bar{\mathrm{g}}_k \boldsymbol{C}_{k-1} + \overline{\boldsymbol{K}}^{(k)\top} \boldsymbol{V}^{(k)} \tag{34}$$

$$\boldsymbol{n}_k = \bar{\mathrm{g}}_k \boldsymbol{n}_{k-1} + \left(\overline{\mathbf{a}}_k \odot \boldsymbol{K}^{(k)}\right)^\top \mathbf{1} \qquad\qquad = \bar{\mathrm{g}}_k \boldsymbol{n}_{k-1} + \overline{\boldsymbol{K}}^{(k)\top} \boldsymbol{V}^{(k)}. \tag{35}$$

with the running max state integrated into the gates.

**chunkwise recurrent:**

$$\boldsymbol{C}_k = \boxed{\mathrm{g}_k}\, \boldsymbol{C}_{k-1} + \left(\boxed{\mathbf{a}_k} \odot \boldsymbol{K}^{(k)}\right)^{\top} \boldsymbol{V}^{(k)}$$

$$\mathbf{H}^{(k)} = \left(\boxed{\mathbf{b}_k} \odot \boldsymbol{Q}^{(k)}\right) \boldsymbol{C}_{k-1}$$

**fully recurrent:**

$$C_t = \mathrm{f}_t\, C_{t-1} + \mathrm{i}_t\, k_t v_t^{\top}$$
$$h_t = C_t^{\top} q_t$$

chunk size L=4

$$\log \mathbf{b}_1 = \begin{pmatrix} \mathrm{f}_1 \\ \mathrm{f}_1 + \mathrm{f}_2 \\ \mathrm{f}_1 + \mathrm{f}_2 + \mathrm{f}_3 \\ \mathrm{f}_1 + \mathrm{f}_2 + \mathrm{f}_3 + \mathrm{f}_4 \end{pmatrix} \qquad \log \mathbf{b}_2 = \begin{pmatrix} \mathrm{f}_5 \\ \mathrm{f}_5 + \mathrm{f}_6 \\ \mathrm{f}_5 + \mathrm{f}_6 + \mathrm{f}_7 \\ \mathrm{f}_5 + \mathrm{f}_6 + \mathrm{f}_7 + \mathrm{f}_8 \end{pmatrix} \qquad \log \mathbf{b}_3 = \begin{pmatrix} \mathrm{f}_9 \\ \mathrm{f}_9 + \mathrm{f}_{10} \\ \mathrm{f}_9 + \mathrm{f}_{10} + \mathrm{f}_{11} \\ \mathrm{f}_9 + \mathrm{f}_{10} + \mathrm{f}_{11} + \mathrm{f}_{12} \end{pmatrix}$$

$$\log \mathbf{a}_1 = \begin{pmatrix} \mathrm{f}_2 + \mathrm{f}_3 + \mathrm{f}_4 \\ \mathrm{f}_3 + \mathrm{f}_4 \\ \mathrm{f}_4 \\ 0 \end{pmatrix} + \begin{pmatrix} i_1 \\ i_2 \\ i_3 \\ i_4 \end{pmatrix} \qquad \log \mathbf{a}_2 = \begin{pmatrix} \mathrm{f}_6 + \mathrm{f}_7 + \mathrm{f}_8 \\ \mathrm{f}_3 + \mathrm{f}_8 \\ \mathrm{f}_8 \\ 0 \end{pmatrix} + \begin{pmatrix} i_5 \\ i_6 \\ i_3 \\ i_8 \end{pmatrix} \qquad \log \mathbf{a}_3 = \begin{pmatrix} \mathrm{f}_9 + \mathrm{f}_{10} + \mathrm{f}_{12} \\ \mathrm{f}_{11} + \mathrm{f}_{12} \\ \mathrm{f}_{12} \\ 0 \end{pmatrix} + \begin{pmatrix} i_9 \\ i_{10} \\ i_{11} \\ i_{12} \end{pmatrix}$$

$$\log \mathrm{g}_1 = \mathrm{f}_1 + \mathrm{f}_2 + \mathrm{f}_3 + \mathrm{f}_4 \qquad\qquad \log \mathrm{g}_2 = \mathrm{f}_5 + \mathrm{f}_6 + \mathrm{f}_7 + \mathrm{f}_8 \qquad\qquad \log \mathrm{g}_3 = \mathrm{f}_9 + \mathrm{f}_{10} + \mathrm{f}_{11} + \mathrm{f}_{12}$$

Figure 9: Illustration of the chunkwise gate computation.

**Intra-chunk Parallel Contribution.** The recurrent part is followed by the intra-chunk parallel contribution given by

$$\widetilde{\mathbf{D}}^{(k)} = \begin{cases} -\infty & \text{for } i < j \\ \mathbf{b}_k - \mathbf{b}_k^{\top} + \bar{\mathbf{i}}^{(k)\top} & \text{for } i \geqslant j \end{cases} \tag{36}$$

$$\boldsymbol{m}_k^{(\mathrm{intra})} = \max_j \widetilde{\mathbf{D}}_{ij}^{(k)} \tag{37}$$

$$\mathbf{D}^{(k)} = \exp(\widetilde{\mathbf{D}}^{(k)} - \boldsymbol{m}_k^{(\mathrm{intra})}) \tag{38}$$

$$\mathbf{S}^{(k)} = \frac{1}{\sqrt{d_{qk}}} \boldsymbol{Q}^{(k)} \boldsymbol{K}^{(k)\top} \tag{39}$$

$$\overline{\mathbf{S}}^{(k)} = \mathbf{S}^{(k)} \odot \mathbf{D}^{(k)}. \tag{40}$$

where $\exp$ is acting component-wise.

**Output computation.** The contributions from the intra-chunk parallel part $\mathbf{H}_{\mathrm{intra}}^{(k)}$ are combined with the inter-chunk recurrent part $\mathbf{H}_{\mathrm{inter}}^{(k)}$ to obtain the hidden states $\mathbf{H}^{(k)}$ for each chunk $k$ (see

Figure 1):

$$m_k^{(\text{combine})} = \max\left\{ \mathbf{b}_k + m_{k-1}^{(\text{inter})}, m_k^{(\text{intra})} \right\} \tag{41}$$

$$\mathbf{H}_{\text{inter}}^{(k)} = \left( \exp\left( \mathbf{b}_k + m_{k-1}^{(\text{inter})} - m_k^{(\text{combine})} \right) \odot \frac{\boldsymbol{Q}^{(k)}}{\sqrt{d_{qk}}} \right) \boldsymbol{C}_{k-1} \tag{42}$$

$$= \left( \overline{\mathbf{b}}_k \odot \frac{\boldsymbol{Q}^{(k)}}{\sqrt{d_{qk}}} \right) \boldsymbol{C}_{k-1} \tag{43}$$

$$= \overline{\boldsymbol{Q}}^{(k)} \boldsymbol{C}_{k-1} \tag{44}$$

$$\mathbf{H}_{\text{intra}}^{(k)} = \overline{\mathbf{S}}^{(k)} \boldsymbol{V}^{(k)} \tag{45}$$

$$\mathbf{H}^{(k)} = \frac{\left( \overline{\mathbf{b}}_k \odot (\boldsymbol{Q}^{(k)}/\sqrt{d_{qk}}) \right) \boldsymbol{C}_{k-1} + \overline{\mathbf{S}}^{(k)} \boldsymbol{V}^{(k)}}{\max\left\{ | \left( \overline{\mathbf{b}}_k \odot (\boldsymbol{Q}^{(k)}/\sqrt{d_{qk}}) \right) \boldsymbol{n}_{k-1} + \overline{\mathbf{S}}^{(k)} \mathbf{1} |, \ \exp\left( -m_k^{(\text{combine})} \right) \right\}} \tag{46}$$

$$= \frac{\overline{\boldsymbol{Q}}^{(k)} \boldsymbol{C}_{k-1} + \overline{\mathbf{S}}^{(k)} \boldsymbol{V}^{(k)}}{\max\left\{ |\overline{\boldsymbol{Q}}^{(k)} \boldsymbol{n}_{k-1} + \overline{\mathbf{S}}^{(k)} \mathbf{1}|, \ \exp\left( -m_k^{(\text{combine})} \right) \right\}} \tag{47}$$

$$= \left( \overline{\boldsymbol{Q}}^{(k)} \boldsymbol{C}_{k-1} + \overline{\mathbf{S}}^{(k)} \boldsymbol{V}^{(k)} \right) / \mathbf{h}_{\text{denom}}^{(k)}. \tag{48}$$

### B.3 Chunkwise-Parallel Backward Pass

In this section we provide a detailed description of the backward pass of the chunkwise-parallel mLSTM.

**Gradients Through Normalizer States.** Following Sun et al. (2023), we do not compute the gradients through the normalizer states $\boldsymbol{n}$. The gradients cancel out due to the Layer- or RMS-Norm on the mLSTM cell hidden states $\mathbf{H}$, since the normalizer state is constant over the embedding or feature dimension, which is the normalization dimension.

**Inter-chunk Recurrent Backward Pass.** Given the incoming memory cell state gradients from the next chunk $\delta \boldsymbol{C}_k$ and the hidden state output gradients $\delta \mathbf{H}^{(k)}$ for chunk $k$, we can compute the inter-chunk recurrent backward pass. The query, key and value gradients $\delta \boldsymbol{Q}_{\text{inter}}^{(k)}, \delta \boldsymbol{K}_{\text{inter}}^{(k)}$ and $\delta \boldsymbol{V}_{\text{inter}}^{(k)}$ of the inter-chunk recurrent part are computed by:

$$\delta \widetilde{\mathbf{H}}^{(k)} = \frac{\delta \mathbf{H}^{(k)}}{\mathbf{h}_{\text{denom}}^{(k)}} \tag{49}$$

$$\delta \boldsymbol{V}_{\text{inter}}^{(k)} = \overline{\boldsymbol{K}}^{(k)} \, \delta \boldsymbol{C}_k \tag{50}$$

$$\delta \overline{\boldsymbol{K}}^{(k)} = \boldsymbol{V}^{(k)} \, \delta \boldsymbol{C}_k^\top \tag{51}$$

$$\delta \boldsymbol{K}_{\text{inter}}^{(k)} = \delta \overline{\boldsymbol{K}}^{(k)} \odot \overline{\boldsymbol{a}}_k \, \mathbf{1}^\top \tag{52}$$

$$\delta \overline{\boldsymbol{Q}}^{(k)} = \delta \widetilde{\mathbf{H}}^{(k)} \, \boldsymbol{C}_{k-1}^\top \tag{53}$$

$$\delta \boldsymbol{Q}_{\text{inter}}^{(k)} = \frac{1}{\sqrt{d_{qk}}} \, \delta \overline{\boldsymbol{Q}}^{(k)} \odot \overline{\boldsymbol{b}}_k \, \mathbf{1}^\top \tag{54}$$

The memory cell state gradients $\delta \boldsymbol{C}_{k-1}$ have incoming contributions from the next timestep $\delta \boldsymbol{C}_{k-1}^{(\text{rec})}$ and output $\delta \boldsymbol{C}_{k-1}^{(\text{out})}$. They are given as

$$\delta \boldsymbol{C}_{k-1} = \delta \boldsymbol{C}_{k-1}^{(\text{rec})} + \delta \boldsymbol{C}_{k-1}^{(\text{out})} \tag{55}$$

$$= \overline{\mathbf{g}} \odot \delta \boldsymbol{C}_k + \overline{\boldsymbol{Q}}^{(k)\top} \delta \widetilde{\mathbf{H}}^{(k)}. \tag{56}$$

Finally, we can compute the cumulative gate gradients $\delta\overline{g}_k$, $\delta\boldsymbol{a}_k$ and $\delta\boldsymbol{b}_k$ for chunk $k$ as

$$\delta\overline{g}_k = \mathbf{1}^\top (\boldsymbol{C}_{k-1} \odot \delta\boldsymbol{C}_k)\, \mathbf{1} \tag{57}$$

$$\delta g_k = \delta\overline{g}_k \odot \overline{g}_k \tag{58}$$

$$\delta\overline{\boldsymbol{a}}_k = (\delta\overline{\boldsymbol{K}}^{(k)} \odot \boldsymbol{K}^{(k)})\, \mathbf{1} \tag{59}$$

$$\delta\boldsymbol{a}_k = \delta\overline{\boldsymbol{a}}_k \odot \overline{\boldsymbol{a}}_k \tag{60}$$

$$\delta\overline{\boldsymbol{b}}_k = (\delta\overline{\boldsymbol{Q}}^{(k)} \odot \frac{\boldsymbol{Q}^{(k)}}{\sqrt{d_{qk}}})\, \mathbf{1} \tag{61}$$

$$\delta\boldsymbol{b}_k = \delta\overline{\boldsymbol{b}}_k \odot \overline{\boldsymbol{b}}_k. \tag{62}$$

**Intra-chunk Parallel Backward Pass.** Given the mLSTM hidden state output gradients $\delta\mathbf{H}^{(k)}$ the intra chunk query, key and value gradients $\delta\boldsymbol{Q}_{\text{intra}}^{(k)}$, $\delta\boldsymbol{K}_{\text{intra}}^{(k)}$ and $\delta\boldsymbol{V}_{\text{intra}}^{(k)}$ gradients are computed by

$$\delta\widetilde{\mathbf{H}}^{(k)} = \frac{\delta\mathbf{H}^{(k)}}{\mathbf{h}_{\text{denom}}^{(k)}} \tag{63}$$

$$\mathbf{S}^{(k)} = \frac{1}{\sqrt{d_{qk}}} \boldsymbol{Q}^{(k)} \boldsymbol{K}^{(k)\top} \tag{64}$$

$$\overline{\mathbf{S}}^{(k)} = \mathbf{S}^{(k)} \odot \mathbf{D}^{(k)} \tag{65}$$

$$\delta\boldsymbol{V}_{\text{intra}}^{(k)} = \overline{\mathbf{S}}^{(k)\top} \delta\widetilde{\mathbf{H}}^{(k)} \tag{66}$$

$$\delta\overline{\mathbf{S}}^{(k)} = \delta\widetilde{\mathbf{H}}^{(k)} \boldsymbol{V}^{(k)\top} \tag{67}$$

$$\delta\mathbf{S}^{(k)} = \delta\overline{\mathbf{S}}^{(k)} \odot \mathbf{D}^{(k)} \tag{68}$$

$$\delta\boldsymbol{Q}_{\text{intra}}^{(k)} = \frac{1}{\sqrt{d_{qk}}} \delta\mathbf{S}^{(k)} \boldsymbol{K}^{(k)} \tag{69}$$

$$\delta\boldsymbol{K}_{\text{intra}}^{(k)} = \frac{1}{\sqrt{d_{qk}}} \delta\mathbf{S}^{(k)\top} \boldsymbol{Q}^{(k)} \tag{70}$$

In order to compute the cumulative intra gate gradients, we compute the gradients through the gate matrix $\mathbf{D}^{(k)}$, which is computed from the cumulative forget gates

$$\boldsymbol{b}_k^{(q)} = \text{cumsum}(\overline{\mathbf{f}}_q^{(k)}) \in \mathbb{R}^{L_q} \tag{71}$$

$$\boldsymbol{b}_k^{(kv)} = \text{cumsum}(\overline{\mathbf{f}}_{kv}^{(k)}) \in \mathbb{R}^{L_{kv}}, \tag{72}$$

where we use the logarithmic forget gates $\overline{\mathbf{f}} = \log\sigma(\tilde{\mathbf{f}})$. We denote the dimensions as $L_q$ and $L_{kv}$ for the query and key-value dimensions, respectively. Omitting the masking operation, we compute the gate matrix as

$$\mathbf{D}^{(k)} = \boldsymbol{b}_k^{(q)} \mathbf{1}_{kv}^\top - \mathbf{1}_q \boldsymbol{b}_k^{(kv)\top} + \mathbf{1}_q \overline{\mathbf{i}}_{kv}^{(k)\top}, \tag{73}$$

where $\mathbf{1}_q \in \mathbb{R}^{L_q}$ and $\mathbf{1}_{kv} \in \mathbb{R}^{L_{kv}}$ are vectors of ones used to indicate broadcast operations, and $\overline{\mathbf{i}}_{kv}^{(k)} \in \mathbb{R}^{L_{kv}}$ are the logarithmic input gates for chunk $k$.

The gradients are computed as

$$\delta\mathbf{D}^{(k)} = \delta\overline{\mathbf{S}}^{(k)} \odot \mathbf{S}^{(k)} \tag{74}$$

$$\delta\boldsymbol{b}_k^{(q)} = \delta\mathbf{D}^{(k)} \mathbf{1}_{kv} \tag{75}$$

$$\delta\boldsymbol{b}_k^{(kv)} = -\delta\mathbf{D}^{(k)\top} \mathbf{1}_q \tag{76}$$

$$\delta\overline{\mathbf{i}}_{kv}^{(k)} = \delta\mathbf{D}^{(k)\top} \mathbf{1}_q. \tag{77}$$

**Combined input and gate gradients.** The intra and inter chunk gradients are combined by summing up the contributions. This yields for the query, key and value gradients

$$\delta \boldsymbol{Q}^{(k)} = \delta \boldsymbol{Q}_{\text{inter}}^{(k)} + \delta \boldsymbol{Q}_{\text{intra}}^{(k)} \tag{78}$$

$$\delta \boldsymbol{K}^{(k)} = \delta \boldsymbol{V}_{\text{inter}}^{(k)} + \delta \boldsymbol{K}_{\text{intra}}^{(k)} \tag{79}$$

$$\delta \boldsymbol{V}^{(k)} = \delta \boldsymbol{V}_{\text{inter}}^{(k)} + \delta \boldsymbol{V}_{\text{intra}}^{(k)}. \tag{80}$$

The input and forget gate gradients $\bar{\mathbf{i}}^{(k)}$ and $\bar{\mathbf{f}}^{(k)}$ can be computed from the cumulative gate gradients $\delta \mathrm{g}_k$, $\delta \boldsymbol{b}_k$ and $\delta \boldsymbol{a}_k$ with the following equalities

$$\delta \bar{\mathbf{f}}^{(k)} = \delta \mathrm{g}_k \tag{81}$$

$$\delta \bar{\mathbf{f}}^{(k)} = \text{rev\_cumsum}(\delta \boldsymbol{b}_k) \tag{82}$$

$$\delta \bar{\mathbf{f}}^{(k)} = \text{rev\_cumsum}(\delta \boldsymbol{a}_k) \tag{83}$$

$$\delta \bar{\mathbf{i}}^{(k)} = \delta \boldsymbol{a}_k \tag{84}$$

# C  Extended Tiled Flash Linear Attention

## C.1  GPU Fundamentals

We review the GPU fundamentals for writing efficient kernels. Since we perform our experiments on NVIDIA GPUs, our review is targeted towards NVIDIA's terminology, though the principles also apply to other hardware. For a more extensive overview we refer to (Spector et al., 2024).

**GPU Overview.**  A GPU (Graphics Processing Unit) is a specialized processor designed to efficiently handle large-scale parallel computation tasks, such as matrix multiplications in neural networks. These tasks are divided into small programs called kernels, that are executed on GPUs. A kernel loads data from high bandwidth memory (HBM), performs work on it, and writes the results back to HBM. For writing efficient kernels, it is important to understand the software hierarchy of the GPU, which closely follows its physical hardware hierarchy.

**GPU Hierarchy.**  At the lowest level the GPU runs multiple Threads, operating on small but fast register memory in parallel. On the software side usually multiple (e.g. 32) Threads are grouped together into Warps. Again, multiple Warps are grouped into Thread blocks which together execute a kernel on a physical core, called streaming multiprocessor (SM). Warps or Threads within the same Thread block can communicate data through special on-chip shared memory (SRAM). When executing a kernel, a grid (with typically 3 dimensions) of Thread blocks that run in parallel is launched on the GPU. All Thread blocks have access to the large but slow off-chip high-bandwidth memory (HBM), which has both the largest latency and least bandwidth of all GPU memories. *For efficient kernels it is important to minimize memory read and writes from and to HBM.*

**Specialized Compute Units.** Modern GPUs have specialized compute units – called tensor cores – that accelerate matrix multiplications on GPUs. Tensor cores have most of the GPU compute and are accessed at the warp or block level. *For efficient kernels it is important to maximize tensor core utilization.*

**Triton Language.** Triton is a GPU kernel programming language with an associated compiler, that provides a Python-based environment for GPU programming. The user can load data from HBM via a `tl.load` instruction and store data to HBM via `tl.store`. `tl.dot` is an instruction, that leverages tensor cores for matrix multiplications. While this Triton interface of increases productivity in writing very fast custom kernels, peak performance can be achieved sometimes only with CUDA kernels. We write our kernels in Triton and leave a CUDA implementation for future work. In contrast to NVIDIAs programming model CUDA, which provides access to all levels of the GPU hierarchy, Triton programs operate on the Thread block level and hide register and thread management from the user. Therefore, we describe TFLA on the more abstract Thread block or program level in the following section.

## C.2 Tiled Computation

For the tiled computation of the intra-chunk hidden state contribution $\mathbf{H}_{\text{intra}}$ within a chunk, we consider blocks of the matrix $\mathbf{S} = \begin{bmatrix} \mathbf{S}^{(1)} & \mathbf{S}^{(2)} \end{bmatrix}$ and the gate matrix $\mathbf{D} = \begin{bmatrix} \mathbf{D}^{(1)} & \mathbf{D}^{(2)} \end{bmatrix}$, with $\mathbf{S}^{(i)}, \mathbf{D}^{(i)} \in \mathbb{R}^{B_{Lhq} \times B_{Lkv}}$. Here, the superscript $i$ denotes the block index along the $L_{kv}$ dimension (and not the chunk index). Similarly, we consider blocks of the value matrix $\boldsymbol{V} = \begin{bmatrix} \boldsymbol{V}^{(1)} \\ \boldsymbol{V}^{(2)} \end{bmatrix}$, with $\boldsymbol{V}^{(i)} \in \mathbb{R}^{B_{kv} \times B_{dhv}}$. We then accumulate the unnormalized hidden state blocks $\mathbf{H}_{\text{intra,num}}^{(i)} \in \mathbb{R}^{B_{Lkv} \times B_{dhv}}$ and the corresponding normalizer $\boldsymbol{l}^{(i)} \in B_{Lkv}$ as

$$\boldsymbol{m}^{(1)} = \max_j \widetilde{\mathbf{D}}_{ij}^{(1)} \tag{85}$$

$$\boldsymbol{l}^{(1)} = (\mathbf{S}^{(1)} \odot \exp(\widetilde{\mathbf{D}}^{(1)} - \boldsymbol{m}^{(1)}))\,\mathbf{1} \tag{86}$$

$$\mathbf{H}_{\text{intra,num}}^{(1)} = (\mathbf{S}^{(1)} \odot \exp(\widetilde{\mathbf{D}}^{(1)} - \boldsymbol{m}^{(1)}))\,\boldsymbol{V}^{(1)} \tag{87}$$

$$\boldsymbol{m}^{(2)} = \max\left(\boldsymbol{m}^{(1)},\ \max_j \widetilde{\mathbf{D}}_{ij}^{(2)}\right) \tag{88}$$

$$\boldsymbol{l}^{(2)} = \exp(\boldsymbol{m}^{(1)} - \boldsymbol{m}^{(2)})\,\boldsymbol{l}^{(1)} + (\mathbf{S}^{(2)} \odot \exp(\widetilde{\mathbf{D}}^{(2)} - \boldsymbol{m}^{(2)}))\,\mathbf{1} \tag{89}$$

$$\mathbf{H}_{\text{intra,num}}^{(2)} = \exp(\boldsymbol{m}^{(1)} - \boldsymbol{m}^{(2)})\,\mathbf{H}_{\text{intra,num}}^{(1)} + (\mathbf{S}^{(2)} \odot \exp(\widetilde{\mathbf{D}}^{(2)} - \boldsymbol{m}^{(2)}))\,\boldsymbol{V}^{(2)}. \tag{90}$$

After computing this intra-chunk part, we need to do one more rescaling step to combine the intra-chunk and inter-chunk parts of the hidden state output $\mathbf{H}^{(k)}$ since $\mathbf{H}_{\text{intra}}^{(k)}$ and $\mathbf{H}_{\text{inter}}^{(k)}$ were computed with different max states. Therefore, we compute the final hidden state output $\mathbf{H}^{(k)}$ as

$$\boldsymbol{m}_k^{(\text{combine})} = \max\left\{\mathbf{b}_k + m_{k-1}^{(\text{inter})}, m_k^{(2)}\right\} \tag{91}$$

$$\mathbf{H}^{(k)} = \frac{\overline{\boldsymbol{Q}}^{(k)} \boldsymbol{C}_{k-1} + \exp\left(\boldsymbol{m}_k^{(2)} - \boldsymbol{m}_k^{(\text{combine})}\right) \overline{\mathbf{S}}^{(k)} \boldsymbol{V}^{(k)}}{\max\left\{|\overline{\boldsymbol{Q}}^{(k)} \boldsymbol{n}_{k-1} + \exp\left(\boldsymbol{m}_k^{(2)} - \boldsymbol{m}_k^{(\text{combine})}\right) \boldsymbol{l}_k^{(2)}|,\ \exp\left(-\boldsymbol{m}_k^{(\text{combine})}\right)\right\}}, \tag{92}$$

where we assume that $\boldsymbol{m}_k^{(2)}$ is the block maximum and $\boldsymbol{l}_k^{(2)}$ is the normalizer after the last $B_{Lkv}$ block of the intra-chunk computation for chunk $k$.

## C.3 TFLA Forward Pass

For notational simplicity we drop the $k$ index for the query, key and value matrices as $\boldsymbol{Q} \in \mathbb{R}^{L_{hq} \times d_{qk}}$, $\boldsymbol{K} \in \mathbb{R}^{L_{kv} \times d_{qk}}$ and $\boldsymbol{V} \in \mathbb{R}^{L_{kv} \times d_v}$, respectively. We make use of reweighting (as discussed in Appendix C.2) in order to keep track of the maximum value over the gate matrix tiles, similar to (Dao et al., 2022).

The forward pass algorithm of TFLA for one thread block is described in Algorithm 1.

Note that the loop in line 27 of Algorithm 1 is the same as the loop in line 6. In both loops we load the same blocks of the matrix $\boldsymbol{Q}$. Fusing these loops would avoid loading this data twice. Unfortunately, fusing these loops efficiently is problematic due to the online computation of the maximum $\boldsymbol{m}_{old}$ and $\boldsymbol{m}_{new}$ in the loop in line 4 and the dependence of $\boldsymbol{m}_k^{(\text{combine})}$ and $\overline{\mathbf{b}}_k$ on the final $\boldsymbol{m}_{new}$ (see Appendix D.1 and C.2).

We address this issue in Section 4 by modifying the input gate of the mLSTM.

**Algorithm 1** TFLA Intra-Chunk Forward Pass for mLSTMexp ($\mathbf{H}^{(k)}$ Kernel)

---

**Require:** Matrices $\boldsymbol{Q} \in \mathbb{R}^{L_{hq} \times d_{qk}}$, $\boldsymbol{K} \in \mathbb{R}^{L_{kv} \times d_{qk}}$, $\boldsymbol{V} \in \mathbb{R}^{L_{kv} \times d_{hv}}$.
    States $\boldsymbol{C}_{k-1} \in \mathbb{R}^{d_{qk} \times d_v}$, $\boldsymbol{n}_{k-1} \in \mathbb{R}^{d_{qk}}$.
    Input- and cumulative forget gate vectors $\mathbf{i}_k, \mathbf{b}_k \in \mathbb{R}^{L_{hq}}$.
    Block sizes $B_{dqk}, B_{dhv}, B_{Lhq}$ and $B_{Lkv}$, where $B_{Lhq} \geqslant B_{Lkv}$.
    Block Q index $i_{Lq}$ and Block HV index $i_{dhv}$.

1: Initialize $\boldsymbol{m}_{old}, \boldsymbol{m}_{new} \in \mathbb{R}^{L_q}$ to $-\infty$ in SRAM.
    ▷ Compute intra-chunk contribution
2: Initialize accumulators $\mathbf{H}_{\text{intra}} \in \mathbb{R}^{B_{Lhq} \times B_{dv}}$ and $\mathbf{n}^{(\text{intra})} \in \mathbb{R}^{B_{Lhq}}$ in SRAM.
3: Load $\mathbf{b}_k^{(q)} \in \mathbb{R}^{B_{Lhq}}$ from HBM to SRAM.
4: **for** $i = 1$ to $\left\lfloor \frac{(i_{Lq}+1) \cdot B_{Lhq}}{B_{Lkv}} \right\rfloor$ **do**
5:    Initialize accumulator $\mathbf{S}^{(i)} \in \mathbb{R}^{B_{Lhq} \times B_{Lkv}}$ in SRAM.
6:    **for** $j = 1$ to $\left\lceil \frac{d_{qk}}{B_{dqk}} \right\rceil$ **do**
7:        Load $\boldsymbol{Q}^{(j)} \in \mathbb{R}^{B_{Lhq} \times B_{dqk}}$ and $\boldsymbol{K}^{(j)} \in \mathbb{R}^{B_{Lkv} \times B_{dqk}}$ from HBM to SRAM.
8:        Accumulate $\mathbf{S}^{(i)} \mathrel{+}= \boldsymbol{Q}^{(j)} \boldsymbol{K}^{(j)^\top}$.
9:    **end for**
10:    Load $\mathbf{b}_k^{(kv)} \in \mathbb{R}^{B_{Lkv}}$ and $\mathbf{i}_k^{(kv)} \in \mathbb{R}^{B_{Lkv}}$ from HBM to SRAM.
11:    Compute $\widetilde{\mathbf{D}}^{(i)} = \mathbf{b}_k^{(q)} - \mathbf{b}_k^{(kv)^\top} + \mathbf{i}_k^{(kv)^\top} \in \mathbb{R}^{B_{Lhq} \times B_{Lkv}}$.
12:    **if** $i \cdot B_{Lkv} \geqslant i_{Lq} \cdot B_{Lhq}$ **then**
13:        Apply causal mask to $\widetilde{\mathbf{D}}^{(i)}$.
14:    **end if**
15:    Compute $\boldsymbol{m}_{new} = \text{maximum}\{\boldsymbol{m}_{old}, \text{rowmax}\,\widetilde{\mathbf{D}}^{(i)}\}$.
16:    Compute $\mathbf{D}^{(i)} = \exp(\widetilde{\mathbf{D}}^{(i)} - \boldsymbol{m}_{new})$.
17:    Compute $\overline{\mathbf{S}}^{(i)} = \frac{1}{\sqrt{d_{qk}}} \mathbf{S} \odot \mathbf{D}^{(i)}$.
18:    Load $\boldsymbol{V}^{(i)} \in \mathbb{R}^{B_{Lkv} \times B_{dhv}}$ for Block $i_{dhv}$ from HBM to SRAM.
19:    Accumulate $\mathbf{H}_{\text{intra}} = \exp(\boldsymbol{m}_{old} - \boldsymbol{m}_{new}) \cdot \mathbf{H}_{\text{intra}} + \overline{\mathbf{S}}^{(i)} \boldsymbol{V}$.
20:    Accumulate $\mathbf{n}^{(\text{intra})} = \exp(\boldsymbol{m}_{old} - \boldsymbol{m}_{new}) \cdot \mathbf{n}^{(\text{intra})} + \overline{\mathbf{S}}^{(i)} \mathbf{1}$.
21:    Update $\boldsymbol{m}_{old} = \boldsymbol{m}_{new}$.
22: **end for**
    ▷ Compute inter-chunk contribution
23: Load $m_{k-1}^{(\text{inter})} \in \mathbb{R}$ from HBM to SRAM.
24: Compute $\boldsymbol{m}_k^{(\text{combine})} = \text{maximum}\left\{\mathbf{b}_k^{(q)} + m_{k-1}^{(\text{inter})}, \boldsymbol{m}_{new}\right\}$.
25: Compute $\overline{\mathbf{b}}_k = \exp\left(\mathbf{b}_k^{(q)} + m_{k-1}^{(\text{inter})} - \boldsymbol{m}_k^{(\text{combine})}\right)$.
26: Initialize accumulators $\mathbf{H}_{\text{inter}} \in \mathbb{R}^{B_{Lhq} \times B_{dhv}}$ for Block $i_{dhv}$ and $\mathbf{n}^{(\text{inter})} \in \mathbb{R}^{B_{Lhq}}$ in SRAM.
    ▷ Note: This is the same loop as the inner one above. They cannot be merged because of the max state computation.
27: **for** $j = 1$ to $\left\lceil \frac{d_{qk}}{B_{dqk}} \right\rceil$ **do**
28:    Load $\boldsymbol{Q}^{(j)} \in \mathbb{R}^{B_{Lhq} \times B_{dqk}}$ and $\boldsymbol{C}_{k-1}^{(j)} \in \mathbb{R}^{B_{dqk} \times B_{dhv}}$ for Block $i_{dhv}$ from HBM to SRAM.
29:    Compute $\overline{\boldsymbol{Q}}^{(j)} = \frac{1}{\sqrt{d_{qk}}} \boldsymbol{Q}^{(j)} \odot \mathbf{b}_k^{(q)}$.
30:    Accumulate $\mathbf{H}_{\text{inter}} \mathrel{+}= \overline{\boldsymbol{Q}}^{(j)} \boldsymbol{C}_{k-1}^{(j)}$.
31:    Load $\boldsymbol{n}_{k-1}^{(j)} \in \mathbb{R}^{B_{dqk}}$.
32:    Accumulate $\mathbf{n}^{(\text{inter})} \mathrel{+}= \overline{\boldsymbol{Q}}^{(j)} \boldsymbol{n}_{k-1}^{(j)}$.
33: **end for**                       ▷ Combine inter- and intra-chunk contributions
34: Compute $\mathbf{H}^{(\text{comb})} = \mathbf{H}_{\text{intra}} + \exp(\boldsymbol{m}_{new} - \boldsymbol{m}_k^{(\text{combine})}) \, \mathbf{H}_{\text{inter}}$.
35: Compute $\mathbf{n}^{(\text{comb})} = \mathbf{n}^{(\text{intra})} + \exp(\boldsymbol{m}_{new} - \boldsymbol{m}_k^{(\text{combine})}) \, \mathbf{n}^{(\text{inter})}$.
36: Compute $\mathbf{H}^{(k)} = \frac{\mathbf{H}^{(\text{comb})}}{\max\left\{|\mathbf{n}^{(\text{comb})}|, \exp(-\boldsymbol{m}_k^{(\text{combine})})\right\}}$.
37: Store $\mathbf{H}^{(k)}$, $\mathbf{n}^{(\text{comb})}$ and $\boldsymbol{m}_k^{(\text{combine})}$ to HBM.

---

## C.4 TFLA Backward Pass

For the TFLA backward pass, we need to compute the gradients of the queries, keys and values $\delta \boldsymbol{Q}^{(k)}, \delta \boldsymbol{K}^{(k)}$ and $\delta \boldsymbol{V}^{(k)}$. Omitting the gate computations and normalization, we write a simplified version of these gradients as

$$
\underset{(L_{hq} \times d_{qk})}{\delta \boldsymbol{Q}^{(k)}} = \underbrace{\left( \underset{(L_{hq} \times d_{hv})}{\delta \mathbf{H}^{(k)}} \underset{(d_{hv} \times L_{kv})}{\boldsymbol{V}^{(k)^\top}} \right) \underset{(L_{kv} \times d_{qk})}{\boldsymbol{K}^{(k)}}}_{\delta \boldsymbol{Q}^{(k)}_{\text{intra}}} + \underbrace{\underset{(L_{hq} \times d_{hv})}{\delta \mathbf{H}^{(k)}} \underset{(d_{hv} \times d_{qk})}{\boldsymbol{C}^\top_{k-1}}}_{\delta \boldsymbol{Q}^{(k)}_{\text{inter}}} \tag{93}
$$

$$
\underset{(L_{kv} \times d_{qk})}{\delta \boldsymbol{K}^{(k)}} = \underbrace{\left( \underset{(L_{kv} \times d_{hv})}{\boldsymbol{V}^{(k)}} \underset{(d_{hv} \times L_{hq})}{\delta \mathbf{H}^{(k)^\top}} \right) \underset{(L_{hq} \times d_{qk})}{\boldsymbol{Q}^{(k)}}}_{\delta \boldsymbol{K}^{(k)}_{\text{intra}}} + \underbrace{\underset{(L_{kv} \times d_{hv})}{\boldsymbol{V}^{(k)}} \underset{(d_{hv} \times d_{qk})}{\delta \boldsymbol{C}^\top_k}}_{\delta \boldsymbol{K}^{(k)}_{\text{inter}}} \tag{94}
$$

$$
\underset{(L_{kv} \times d_{hv})}{\delta \boldsymbol{V}^{(k)}} = \underbrace{\left( \underset{(L_{kv} \times d_{qk})}{\boldsymbol{K}^{(k)}} \underset{(d_{qk} \times L_{hq})}{\boldsymbol{Q}^{(k)^\top}} \right) \underset{(L_{hq} \times d_{hv})}{\delta \mathbf{H}^{(k)}}}_{\delta \boldsymbol{V}^{(k)}_{\text{intra}}} + \underbrace{\underset{(L_{kv} \times d_{qk})}{\boldsymbol{K}^{(k)}} \underset{(d_{qk} \times d_{hv})}{\delta \boldsymbol{C}_k}}_{\delta \boldsymbol{V}^{(k)}_{\text{inter}}} . \tag{95}
$$

We see that each of the query, key and value gradients has a similar structure as the forward pass in Equation (12). They can be computed with the same work partitioning scheme, where we parallelize over the outer chunk size and outer embedding dimension of the matrix multiplications and loop over the inner dimensions, respectively. For example, for the key gradients $\delta \boldsymbol{K}^{(k)}$ we parallelize over the outer chunk size $L_{kv}$ and the outer embedding dimension $d_{qk}$ and loop over the inner dimensions $L_{hq}$ and $d_{hv}$. Table 1 summarizes the TFLA work partitioning scheme for the forward and backward pass kernels.

# D Extended mLSTM with Sigmoid Input Gate

## D.1 Stabilization of the Exponential Input Gate

In this section we show how the exponential input gate is stabilized with the max state $m_t$ (Beck et al., 2024). The stabilization is based on the idea of Safe Softmax (Milakov & Gimelshein, 2018). We will see that the max state stabilization ensures that the argument of the exponential input gate activation is always smaller than 1. We will also see that the normalizer state guarantees cancellation of the max state, so that the overall outputs of the mLSTM remain unaffected by the max state.

Without stabilization mLSTM hidden state output is computed as

$$h_t = \tilde{o}_t \odot \frac{C_t^\top q_t}{\max\left\{\left|n_t^\top q_t\right|, 1\right\}}, \tag{96}$$

where we omit the scaling factor $\sqrt{d_{qk}}$ for $q$. To simplify we also omit the lower bound and the absolute value on the dot product in the denominator. We obtain

$$h_t = \sigma\left(\tilde{o}_t\right) \odot \frac{C_t^\top q_t}{n_t^\top q_t}. \tag{97}$$

Inserting the update formulas for the memory cell state $C_t$ and the normalizer state $n_t$ gives

$$h_t = \sigma\left(\tilde{o}_t\right) \odot \frac{\left(\sigma(\tilde{f}_t)\, C_{t-1} + \exp(\tilde{i}_t)\, k_t\, v_t^\top\right)^\top q_t}{\left(\sigma(\tilde{f}_t)\, n_{t-1} + \exp(\tilde{i}_t)\, k_t\right)^\top q_t}. \tag{98}$$

We now show that from this unstabilized version of the mLSTM we can derive the stabilized form in three steps. At first we use the identity $\sigma(\tilde{i}) = \exp(\log(\sigma(\tilde{f}_t)))$, extend the fraction in Equation (98) by $\exp(-m_t)$ and select $m_t = \max\{\log(\sigma(\tilde{f}_t)), \tilde{i}_t\}$ to be the maximum of the two arguments of the exponential function. This gives

$$h_t = \sigma\left(\tilde{o}_t\right) \odot \frac{C_t^\top q_t \cdot \exp(-m_t)}{n_t^\top q_t \cdot \exp(-m_t)} = \sigma\left(\tilde{o}_t\right) \odot \frac{\left(\exp(\log(\sigma(\tilde{f}_t)) - m_t)\, C_{t-1} + \exp(\tilde{i}_t - m_t)\, k_t\, v_t^\top\right)^\top q_t}{\left(\exp(\log(\sigma(\tilde{f}_t)) - m_t)\, n_{t-1} + \exp(\tilde{i}_t - m_t)\, k_t\right)^\top q_t}. \tag{99}$$

In this way, we ensure that the arguments of the exponential function are always smaller than 1, such that numerical overflow due to large values can never occur.

As next step we reparameterize $C_t$ and $n_t$ to $\tilde{C}_t$ and $\tilde{n}_t$.

$$
\begin{aligned}
\widetilde{C}_t &= C_t\, \exp(-m_t) &\rightarrow\quad \widetilde{C}_{t-1} &= C_{t-1}\, \exp(-m_{t-1}) &\Leftrightarrow\quad C_{t-1} &= \widetilde{C}_{t-1}\, \exp(m_{t-1}) \\
\tilde{n}_t &= n_t\, \exp(-m_t) &\rightarrow\quad \tilde{n}_{t-1} &= n_{t-1}\, \exp(-m_{t-1}) &\Leftrightarrow\quad n_{t-1} &= \tilde{n}_{t-1}\, \exp(m_{t-1})
\end{aligned}
\tag{100}
$$

Finally, we replace $C_t$ and $n_t$ with the stabilized states $\widetilde{C}_t$ and $\tilde{n}_t$ in the recurrence. We arrive at

$$
\begin{aligned}
h_t &= \sigma\left(\tilde{o}_t\right) \odot \frac{\left(\exp(\log(\sigma(\tilde{f}_t)) + m_{t-1} - m_t)\, \widetilde{C}_{t-1} + \exp(\tilde{i}_t - m_t)\, k_t\, v_t^\top\right)^\top q_t}{\left(\exp(\log(\sigma(\tilde{f}_t)) + m_{t-1} - m_t)\, \tilde{n}_{t-1} + \exp(\tilde{i}_t - m_t)\, k_t\right)^\top q_t} \\
&= \sigma\left(\tilde{o}_t\right) \odot \frac{\widetilde{C}_t^\top q_t}{\tilde{n}_t\, q_t}
\end{aligned}
\tag{101}
$$

Now we choose the max state as $m_t = \max\{\log(\sigma(\tilde{f}_t)) + m_{t-1}, \tilde{i}_t\}$ and arrive at the stabilized mLSTM formulas by changing the denominator to $\max\left\{\left|\tilde{n}_t^\top q_t\right|, \exp(m_{t-1})\right\}$. We have to add $\exp(m_{t-1})$ also to the right side of the maximum, so that it cancels out.

To summarize, we see that the normalizer is necessary for the max state to cancel out and the exponential input gate argument is bounded through the max state.

## D.2 Empirical Transfer Behavior Analysis of the mLSTM

We provide details on the transfer behavior analysis of mLSTMexp and mLSTMsig in Section 4.2.

**Experiment Setup.** We analyze the transfer behavior of the mLSTM for a single head and a single input sequence of length $T = 512$. The inputs are for the queries, keys and values $q_t$, $k_t$ and $v_t$ are sampled from the standard normal distribution $\mathcal{N}(0, 1)$. We set the head dimensions to $d_{qk} = 128$ and $d_{hv} = 128$. As norm layer $\text{NORM}(x)$ we use the RMS-norm. Changing the norm to layernorm does not alter the results, as for this experiment we set the mean of the inputs to zero. For every plot we measure the gains $G_{\text{before}}$ and $G_{\text{after}}$ (as defined in (16)) for input and forget gate preactivation values in the ranges [-12, 8] and [-5, 12], respectively.

**Effect of Normalization Layer Epsilon on Transfer Behavior.** Based on our analysis on the normalization layer after the gated linear RNN operation in Section 4.2, we hypothesize that the normalization layer and especially the norm epsilon $\epsilon$ is integral to the gating mechanism. In this experiment, we probe the effect of the epsilon value on the transfer behavior of the mLSTM. Figure 10a and Figure 11a show the transfer behavior of mLSTMexp and mLSTMsig for $\epsilon = $[1e-2, 1e-6, 1e-8], respectively.

We observe that the epsilon acts in the same way for mLSTMexp and mLSTMsig. Increasing $\epsilon$ causes an offset of the gain in positive y-direction, increasing $\epsilon$ in negative y-direction. We set our default value $\epsilon = $1e-6, which yields the best performance in our experiments (see Sec. 5.1).

**Normalizers of mLSTMexp and mLSTMsig.** In this experiment, we test the effect of different normalizers $n$ in Equation 24 for mLSTMexp and mLSTMsig. The parallel formulation in Section B.1 is presented for the mLSTM with exponential input gate, but applies similarly to the mLSTM with sigmoid input gate. For the default mLSTMsig, we set the normalizer to $n = 1$ and modify the calculation of the gate matrix $D$ for sigmoid input gates.

In Figure 10, we show the results of different normalizers for the mLSTM with exponential input gate. Only the default mLSTMexp with correct normalizer and max state (in Fig. 10a) shows a transfer behavior that depends on the input gate.

In contrast, in Figure 11a and 11b we observe that incorporating a normalizer similar to mLSTMexp (excluding the max state) into mLSTMsig does not alter its transfer behavior.

The other two normalizer variants for mLSTMsig in Figure 11c and 11d show a clearly different transfer behavior and do not train successfully. Similarly, the variants in Figure 10b and 10c also fail to train successfully.

In summary, we find that if the mLSTM exhibits the characteristic gate dependent transfer behavior it trains successfully and shows good performance in our language modeling experiments. In order to achieve this behavior for the mLSTMexp we need to normalize correctly as derived in Section D.1. Adding a normalizer to the mLSTMsig does not change performance and transfer behavior, if the normalizer incorporates a lower bound on the dot-product $n_t^\top q_t$. However, our default mLSTMsig omits the normalizer in order to reduce computational cost and runtime.

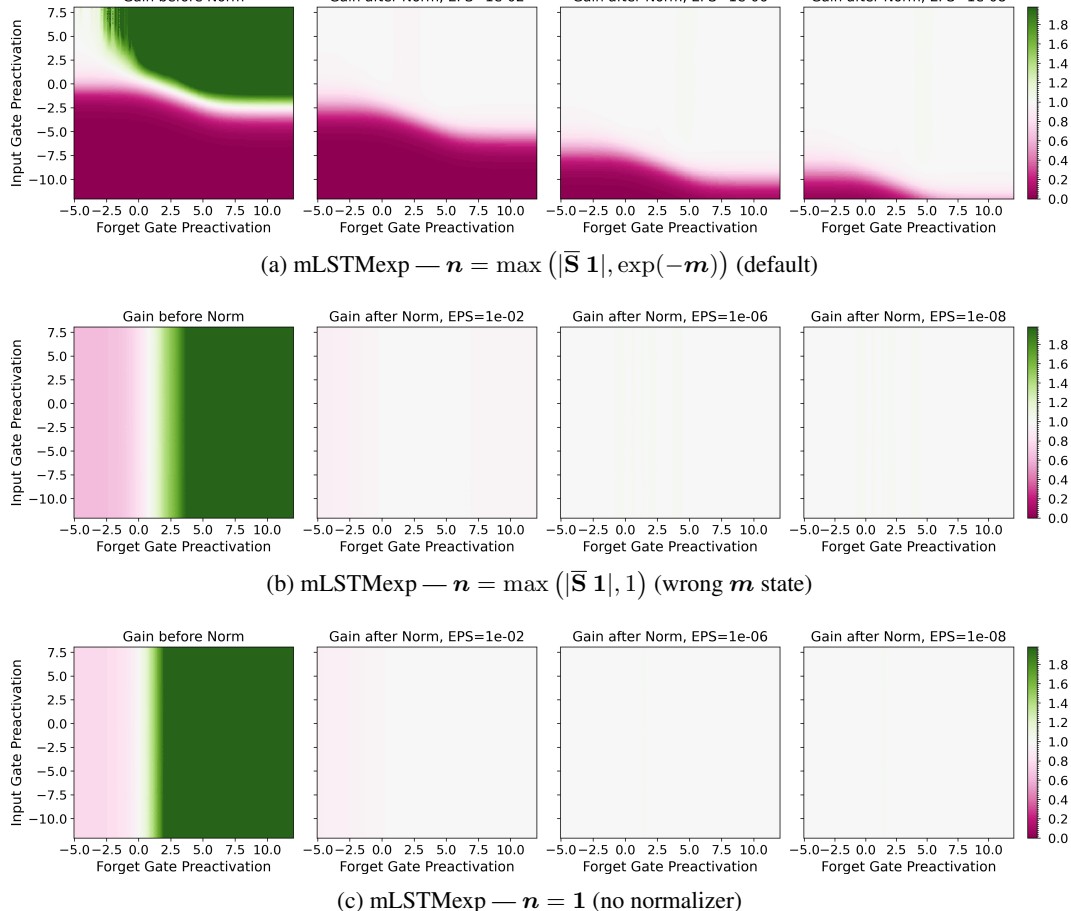

Figure 10: Transfer behavior of the **mLSTM with exponential input gate** for different normalization layer epsilons (EPS) and different normalizer variants. Only the default normalization shows the input gate dependent transfer behavior. Varying the normalization layer epsilon causes a shift of the gain curve in y-direction.

# E  Extended Experiments

In this section, we provide additional experiments and details to Section 5.

## E.1  Numerical Validation of TFLA Kernels

Before we begin our experiments on langauge modeling, we first verify that our kernels yield the same result as a reference implementation in pure JAX based on the fully parallel formulation (see Appendix B.1).

**Validation Perplexity Match (Table 3).**    We compare the validation perplexity at the end of training for 160M parameter mLSTMexp and mLSTMsig models trained on 19B tokens. We use context length 4096 since the parallel JAX implementation go out-of-memory for longer contexts. Model architecture and training recipe follows or general setup described in Appendix E.2.

In Table 3 we confirm that our kernels yield the same results as our reference implementation in JAX.

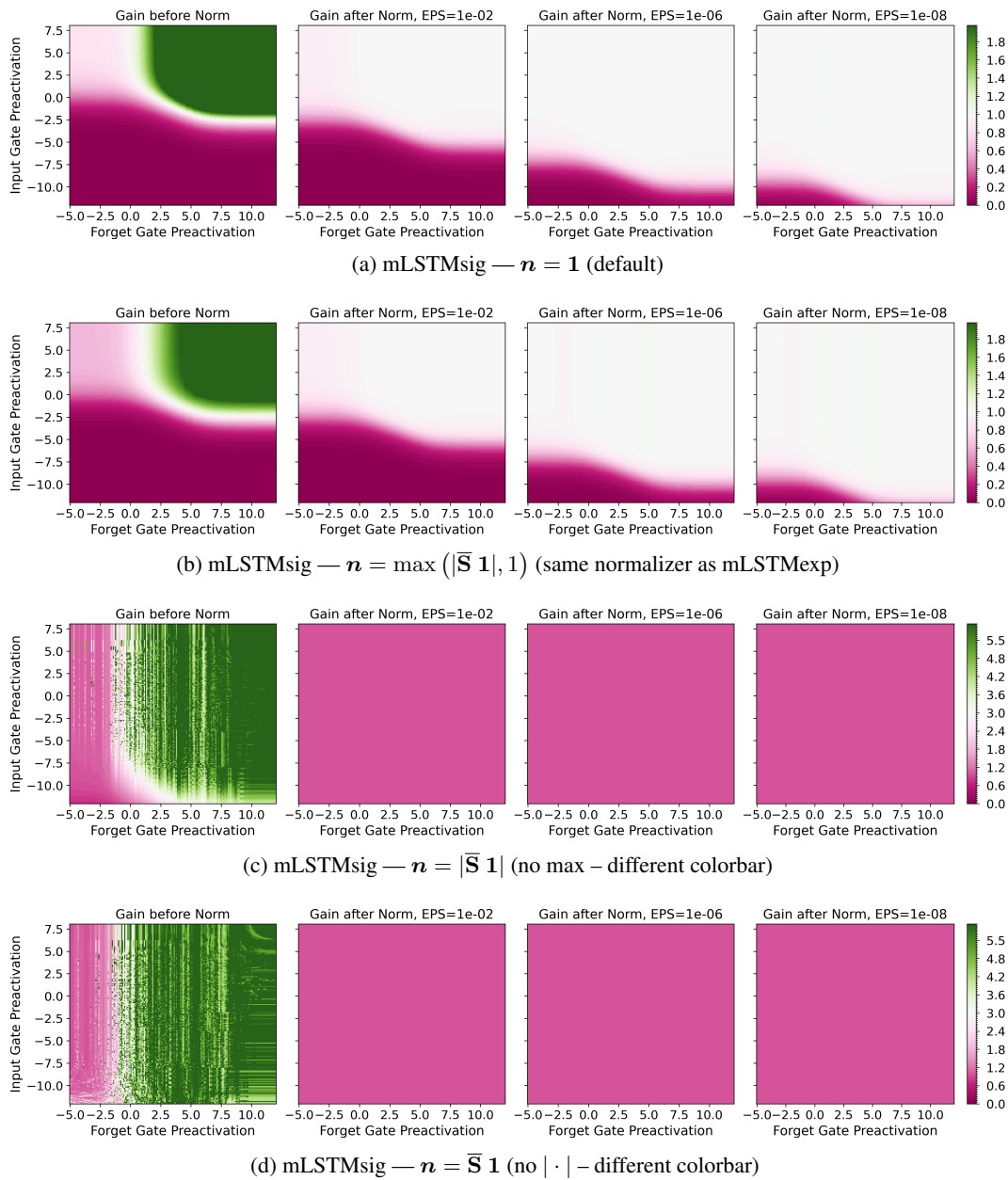

Figure 11: Transfer behavior of the **mLSTM with sigmoid input gate** for different normalization layer epsilons (EPS) and different normalizer variants. Removing the normalizer from mLSTMsig (which is our default setting in (a)) has no effect on the transfer behavior. If the normalizer is added, it should be bounded by 1 (see (b)). Varying the normalization layer epsilon causes a shift of the gain curve in y-direction.

Table 3: Validation Perplexity for 160M parameter models at context length 4096 trained on 19B tokens.

| Heads | EXP | | | SIG | |
| --- | --- | --- | --- | --- | --- |
| | JAX PARALLEL | LIMIT chunk | XL chunk | JAX PARALLEL | XL chunk |
| 6 | 21.02 | 21.03 | 21.18 | 21.01 | 21.05 |
| 12 | 21.01 | 21.03 | 21.07 | 21.02 | 21.06 |

### E.2 Extended Language Modeling Experiments with mLSTM

In this section we provide details on our experiment setup, model architecture and training recipe and add additional performance results on context length 8192 as well as analyze the effect of the epsilon parameter in the norm layer.

**Software and Hardware Setup.** We run our language modeling experiments in JAX 0.4.34 (Bradbury et al., 2018) and use FLAX 0.9.0 (Heek et al., 2024) to implement our models. We implement our kernels in Triton 3.1.0 (Tillet et al., 2019; Tillet, 2024) and use JAX-Triton 0.2.0 (Vikram et al., 2022) to integrate the kernels into JAX. Our kernel benchmark experiments are run in PyTorch 2.5.1 (Paszke et al., 2019), because most kernel baselines are available in PyTorch. All experiments are run on NVIDIA H100 80GB GPUs.

**Model Architecture.** The model architecture for mLSTMexp and mLSTMsig follows the design of most dense Transformer decoder only large language models (Radford et al., 2019; Brown et al., 2020; Touvron et al., 2023a,b).

An embedding layer, is followed by a stack of blocks and a language model head that produces the output logits (i.e. the values before softmax), which typically consists of a normalization layer and a linear (unembedding) layer. We apply logit soft-capping (Team, 2024), such that the value of the logits stay between $-c$ and $c$ for a specific cap value $c$. We choose $c = 30$. The logits are capped with the following function:

$$\mathrm{softcap}(\boldsymbol{x}) = c \cdot \tanh(\boldsymbol{x}/c) \tag{102}$$

We use the GPT-NeoX tokenizer (Black et al., 2022) with vocabulary size 50257 and do not tie the weights for the embedding layers and and the last (unembedding) layer.

Each block consists of two layers, where each layer has skip a connection and a normalization layer before the layer input (i.e. we use the pre-norm block architecture). As normalization layer we use the RMS-norm (Zhang & Sennrich, 2019) with epsilon $\epsilon = 1e\text{-}6$.

The first layer is a sequence-mix layer, that mixes the tokens along the sequence or time dimension. For standard Transformers this is the Attention operation (Vaswani et al., 2017). In our case, we replace Attention by the mLSTM operation with exponential or sigmoid input gate. Similar to Attention, mLSTM processes each token in multiple parallel heads. The second layer in the block is a feedforward linear layer that mixes the tokens per timestep channelwise. We use the SwiGLU feedforward linear layers (Shazeer, 2020; Touvron et al., 2023a).

For the mLSTM we set the head dimension for the queries and keys to be half of the values, i.e. $d_{qk} = 0.5\, d_{hv}$. We use Layernorm (Ba et al., 2016) as $\mathrm{NORM}(\boldsymbol{x})$ operation with epsilon $\epsilon = 1e\text{-}6$ in our experiments. [2] We apply soft-capping from equation (102) on the input and forget gate preactivations, as we found that this improves training stability. For the gate preactivations we set $c = 15$.

We provide the remaining model parameters in Table 4.

**Training Recipe.** We train our models with the AdamW optimizer (Loshchilov & Hutter, 2019) with $\beta_1 = 0.9$, $\beta_2 = 0.95$ and $\epsilon = 1e\text{-}8$. We use learning rates and batch sizes as specified in Table 4. We apply a weight decay of 0.1 to all linear layers (including the last linear layer or unembedding) and exclude biases and the token embeddings from weight decay. We clip the gradient norm at 0.5. We use a cosine learning rate scheduler with a linear warmup for the first 750 steps and decay to 0.1 of the peak learning rate, followed by a linear cooldown to 0 for the last 1000 steps. We list the number of training steps for every model size in Table 4. During pre-training we ensure that no information is leaked across document borders by resetting the memory states at the beginning of each new document. We implement this by manually setting the forget gate preactivations to a large negative values at the beginning of each new document.

**Additional Performance Results (Table 6).** In Table 6 we show the validation perplexity for mLSTMexp and mLSTM for context length 8192 (the results for context length 4096 are shown in

---

[2]We confirmed empirically that the type of normalization layer does not affect the performance as well as our qualitative results on transfer behavior and gradient norm variance. Therefore, we generally prefer RMS-norm as it faster.

Table 4: Training and Model Architecture Hyperparameters for our model sizes 160M, 400M and 1.4B.

| Model Size | Blocks | Embedding Dim | Heads | Head Dim | LR | Batch Size | Steps | Tokens 4k ctx | Tokens 8k ctx |
|---|---|---|---|---|---|---|---|---|---|
| 160M | 12 | 768 | 6
12 | 128
64 | 3e-3 | 128 | 36k | 19B | 38B |
| 400M | 24 | 1024 | 8
16 | 128
64 | 1e-3 | 128 | 46k | 24B | 48B |
| 1.4B | 24 | 2048 | 4
8
16 | 512
256
128 | 8e-4 | 256 | 31k | 33B | 65B |

Table 2). For some head dimension configurations we observed irrecoverable gradient norm spikes during training (indicated by -).

Table 5: Validation Perplexity at context length 8192. EXP and SIG denote mLSTMexp and mLSTMsig. LIMIT and XL correspond to `limit_chunk` and `xl_chunk` kernels. - indicates that the run experienced irrecoverable loss spikes during training.

| Size | Tokens | Heads | Llama | EXP LIMIT | EXP XL | SIG XL |
|---|---|---|---|---|---|---|
| 160M | 38B | 6
12 | 
19.99 | 20.29
20.31 | 20.43
20.42 | 20.46
20.52 |
| 400M | 48B | 8
16 | 
16.05 | 15.91
15.95 | 16.01
16.01 | 16.08
- |
| 1.4B | 65B | 4
8
16 | 

12.97 | 12.69
12.62
12.59 | 12.71
12.65
- | 12.91
12.67
12.75 |

Table 6: Training Step Time in seconds at context length 8192. EXP and SIG denote mLSTMexp and mLSTMsig. LIMIT and XL correspond to `limit_chunk` and `xl_chunk` kernels. All runs use the same training setup with global batch size of 256 sharded across 32 GPUs corresponding to a local batch size of 8.

| Size | Batch Size Global / Local | Heads | Llama | EXP LIMIT | EXP XL | SIG XL |
|---|---|---|---|---|---|---|
| 1.4B | 256 / 8 | 4
8
16 | 

2.39 | 1.77
1.67
1.60 | 1.73
1.66
1.61 | 1.67
1.64
1.62 |

**Effect of Trainable Input Gate (Table 7).** We investigate the effect of the input gate on the performance. Table 7 shows that having the input gate learnable consistently improves performance for both mLSTMexp and mLSTMsig.

**Effect of Input Gate Bias Initialization (Figure 12 and 13).** In our transfer behavior analysis in Section 4.2 we find that there is a transition from suppressing the signal to passing the signal at negative input gate values of around -8 (see Figure 4). Since we initialize the weights of the gates $w_{\{i,f\}}$ to 0, the biases of the input and forget gates determine the actual position in the x-y plane in the beginning of training. Initially, with input gate biases initialized to 0, we observe a high gradient norm variance, which was more pronounced for mLSTMsig (see Figure 12a and 13a).

Table 7: Validation Perplexity for 160M mLSTMs at context length 4096 with learnable and fixed input gate (bias initialized at -10).

| Input Gate | EXP LIMIT | SIG XL |
|---|---|---|
| Fixed | 21.23 | 21.24 |
| Learnable | 20.95 | 21.04 |

Therefore, we test to initialize the input gate biases at larger negative values. The forget gate biases are initialized equally spaced in the range [3,6]. As the weights $w_{\{i,f\}}$ grow during training, so do the gate preactivations and the model could learn to gradually move into the dynamical region of Figure 4, where the input signal is passed.

Indeed, as we observe in Figure 12 and 13 initializing the input gate biases to -10 effectively mitigates gradient norm spikes and reduces high gradient norm variance during training for both mLSTMexp and mLSTMsig. We therefore conclude that the additional input gate not only improves performance (see Table 7), but also improves training stability, if initialized correctly.

We use the `limit_chunk` kernel for mLSTMexp and our `xl_chunk` kernel for mLSTMsig and confirm that we obtain the same behavior with the `xl_chunk` kernel for mLSTMexp.

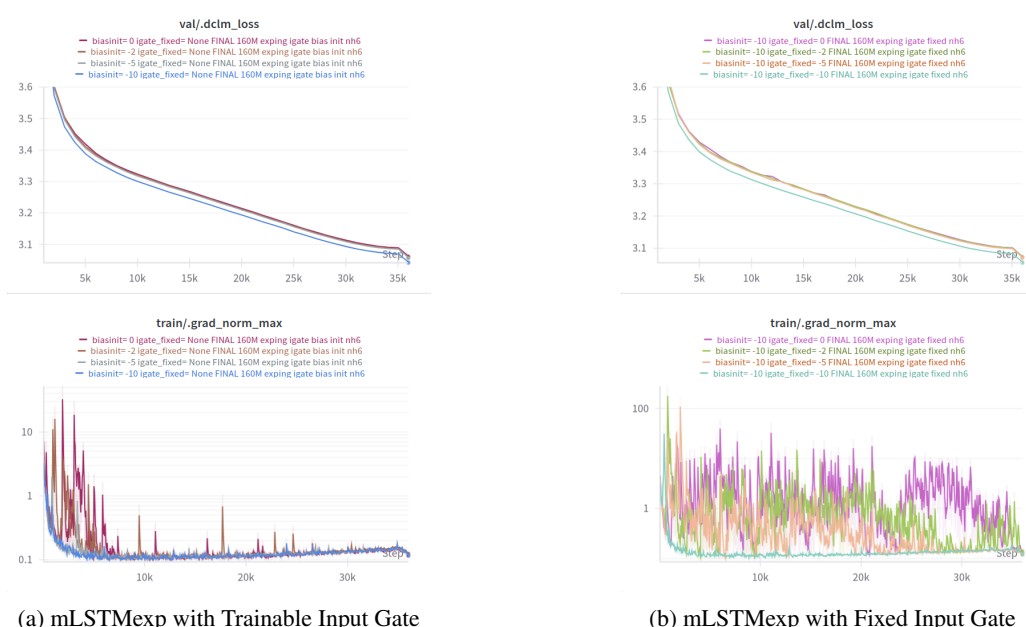

(a) mLSTMexp with Trainable Input Gate  (b) mLSTMexp with Fixed Input Gate

Figure 12: Trainable and fixed **exponential input gate** for bias initializations [0, -2, -5, -10] and norm epsilon $\epsilon$ =1e-6.

**Effect of Normalization Layer Epsilon on Performance (Figure 14).** In our empirical transfer behavior analysis of the mLSTM in Section 4.2 and D.2 we find that the transfer behavior depends on the input and forget gate preactivations, as well as the normalization layer epsilon (see Figure 10a and 11a). Therefore, we perform a grid search over different normalization layer epsilons and input gate bias initializations for the mLSTM with exponential input gate with 160M parameters and 6 heads at context length 4096. We show the results in Figure 14.

We observe that there is a diagonal region from norm layer epsilon and input gate bias $(\epsilon, b_i)$=(1e-6, -10) to (1e-4, -5) with improved performance. This indicate that if we increase the norm layer epsilon we can or should also increase the input gate bias initialization, as the shift of the gain curve in positive

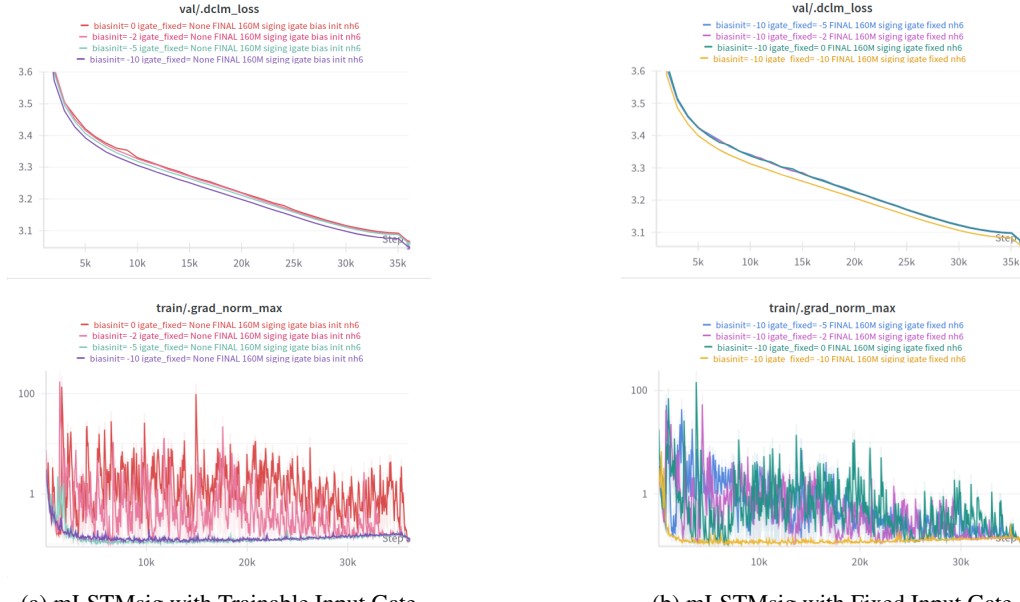

(a) mLSTMsig with Trainable Input Gate       (b) mLSTMsig with Fixed Input Gate

Figure 13: Trainable and fixed **sigmoid input gate** for bias initializations [0, -2, -5, -10] and norm epsilon $\epsilon =$1e-6.

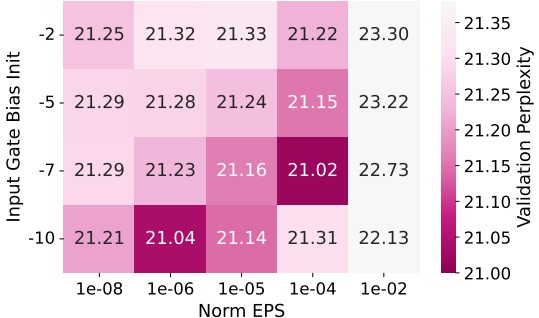

Figure 14: Validation Perplexity of mLSTMexp with 160M parameters with 6 heads. Grid search over norm layer epsilon and input gate bias initialization. The diagonal region of improved performance indicates, that there exists an interplay between the norm layer epsilon and input gate bias initialization. This supports the hypothesis that the norm layer is important for the gating mechanism.

y-direction for larger epsilons in Figure 10a suggests. This supports our hypothesis in Section 4.2, that the norm layer is important for the gating mechanism.

We use $(\epsilon, b_{\mathrm{i}})$=(1e-6, -10) as our default configuration.

**Input Gate Activations over Training (Figure 15).** We show the maximum input gate pre-activations (maximum over batch, sequence and head dimension) over training for mLSTMexp and mLSTMsig with 160M parameters in Figure 15. Both models have the input gate bias initialized to -10.

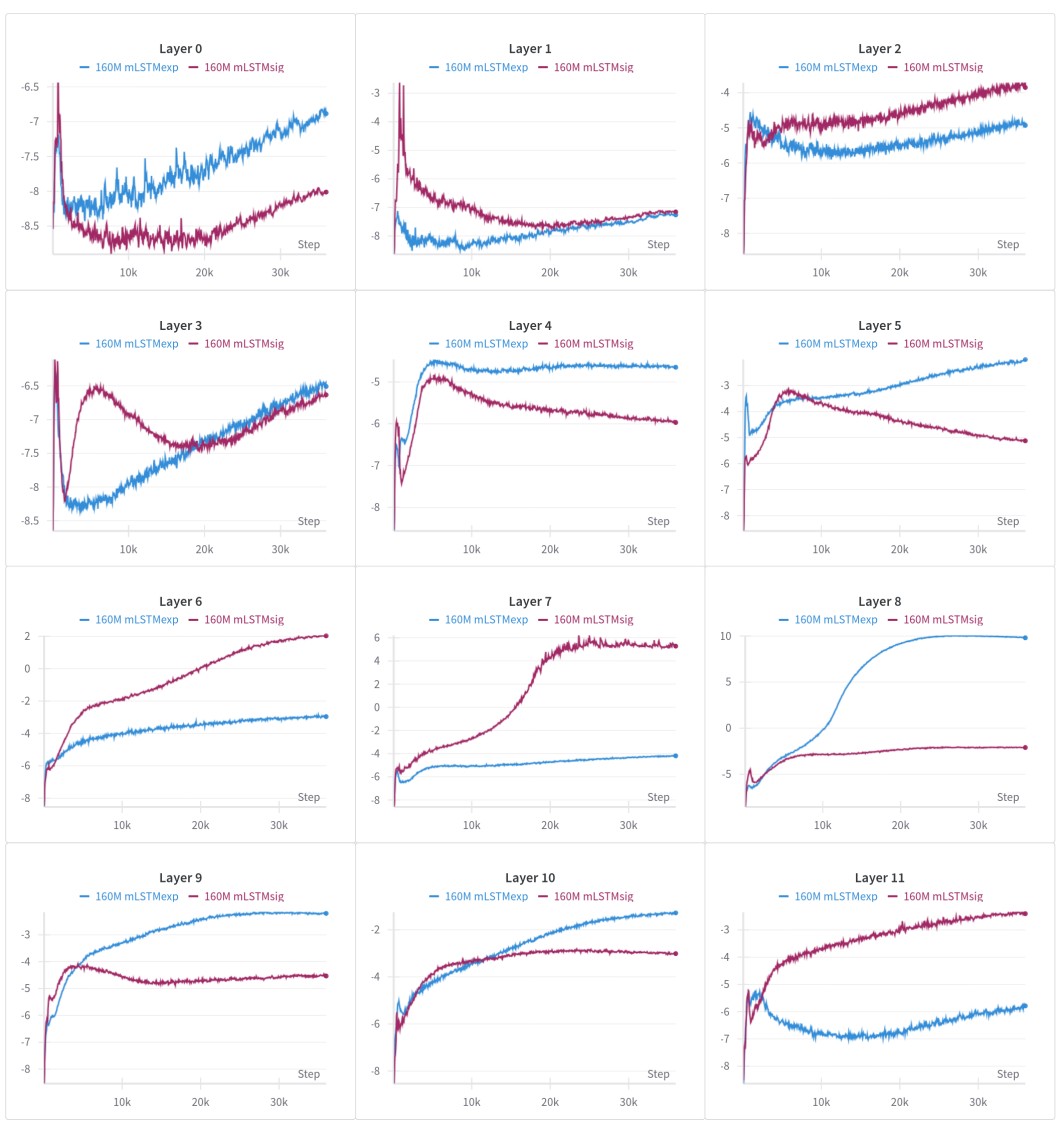

Figure 15: Maximum input gate pre-activation values $\tilde{i}_t$ over training for mLSTMexp and mLSTMsig with 160M parameters. Maximum taken over batch, sequence and head dimension. Both models have the input gate bias initialized to -10. In most cases the input gate pre-activations remain below zero.

### E.3 Extended Kernel Benchmark

In this section, we provide details on our benchmark setup and add additional benchmark results.

**Details on GPU Memory Measurement.** In Figure 6 and 16 we measure the GPU memory used by the kernels. For this, we use the PyTorch torch.cuda.max_memory_allocated API to measure the peak memory allocated during one kernel iteration. We make sure that the memory statistics are reset after each iteration and that the PyTorch caches are cleared before the start of each benchmark.

**Details on the Runtime Benchmark (Figure 5).** In our TFLA kernel runtime benchmark in Section 5.2, Figure 5 we report the median runtime of 30 iterations, after 10 warmup iterations in milliseconds. We run all kernels in bfloat16 precision.

We use the standard embedding dimension of 4096 for 7B Transformer models for our benchmark. Since different models and kernels have different default input sizes at this embedding dimension, we adapt the head dimension, number of heads and remaining input dimensions for each kernel accordingly. Following the practice of Shah et al. (2024) we keep the number of tokens constant at 65,536 and vary the sequence length (i.e. $T = [512, 1024, 2048, 4096, 8192, 16384, 32768, 65536]$) and batch size accordingly (i.e. $N_{\text{batch}} = 65536/T$).

We benchmark the following mLSTM kernels:

- **mLSTMexp (FLA limit_chunk):** Our own baseline kernel for the mLSTM with exponential input gate with limited chunk size based on FLA. Similar to FLA this kernel employs only single level sequence parallelism across chunks. We report the best performing chunk size of 64. The chunk size of 128 would still fit in SRAM, but is considerably slower.

- **mLSTMexp (TFLA xl_chunk):** TFLA kernel for the mLSTM with exponential input gate with two levels of sequence parallelism. We set the chunk size to the best performing chunk size of 128.

- **mLSTMsig (TFLA xl_chunk):** TFLA kernel for the mLSTM with sigmoid input gate. We set the chunk size to 128, but find chunk size 256 to perform equally well in terms of runtime (see Fig. 16 and 6).

For all our mLSTM kernels we use 16 heads, which results in head dimension $d_{hv} = 4096/16 = 256$ for the values. Similar to GLA (Yang et al., 2024b), we set the query and key head dimension to $d_{qk} = d_{hv}/2$, i.e. $d_{qk} = 128$.

We compare our mLSTM kernels with the following baselines:

- **Torch FlashAttention:** PyTorch 2.5.1 implementation of FlashAttention 2.
  Accessed via SDPBackend.FLASH_ATTENTION [3]

- **cuDNN FlashAttention:** NVIDIA cuDNN implementation of FlashAttention 2 integrated in PyTorch 2.5.1.
  Accessed via SDPBackend.CUDNN_ATTENTION.

- **FlashAttention 3:** FlashAttention 3 implementation[4], which has been optimized for NVIDIA H100 GPUs (Shah et al., 2024).

- **GLA (FLA):** Gated Linear Attention Triton kernel based on the Flash Linear Attention algorithm with one level of sequence parallelism (Yang et al., 2024b). Implementation from the official FLA repository, version 0.1[5]

- **Simple GLA (FLA):** A simple version of GLA with scalar forget gates per head. This primitive is not published as a new sequence modeling primitive but serves as a reference implementation for kernels for RetNet (Sun et al., 2023) or Mamba 2 (Dao & Gu, 2024) in the FLA library Yang & Zhang (2024). Moreover, Simple GLA is similar to mLSTMsig, but has no input gate. Therefore, we find it interesting to add it as baseline. We use the implementation from the official FLA repository, version 0.1.

---

[3]See torch.nn.attention.SDPBackend
[4]See https://github.com/Dao-AILab/flash-attention
[5]See https://github.com/fla-org/flash-linear-attention

- **Mamba:** Mamba CUDA kernel Gu & Dao (2024). Implementation from the official Mamba repository, version 2.2.4.
- **Mamba 2:** Mamba 2 Triton kernels Dao & Gu (2024). Implementation from the official Mamba repository, version 2.2.4.[6]

For all FlashAttention baselines we use 32 heads with head dimension 128 for queries, keys and values. For the Flash Linear Attention (FLA) kernels GLA and Simple GLA, we use the identical head configuration as for our TFLA mLSTM kernels (i.e. 16 heads, $d_{hv} = 256$, $d_{qk} = 128$). For Mamba, we use our embedding dimension of 4096 and set the state dimension to 16 similar to Gu & Dao (2024). For Mamba 2, we use their default head dimension of 64 and set the number of heads to $4096/64 = 64$. Note that smaller head dimension can yield faster runtimes (see Figure 18).

We show the results of this benchmark for varying sequence length and constant number of tokens in Figure 5. When comparing the forward pass runtime only, we find that Mamba2 and Simple GLA kernels are slightly faster than our mLSTMsig kernels. However, this difference is within 1 ms. In training, when forward and backward pass runtime is measured, our TFLA kernels are faster than FlashAttention 3 for longer sequence lengths and more than two times faster than Mamba 2 kernels for all sequence lengths. Only Simple GLA (FLA) can keep up in training speed with our TFLA mLSTM kernels. Therefore, we compare the runtime and memory usage for a larger head dimension in Figure 16 and find that this comes at the cost of almost 2 times the GPU memory usage compared to our TFLA mLSTM kernels. These memory savings are achieved by leveraging a larger chunk size, enabled through the two levels of sequence parallelism outlined in Section 3.

**Runtime and Memory Comparison with FLA Kernels (Figure 16).** In this experiment we compare the runtime and memory consumption of our TFLA mLSTM kernels with prominent kernels from the Flash Linear Attention library. We use a similar setup to our previous benchmark, but perform this comparison with 8 heads at a larger head dimension of 512 for the values and 256 for the queries and keys, since both Beck et al. (2024) and Yang et al. (2024b) report better language modeling performance for larger head dimensions.

In addition to GLA (chunk) and Simple GLA (chunk), we also compare with GLA (fused) which is the non-materialization version of Gated Linear Attention (GLA) (Yang et al., 2024b).

The non-materialization version of GLA has been also proposed by Qin et al. (2024a) as Lightning Attention-2 (see also Section A). For the forward pass it fuses the inter- and intra-chunk part of the chunkwise-parallel Linear Attention formulation (see Section 2.2) and therefore does not materialize the hidden states in GPU memory.

Interestingly, in our experiments we find that even though the non-materialization version uses the least GPU memory of all FLA kernels, it is neither faster nor more memory efficient in training than our TFLA mLSTM kernels (see Figure 16). While Simple GLA is slightly faster (within 3 ms or 15%), it uses almost twice the GPU memory compared to our TFLA mLSTM kernels. The speed of Simple GLA can be partly explained to the fact that it computes less FLOPs (no input-gate) and the fact that in the forward pass the memory cell states are materialized in `bfloat16`, while TFLA materializes states in `float32` for improved stability, which causes twice the memory IO per state.

**Runtime and Memory Comparison with LightningAttention2 Kernels (Figure 17).** Similar to the previous experiment, we compare the runtime and memory consumption of our TFLA mLSTM kernels with LightningAttention2 (Qin et al., 2024a). LightningAttention2 is the core of the recent hybrid large language model MiniMax-01, which combines lightning attention (a linear attention variant with data independent decay) with softmax attention (MiniMax et al., 2025). MiniMax-01 is proposed as a very efficient long-context language model, which makes the comparison between LightningAttention2 and our TFLA mLSTM kernels interesting.

LightningAttention2 also uses the chunkwise-parallel formulation for linear RNNs (see Section 2.2). However, in contrast to Simple GLA and TFLA it does not split the computation in a recurrent and parallel part, but instead processes all chunks fully recurrent (see Section A for more details).

We find that LightningAttention2 supports only identical head dimensions for queries, keys and values up to 128. For this reason, we discuss this comparison separately from the other experiments.

---

[6]See https://github.com/state-spaces/mamba

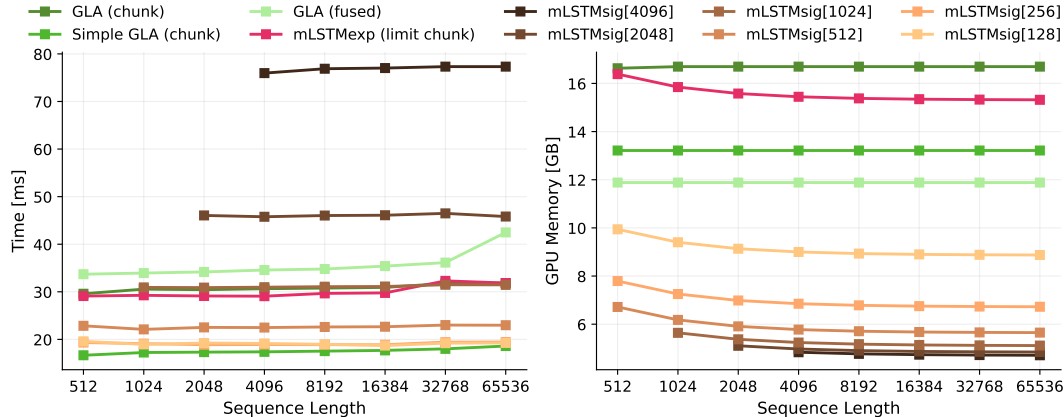

Figure 16: Runtime and Memory Comparison with FLA Kernels. **Left:** Runtime (Forward Backward Pass). **Right:** GPU Memory Usage.
We use 8 heads and head dimension of 512 for values, and 256 for queries and keys. Simple GLA (the fastest FLA kernel in our experiments) is slightly faster than our TFLA mLSTMsig kernels but uses almost twice as much GPU memory.

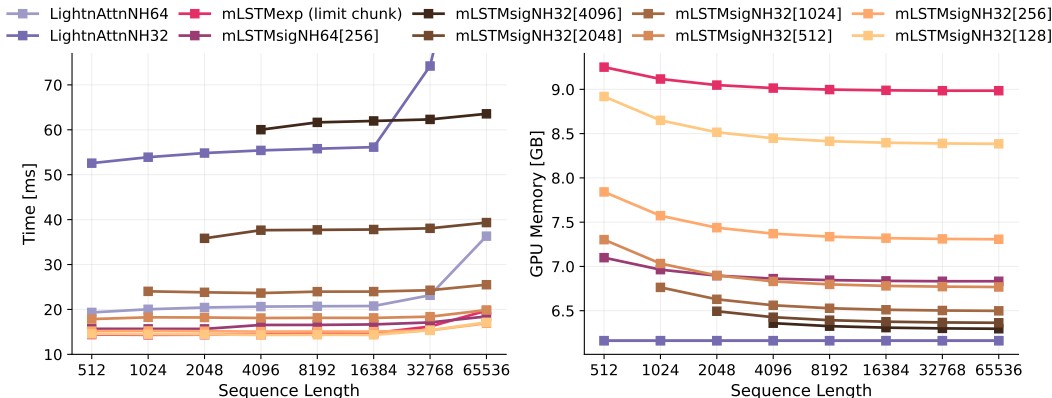

Figure 17: Runtime and Memory Comparison with LightningAttention2. **Left:** Runtime (Forward Backward). **Right:** GPU Memory.
We use 32 and 64 heads with head dimension 128 and 64 for queries, keys and values. LightningAttention has the least memory usage of all kernels, but is more than 3 times slower than our TFLA mLSTM at the larger head dimension of 128.

We compare our TFLA mLSTM kernels with LightningAttenion2 for 32 and 64 heads, corresponding to head dimension 128 and 64. We keep the number of tokens fixed to 65536 and vary sequence length and batch size in the same way as above.

We show the results in Figure 17. Since LightningAttention does not materialize intermediate states, it has the least GPU memory usage with 6.2 GB. However, this GPU memory efficiency comes at the cost of a more than 3 times longer runtime compared to our TFLA mLSTMsig kernel with chunk size 256, which uses about 7.3 GB of GPU memory. This highlights that there exists a trade-off between GPU memory usage and runtime for linear RNN kernels based on the chunkwise-parallel formulation. Our experiments demonstrate that our TFLA kernel algorithm provides an effective method to balance this trade-off via the chunk size parameter (see Figure 6).

**Runtime Benchmark for Varying Head Dimensions (Figure 18).** It has been reported in several other works that larger head dimensions (compared to common Self-Attention head dimensions) lead to improved language modeling performance for linear RNNs (Sun et al., 2023; Beck et al., 2024; Yang et al., 2024b). Consequently, it is desirable for linear RNN kernels to be fast and efficient across

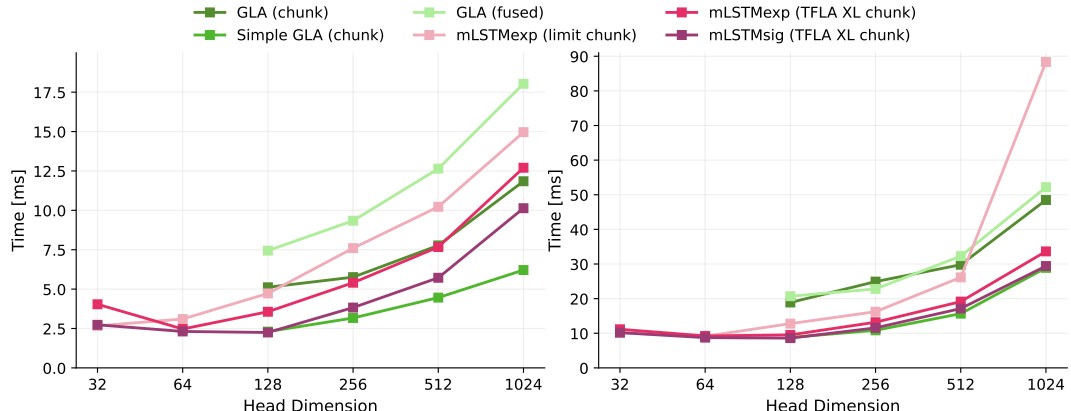

Figure 18: Head Dimension Benchmark for FLA and TFLA mLSTM kernels. **Left:** Forward Pass. **Right:** Forward and Backward Pass.
We measure the runtime for sequence length 8192 and batch size 4 for different head dimensions. We use the same head dimension for queries, keys and values. Our TFLA mLSTM kernels show fast runtimes even for very large head dimensions.

a wide range of head dimensions. In this experiment, we evaluate whether our new TFLA kernels exhibit this property.

We vary the head dimension from 32 to 1024 and adapt the number of heads for a total embedding dimension of 4096 and measure the runtime for inputs of sequence length 8192 and batch size 4. We use the same head dimension for queries, keys and values.

For the FLA kernels the head dimensions 32 and 64 did not run, due to Triton compiler errors. As the FLA library is still being developed at the time of writing this paper, we expect this to be fixed soon.

We observe that for small head dimensions (i.e. 32 and 64) our mLSTM limit chunk kernel is as fast as our TFLA mLSTM kernels in training.

In summary, our results in Figure 18 confirm that our TFLA kernels achieve fast runtimes across a wide range of head dimensions.

# F FLOP and Memory Operation Counts for the mLSTM

We count the number of floating point operations (FLOPs) and the memory operations (load and stores in bytes) in a forward pass (with batch size 1) of the mLSTM with exponential and sigmoid input gate. We use a factor of 2 to describe the multiply accumulate cost for FLOPs.

We do not count FLOPs that belong to recomputation, that happens within kernels. For example, when we parallelize across the embedding dimension in the forward kernel $\mathbf{H}^{(k)}$, each of the $d_{hv}/B_{dhv}$ blocks recomputes the matrix $\mathbf{S}$. Similarly, we do not count the additional memory-loading operations that are necessary for the recomputations. During training, we typically have fixed context lengths. Therefore, we do not count loading the initial state and storing the final state.

We use factors denoted as $F_{OP}$ to describe the number of FLOPs for operation OP (e.g. $F_{exp}$ for the exponential function). By default, we set all of these factors to 1. We do not neglect these factors, as the impact depends on the selected chunk size in some terms, which can be chosen freely in TFLA. Moreover, we might want to do an even more fine-grained FLOP analysis, where we account for the differences in compute cost of some operations.

We use the factors $bytes_X$ to denote the size of each element in the tensor (e.g. $bytes_{qkv}$ for the query, key and value tensors). Typically during training the queries, keys and values are stored in `bfloat16` (i.e. $bytes_{qkv} = 2$), while the memory cell states are kept in `float32` (i.e. $bytes_{Cnm} = 4$). We summarize the notation used in this section in Table 8.

In the remainder of this section, we count the FLOP and memory operation counts for the chunkwise-parallel, fully-parallel and recurrent mLSTM formulations (Section F.1 and F.2), analyze the difference in FLOPs counts between mLSTMexp and mLSTMsig (Section F.3) as well as between the different formulations (Section F.4), and finally compute the FLOP-optimal chunk size for the chunkwise-parallel formulation (Section F.5).

Table 8: **Notation** for FLOP and Memory Operation Counts.

| Symbol | Description |
|---|---|
| $N_{batch}$ | Batch size |
| $N_{head}$ | Number of heads |
| $N_{chunk}$ | Number of chunks |
| $T$ | Sequence length |
| $L$ | Chunk size |
| $d_{hv}$ | Head dimension for values and hidden states |
| $d_{qk}$ | Head dimension for queries and keys |
| $F_{OP}$ | FLOPs for the operation OP (e.g. $\exp$) |
| $F_{causal}$ | Factor that accounts for causality, typically 0.5 |
| $bytes_X$ | Number of bytes used for each element in tensor X |

## F.1 Exact FLOPs and Memory Operations Count for the mLSTM

**Chunkwise-Parallel Formulation (Table 9, 10).** We count the FLOPs (Table 9) and memory operations (Table 10) of the chunkwise-parallel mLSTM formulation (see Section 2.2 and Appendix B.2). All counts are for a single head and a single chunk. To obtain the total counts for a full sequence, we multiply these counts by the number of heads $N_{head}$ and chunks $N_{chunk} = T/L$.

**Fully Parallel Formulation (Table 11, 12).** We count the FLOPs (Table 11) and memory operations (Table 12) of the fully parallel mLSTM formulation (see Appendix B.1). All counts are for a single head and a full sequence of length $T$. To obtain the total counts, we multiply by the number of heads $N_{head}$.

**Recurrent Formulation (Table 13, 14).** We count the FLOPs (Table 13) and memory operations (Table 14) of the recurrent mLSTM formulation (see Section 2.1 and 4.1). For the memory operations we assume that the states are materialized after every timestep, which is the setting during text generation. All counts are for a single head and a single time step. To obtain the total counts for a full sequence, we multiply by the sequence length $T$ and the number of heads $N_{head}$.

Table 9: **FLOP counts** for the **chunkwise-parallel mLSTM formulation** for mLSTMexp and mLSTMsig. All terms denote the FLOP count per head and chunk.

| FLOPs | mLSTMexp | mLSTMsig |
|---|---|---|
| *Recurrent computation of the inter chunk states* | | |
| **Gates:** | $2L + \frac{1}{2}L(L+1)$ $+L(1 + F_{\exp} + F_{\log} + F_{\text{sig}}) + 3 + F_{\max} + F_{\exp}$ | $2L + \frac{1}{2}L(L+1) + LF_{\exp} + F_{\exp}$ $+2L(F_{\log} + F_{\text{sig}})$ |
| **Numerator:** | $2d_{qk}d_{hv} + 2Ld_{qk}d_{hv} + Ld_{qk}$ | $2d_{qk}d_{hv} + 2Ld_{qk}d_{hv} + Ld_{qk}$ |
| **Denominator:** | $2d_{qk} + 2Ld_{qk}$ | — |
| *Parallel computation of the intra chunk outputs* | | |
| **Cumulative Forget Gates:** | $\frac{1}{2}L(L+1) + L(F_{\log} + F_{\text{sig}})$ | $\frac{1}{2}L(L+1) + 2L(F_{\log} + F_{\text{sig}})$ |
| **Gate Matrix:** | $F_{\text{causal}} \times \left(L^2(3 + F_{\exp} + F_{\max}) + L(1 + F_{\max})\right)$ | $F_{\text{causal}} \times \left(L^2(2 + F_{\exp})\right)$ |
| **Intra Outputs:** | $F_{\text{causal}} \times \left(2L^2(d_{qk} + d_{hv}) + 3L^2\right)$ | $F_{\text{causal}} \times \left(2L^2(d_{qk} + d_{hv}) + 3L^2\right)$ |
| *Parallel computation of the inter chunk outputs* | | |
| **Inter Outputs:** | $2Ld_{qk}d_{hv} + 3Ld_{qk}$ | $2Ld_{qk}d_{hv} + Ld_{qk}$ |
| *Combination of inter and intra chunk outputs* | | |
| **Output Combination:** | $2Ld_{hv} + L(1 + F_{\max} + F_{\text{abs}} + F_{\exp})$ | $Ld_{hv}$ |

Table 10: **Memory operation counts** for the **chunkwise-parallel mLSTM formulation** for mLSTMexp and mLSTMsig. All terms denote the memory operation count per head and chunk.

| Bytes | mLSTMexp | mLSTMsig |
|---|---|---|
| *Inter-chunk Recurrent Kernel* | | |
| **Load:** | $L(d_{qk} + d_{hv}) \times \text{bytes}_{qkv} + 2L \times \text{bytes}_{if}$ | $L(d_{qk} + d_{hv}) \times \text{bytes}_{qkv} + 2L \times \text{bytes}_{if}$ |
| **Store:** | $(d_{qk}d_{hv} + d_{qk} + 1) \times \text{bytes}_{Cnm}$ | $d_{qk}d_{hv} \times \text{bytes}_{Cnm}$ |
| *Intra-chunk Parallel Kernel* | | |
| **Load:** | $L(2d_{qk} + d_{hv}) \times \text{bytes}_{qkv} + 2L \times \text{bytes}_{if}$ $+(d_{qk}d_{hv} + d_{qk} + 1) \times \text{bytes}_{Cnm}$ | $L(2d_{qk} + d_{hv}) \times \text{bytes}_{qkv} + 2L \times \text{bytes}_{if}$ $+d_{qk}d_{hv} \times \text{bytes}_{Cnm}$ |
| **Store:** | $Ld_{hv} \times \text{bytes}_{qkv} + 2L \times \text{bytes}_{Cnm}$ | $Ld_{hv} \times \text{bytes}_{qkv}$ |
| **Total:** | $4L \times \text{bytes}_{if}$ $+3L(d_{hv} + d_{qk}) \times \text{bytes}_{qkv}$ $+2(L + d_{hv}d_{qk} + d_{qk} + 1) \times \text{bytes}_{Cmn}$ | $4L \times \text{bytes}_{if}$ $+3L(d_{hv} + d_{qk}) \times \text{bytes}_{qkv}$ $+2d_{hv}d_{qk} \times \text{bytes}_{Cmn}$ |

Table 11: **FLOP counts** for the **fully parallel mLSTM formulation** for mLSTMexp and mLSTMsig. All terms denote the FLOP count for a full sequence per head.

| FLOPs | mLSTMexp | mLSTMsig |
|---|---|---|
| **Cumulative Forget Gates:** | $\frac{1}{2}T(T+1) + T(F_{\log} + F_{\text{sig}})$ | $\frac{1}{2}T(T+1) + 2T(F_{\log} + F_{\text{sig}})$ |
| **Gate Matrix:** | $T^2(3 + F_{\exp} + F_{\max} + F_{\text{mask}})$ | $T^2(3 + F_{\exp} + F_{\max} + F_{\text{mask}})$ |
| **Attention Logits:** | $F_{\text{causal}} \times \left(2T^2d_{qk} + 2T^2\right)$ | $F_{\text{causal}} \times \left(2T^2d_{qk} + 2T^2\right)$ |
| **Normalization:** | $F_{\text{causal}} \times \left(T^2(3 + F_{\text{abs}}) + T(F_{\exp} + F_{\max})\right)$ | – |
| **Outputs:** | $F_{\text{causal}} \times 2T^2d_{hv}$ | $F_{\text{causal}} \times 2T^2d_{hv}$ |

Table 12: **Memory operation counts** for the **fully parallel mLSTM formulation** for mLSTMexp and mLSTMsig. All terms denote the memory operation count for a full sequence per head.

| Bytes | mLSTMexp | mLSTMsig |
|---|---|---|
| **Load:** | $T(2d_{qk} + d_{hv}) \times \text{bytes}_{qkv} + 2T \times \text{bytes}_{if}$ | $T(2d_{qk} + d_{hv}) \times \text{bytes}_{qkv} + 2T \times \text{bytes}_{if}$ |
| **Store:** | $Td_{hv} \times \text{bytes}_{qkv} + 2T \times \text{bytes}_{Cmn}$ | $Td_{hv} \times \text{bytes}_{qkv}$ |
| **Total:** | $2T(\text{bytes}_{if} + (d_{hv} + d_{qk}) \times \text{bytes}_{qkv} + \text{bytes}_{Cmn})$ | $2T(\text{bytes}_{if} + (d_{hv} + d_{qk}) \times \text{bytes}_{qkv})$ |

Table 13: **FLOP counts** for the **recurrent mLSTM formulation** for mLSTMexp and mLSTMsig. All terms denote the FLOP count for a single timestep per head.

| FLOPs | mLSTMexp | mLSTMsig |
|---|---|---|
| **Gates:** | $4 + 2F_{\exp} + F_{\log} + F_{\text{sig}} + F_{\max}$ | $2F_{\text{sig}}$ |
| **Memory Cell Update:** | $4d_{qk}d_{hv}$ | $4d_{qk}d_{hv}$ |
| **Denominator & Scale:** | $6d_{qk} + d_{hv} + 1 + F_{\text{abs}} + F_{\max}$ | – |
| **Output:** | $2d_{hv}d_{qk} + d_{qk}$ | $2d_{hv}d_{qk} + d_{qk}$ |

Table 14: **Memory operation counts** for the **recurrent mLSTM formulation** for mLSTMexp and mLSTMsig. All terms denote the memory operation count for a single timestep per head. We assume the states are materialized at every timestep.

| Bytes | mLSTMexp | mLSTMsig |
|---|---|---|
| **Load:** | $(2d_{qk} + d_{hv}) \times \text{bytes}_{qkv} + 2 \times \text{bytes}_{if}$ $+ (d_{qk}d_{hv} + d_{qk} + 1) \times \text{bytes}_{Cmn}$ | $(2d_{qk} + d_{hv}) \times \text{bytes}_{qkv} + 2 \times \text{bytes}_{if}$ $+ d_{qk}d_{hv} \times \text{bytes}_{Cmn}$ |
| **Store:** | $d_{hv} \times \text{bytes}_{qkv} + (d_{qk}d_{hv} + d_{qk} + 1) \times \text{bytes}_{Cmn}$ | $d_{hv} \times \text{bytes}_{qkv} + d_{qk}d_{hv} \times \text{bytes}_{Cmn}$ |
| **Total:** | $2 \times \text{bytes}_{if} + 2(d_{hv} + d_{qk}) \times \text{bytes}_{qkv}$ $+ 2d_{hv}d_{qk} \times \text{bytes}_{Cmn}$ | $2 \times \text{bytes}_{if} + 2(d_{hv} + d_{qk}) \times \text{bytes}_{qkv}$ $+ 2(d_{hv}d_{qk} + d_{qk} + 1) \times \text{bytes}_{Cmn}$ |

## F.2 Simplified FLOP Count Summary for the mLSTM

In this section we simplify the FLOP count for the mLSTM by setting all factors $F_{\text{OP}}$ to 1. We leave the causal factor $F_{\text{causal}}$ unspecified, but typically set it to 0.5 or slightly larger. Since the attention logit matrix (i.e. the quadratic matrix $\mathbf{S}$) is always computed in blocks due to the blockwise nature of tensor core operation, usually some parts of the upper triangular matrix are computed and then masked out. To account for this, the factor $F_{\text{causal}}$ can be set to a value larger than 0.5 (e.g. 0.66). In Figure 19 we show the impact of the causal factor on the overall FLOP count for the mLSTMsig.

Tables 15, 16 and 17 summarize the simplified FLOP counts for the chunkwise-parallel, fully parallel and recurrent mLSTM formulation.

**Total Flop Count Summary (Table 18).** In Table 18 we summarize the total FLOP counts for all formulations of the mLSTM with exponential and sigmoid input gate for a single head and a full sequence of length $T$ (i.e. batch size $N_{\text{batch}} = 1$). To obtain the total FLOP counts for one sequence, we multiply the chunkwise-parallel FLOP counts per chunk by the number of chunks $N_{\text{chunk}} = T/L$ and the recurrent FLOPs per step by the sequence length $T$.

Table 15: **Simplified FLOP counts** for the **chunkwise-parallel mLSTM formulation** for mLSTM-exp and mLSTMsig. All terms denote the FLOP count per head and chunk. We set all factors $F_{\text{OP}}$ to 1.

| FLOPs | mLSTMexp | mLSTMsig |
|---|---|---|
| *Recurrent computation of the inter chunk states* | | |
| **Gates:** | $0.5L^2 + 6.5L + 5$ | $0.5L^2 + 7.5L + 1$ |
| **Numerator:** | $2d_{qk}d_{hv} + 2Ld_{qk}d_{hv} + Ld_{qk}$ | $2d_{qk}d_{hv} + 2Ld_{qk}d_{hv} + Ld_{qk}$ |
| **Denominator:** | $2d_{qk} + 2Ld_{qk}$ | — |
| *Parallel computation of the intra chunk outputs* | | |
| **Cumulative Forget Gates:** | $0.5L^2 + 2.5L$ | $0.5L^2 + 2.5L$ |
| **Gate Matrix:** | $F_{\text{causal}} \times \left(5L^2 + 2L\right)$ | $F_{\text{causal}} \times 3L^2$ |
| **Intra Outputs:** | $F_{\text{causal}} \times \left(2L^2(d_{qk} + d_{hv}) + 3L^2\right)$ | $F_{\text{causal}} \times \left(2L^2(d_{qk} + d_{hv}) + 3L^2\right)$ |
| *Parallel computation of the inter chunk outputs* | | |
| **Inter Outputs:** | $2Ld_{qk}d_{hv} + 3Ld_{qk}$ | $2Ld_{qk}d_{hv} + Ld_{qk}$ |
| *Combination of inter and intra chunk outputs* | | |
| **Output Combination:** | $2Ld_{hv} + 4L$ | $Ld_{hv}$ |

## F.3 FLOP Comparison between mLSTMexp and mLSTMsig

The mLSTM with sigmoid input gate does not have a normalizer and a max state. Therefore, it has fewer FLOPs and memory operations compared to mLSTM with exponential input gate. In this section we quantify this difference in FLOP counts between mLSTMexp and mLSTMsig.

**FLOP Count Difference between mLSTMexp and mLSTMsig (Table 18).** We compute the FLOP difference of mLSTMexp and mLSTMsig in the last column of Table 18. We observe that in the leading terms, there is no difference and conclude that the FLOP difference between mLSTMexp and mLSTMsig is small. For example for head dimension $d_{qk} = d_{hv} = 64$, we find that mLSTMexp

Table 16: **Simplified FLOP counts** for the **fully parallel mLSTM formulation** for mLSTMexp and mLSTMsig. All terms denote the FLOP count for a full sequence per head. We set all factors $F_{\text{OP}}$ to 1.

| FLOPs | mLSTMexp | mLSTMsig |
|---|---|---|
| **Cumulative Forget Gates:** | $0.5T^2 + 2.5T$ | $0.5T^2 + 4.5T$ |
| **Gate Matrix:** | $6T^2$ | $6T^2$ |
| **Attention Logits:** | $F_{\text{causal}} \times \left(2T^2 d_{qk} + 2T^2\right)$ | $F_{\text{causal}} \times \left(2T^2 d_{qk} + 2T^2\right)$ |
| **Normalization:** | $F_{\text{causal}} \times \left(4T^2 + 2T\right)$ | $-$ |
| **Outputs:** | $F_{\text{causal}} \times 2T^2 d_{hv}$ | $F_{\text{causal}} \times 2T^2 d_{hv}$ |

Table 17: **Simplified FLOP counts** for the **recurrent mLSTM formulation** for mLSTMexp and mLSTMsig. All terms denote the FLOP count for a single timestep per head. We set all factors $F_{\text{OP}}$ to 1.

| FLOPs | mLSTMexp | mLSTMsig |
|---|---|---|
| **Gates:** | $9$ | $2$ |
| **Memory Cell Update:** | $4d_{qk}d_{hv}$ | $4d_{qk}d_{hv}$ |
| **Denominator & Scale:** | $6d_{qk} + d_{hv} + 3$ | $-$ |
| **Output:** | $2d_{hv}d_{qk} + d_{qk}$ | $2d_{hv}d_{qk} + d_{qk}$ |

has less than 2% more FLOPS, while for $d_{qk} = d_{hv} = 512$ the mLSTMexp has only about 0.2% more FLOPs for all formulations.

**Is mLSTMsig faster because it has fewer FLOPs?** We find that the mLSTMexp has only slightly more FLOPs than mLSTMsig. Therefore, the speed difference between mLSTMexp and mLSTMsig cannot be explained by the FLOP count difference alone. However, even though the absolute and relative FLOP count difference is small, the FLOPs that differ are the more "expensive" FLOPs, i.e. pointwise operations and vector operations, which are more expensive than matrix multiplications as they are not performed on tensor cores. So to answer the question, even though the FLOP difference seems negligible, it is expected that the actural relative runtime difference is larger than the relative FLOP difference, since the FLOPs that differ are "slower non tensor core" FLOPs. For the forward pass this indicates that the main reason for the 30% speedup of mLSTMsig over mLSTMexp is the efficient fusion of loops in the mLSTMsig kernel (see Appendix C.3).

## F.4 FLOP Comparison between different mLSTM Formulations

The main advantage of TFLA over Flash Linear Attention is the freely configurable chunksize, wich allows to effectively trade off between memory consumption and runtime (see Figure 6) and as we will see in Figure 19 also between the total number of FLOPs.

We compare the FLOP counts of the chunkwise-parallel formulation for different chunk sizes with the fully parallel and the recurrent formulation. We use the simplified FLOP counts for the mLSTMsig from Table 18 for this analysis.

Table 18: **Total Simplified FLOP counts** for the chunkwise-parallel, fully parallel and recurrent formulation of mLSTMexp and mLSTMsig. All terms denote the FLOP count per head for a full sequence of length $T$. We set all factors $F_{\text{OP}}$ to 1.

| FLOPs | mLSTMexp | mLSTMsig | Difference |
|---|---|---|---|
| **chunkwise-parallel:** | $TLF_{\text{causal}}\left(2(d_{qk} + d_{hv}) + 8\right) + TL + 2TF_{\text{causal}}$ $+T\left(4d_{qk}d_{hv} + 6d_{qk} + 4d_{hv} + 13\right)$ $+\frac{T}{L}\left(2d_{qk}d_{hv} + 2d_{qk} + 5\right)$ | $TLF_{\text{causal}}\left(2(d_{qk} + d_{hv}) + 6\right) + TL$ $+T\left(4d_{qk}d_{hv} + 2d_{qk} + d_{hv} + 11\right)$ $+\frac{T}{L}\left(2d_{qk}d_{hv} + 2d_{qk} + 5\right)$ | $2TLF_{\text{causal}} + 2TF_{\text{causal}}$ $+T\left(4d_{qk} + 3d_{hv} + 2\right)$ $+2\frac{T}{L}d_{qk}$ |
| **fully parallel:** | $T^2F_{\text{causal}}\left(2(d_{qk} + d_{hv}) + 6\right)$ $+2TF_{\text{causal}} + 6.5T^2 + 2.5T$ | $T^2F_{\text{causal}}\left(2(d_{qk} + d_{hv}) + 2\right)$ $+6.5T^2 + 4.5T$ | $4T^2F_{\text{causal}}$ $+2TF_{\text{causal}} - 2.5T$ |
| **recurrent:** | $T\left(6d_{qk}d_{hv} + 7d_{qk} + d_{hv} + 12\right)$ | $T\left(6d_{qk}d_{hv} + d_{qk} + 2\right)$ | $T\left(6d_{qk} + d_{hv} + 10\right)$ |

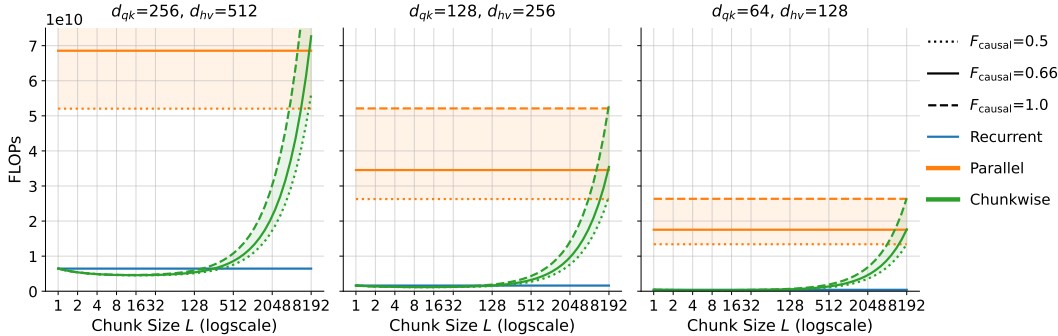

Figure 19: FLOP counts for the recurrent, fully parallel and chunkwise-parallel mLSTM formulation of mLSTMsig. **Left:** $d_{qk}$=256, $d_{hv}$=512. **Middle:** $d_{qk}$=128, $d_{hv}$=256 **Right:** $d_{qk}$=64, $d_{hv}$=128. We count the number of FLOPs for a one head and one sequence of length $T$=8192 for different head dimensions and vary the chunk size $L$. By varying the chunk size $L$ the chunkwise-parallel formulation FLOP counts transition between the recurrent and fully parallel FLOP counts. Smaller head dimensions decrease the overall FLOP count.

**Chunkwise-parallel FLOPs vary between Recurrent and Fully Parallel FLOPs (Figure 19).**
We plot the FLOP counts of the recurrent, fully parallel and chunkwise-parallel mLSTM formulation of mLSTMsig for different chunk sizes $L$ in Figure 19. We observe, that the chunkwise-parallel FLOP counts transition between the recurrent and fully parallel FLOP counts when varying the chunk size $L$ from 1 (recurrent) to $T$ (fully parallel). Smaller head dimensions decrease the overall FLOP count. We also show the impact of the causal factor $F_{\text{causal}}$ on the overall FLOP count. The causal factor $F_{\text{causal}}$ accounts for the causality of the mLSTM and can vary between 0.5 and 1.0. Small values of 0.5 indicate that only past values are computed, while values of 1.0 indicate that all values are computed and then masked out. Efficient implementations achieve values close to 0.5 (see also Appendix F.2).

## F.5    FLOP-Optimal Chunk Size

The number of FLOPs of the chunkwise-parallel formulation of the mLSTM depend on the chunk size $L$. In this section we compute the FLOP-optimal chunk size, i.e. the chunk size $L_{\text{opt,FLOP}}$ that minimizes the FLOP count. We use the simplified FLOP counts for mLSTMsig from Table 18 for this analysis. We denote this FLOP count as $\text{FLOPs}_{\text{mLSTMsig,cwp}}(L)$.

To compute $L_{\text{opt,FLOP}}$ we substitute $d_{qk} = p_{qk}d_{hv}$ and then set the derivative of $\text{FLOPs}_{\text{mLSTMsig,cwp}}(L)$ with respect to $L$ to zero and solve for $L$. This gives

$$L_{\text{opt,FLOP}} = \sqrt{\frac{2d_{hv}^2 p_{qk} + 5}{2F_{\text{causal}}\left(d_{hv}(1 + p_{qk}) + 3\right) + 1}}. \tag{103}$$

The FLOP-optimal chunk size depends on the head dimension $d_{hv}$ and the projection factor $p_{qk}$ and grows proportional to the square root of $d_{hv}$ (i.e. $L_{\text{opt,FLOP}} \propto \mathcal{O}(\sqrt{d_{hv}})$).

**FLOP-Optimal Chunk Size grows with Head Dimension (Figure 20).**    We plot the FLOP-optimal chunk size over head dimension $d_{hv}$ for different projection factors $p_{qk}$, that determine the query-key head dimension $d_{qk} = p_{qk}d_{hv}$ in Figure 20. The FLOP-optimal chunk size grows proportional to the square root of $d_{hv}$ (i.e. $\mathcal{O}(\sqrt{d_{hv}})$), but remains small for typical head dimensions. The projection factor $p_{qk}$ bends the curve, but does not change the overall trend.

In order to minimize the FLOPs to compute, $L_{\text{opt,FLOP}}$ indicates we should use rather small chunk sizes (e.g. $L$=16 for $d_{hv}$=512). However, the chunk size $L$ does not only affect the FLOPs but also the memory IO (for e.g. for loading and storing the memory cell states). Therefore, in order to find the optimal chunksize that minimizes the runtime, we need to consider the memory IO as well, which we do in the next section (see Appendix G).

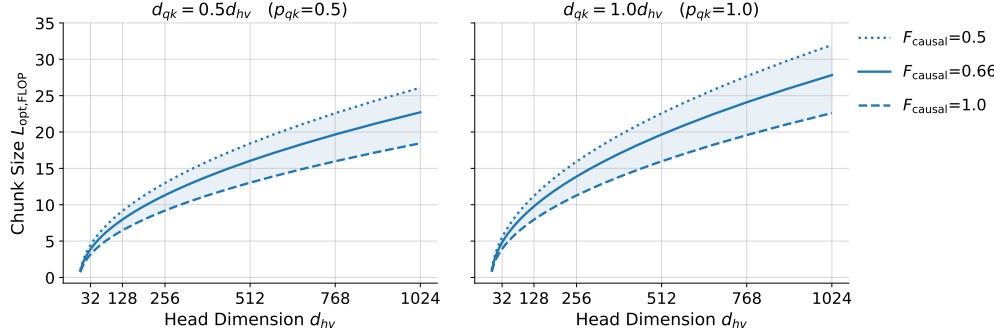

Figure 20: FLOP-Optimal Chunk Size $L_{\text{opt,FLOP}}$ for mLSTMsig. **Left:** $p_{qk}$=0.5. **Right:** $p_{qk}$=1.0. We plot the FLOP-optimal chunk size over head dimension $d_{hv}$ for different projection factors $p_{qk}$, that determine the query-key head dimension $d_{qk} = p_{qk}d_{hv}$. The FLOP-optimal chunk size grows proportional to the square root of $d_{hv}$ (i.e. $\mathcal{O}(\sqrt{d_{hv}})$), but remains small for typical head dimensions.

## G   Theoretical Runtime Analysis for TFLA mLSTM Kernels

In Section 5.2 and E.3 we measure the runtime and memory consumption, experimentally. We see that TFLA needs substantially less GPU memory than other baselines and that there exists an optimal chunksize at which the runtime is minimized.

This runtime minimum is limited and defined by the physical constraints of our hardware (in our case NVIDIA H100 GPUs). Typically, these constraints are how fast the GPU can compute floating point operations (FLOPs) measured in FLOPs per second (FLOPs/s), how fast the GPU can load and store data from and to high-bandwidth memory (HBM) measured as memory bandwidth in bytes per second (B/s), and how much total (HBM) memory is available to store the data in bytes (B) (Austin et al., 2025, Part 1). If the kernel runtime is limited by the maximum FLOPs/s, we say the kernel is compute-bound, and if it is limited by the memory bandwidth, we say the kernel is memory-bound.

In this section, our aim is to theoretically understand to which region our TFLA kernel algorithm for the example of mLSTMsig belongs and what the optimal chunk size would be given the physical constraints of our hardware. We will see that taking hardware constraints into account the optimal chunk size will be much larger than the FLOP optimal chunk size found in the previous section (see Appendix F.5).

We begin with modeling the theoretical runtime of our TFLA kernels in Section G.1 before we compute the arithmetic intensity of TFLA in Section G.2 and use the arithmetic intensity to estimate upper bounds on the peak performance on modern hardware in Section G.3. Finally, in Section G.4 we compute the theoretical runtime optimal chunk size and conclude with a summary of the statements from our analysis in Appendix G.5.

### G.1   Theoretical Runtime

The theoretical runtime of a kernel consists of the time to compute the FLOPs $\tau_{\text{FLOPs}}$ and the time to load and store the inputs, outputs and intermediates from and to the GPU memory $\tau_{\text{Bytes}}$.

Given the number of floating point operations FLOPs$_{\text{algo}}$ and memory operations in bytes to load and store Bytes$_{\text{algo}}$ for a specific algorithm, the accelerator speed $\alpha_{\text{FLOPs}}$ in FLOPS/s and the accelerator memory bandwidth, we can compute the runtimes in seconds as

$$\tau_{\text{FLOPs,algo}} = \frac{\text{FLOPs}_{\text{algo}}}{\alpha_{\text{FLOPs}}} \quad \text{and} \quad \tau_{\text{Bytes,algo}} = \frac{\text{Bytes}_{\text{algo}}}{\beta_{\text{Bytes}}}. \tag{104}$$

For the accelerator speed $\alpha_{\text{FLOPs}}$ and the accelerator memory bandwidth $\beta_{\text{Bytes}}$, we use the hardware specifications of NVIDIA V100[7], A100[8], H100[9] and B200[10] GPUs, which we summarize in Table 19.

If there is no overlap between the computation and the memory operations or in other words if the data is not loaded asynchronously to the computation, the total runtime is the sum of the two, i.e.

$$\tau_{\text{algo,upper}} = \tau_{\text{FLOPs,algo}} + \tau_{\text{Bytes,algo}}. \tag{105}$$

If the computation and memory operations can be completely overlapped, the total runtime is the maximum of the two, i.e.

$$\tau_{\text{algo,lower}} = \max\left(\tau_{\text{FLOPs,algo}}, \tau_{\text{Bytes,algo}}\right). \tag{106}$$

This means the runtime is lower bounded by the maximum of the two and upper bounded by their sum (Austin et al., 2025, Part 1). We use these formulas to compute the theoretical runtime of the TFLA mLSTMsig kernel.

Table 19: **Hardware Accelerator Specification** for NVIDIA GPUs used in this analysis. Values without sparsity. If only the value with sparsity is known, we divide by 2.

| GPU | Year | bfloat16 [FLOPs/s] | Memory Bandwidth [Byte/s] | Arithmetic Intensity [FLOP/byte] |
|---|---|---|---|---|
| V100 SXM2 | 2017 | 120e12 | 0.9e12 | 133 |
| A100 SXM | 2020 | 312e12 | 1.935e12 | 161 |
| H100 SXM | 2022 | 989e12 | 3.35e12 | 295 |
| B200 HGX | 2025 | 2250e12 | 7.7e12 | 292 |

**Theoretical Runtime of TFLA mLSTMsig Forward Pass.** To compute the theoretical runtime of the TFLA mLSTMsig forward pass $\tau_{\text{mLSTMsig}}$, we use the FLOP and memory operation counts for the chunkwise-parallel formulation from Table 18 and Table 10 in Appendix F. We denote the FLOP count as $\text{FLOPs}_{\text{mLSTMsig}}$ and the memory operation count in bytes as $\text{Bytes}_{\text{mLSTMsig}}$.

We assume that memory operations are not overlapped with computation, because (1) in our current implementation of TFLA, we first materialize all states in the recurrent kernel before we launch the parallel kernel (see Figure 1) and (2) we do not use advanced hardware features of NVIDIA GPUs (yet), such as asynchronous memory loading, which would allow to overlap memory operations with computation. Therefore, the total theoretical runtime is the sum of the FLOP and memory operation runtimes, i.e.

$$\tau_{\text{mLSTMsig}} = \frac{\text{FLOPs}_{\text{mLSTMsig}}}{\alpha_{\text{FLOPs}}} + \frac{\text{Bytes}_{\text{mLSTMsig}}}{\beta_{\text{Bytes}}}. \tag{107}$$

By inserting the expressions from above and multiplying by the number of heads $N_{\text{head}}$ and batch size $N_{\text{batch}}$, we obtain the total runtime of the mLSTMsig forward pass as

$$\tau_{\text{mLSTMsig}} = N_{\text{batch}} \cdot N_{\text{head}} \cdot \frac{T}{L} \cdot \left( \frac{6Ld_{hv}\left(1 + p_{qk}\right) + 8L + 2\text{bytes}_{Cmn}d_{hv}^2 p_{qk}}{\beta_{\text{Bytes}}} \right.$$
$$\left. + \frac{L^2 F_{\text{causal}}\left(2d_{hv}p_{qk} + 2d_{hv} + 6\right) + L^2 + L\left(4d_{hv}^2 p_{qk} + 2d_{hv}p_{qk} + d_{hv} + 11\right) + 2d_{hv}^2 p_{qk} + 5}{\alpha_{\text{FLOPs}}} \right), \tag{108}$$

where we assume that the queries, keys, values and input and forget gate are stored in `bfloat16`, i.e. $\text{bytes}_{if} = \text{bytes}_{qkv} = 2$.

The theoretical runtime depends on the model architecture (e.g. head dimension $d_{hv}$ or query-key projection factor $p_{qk}$), kernel parameters (e.g. chunk size $L$, $F_{\text{causal}}$ or $\text{bytes}_{Cmn}$) and the hardware accelerator specifications (e.g. $\alpha_{\text{FLOPs}}$ or $\beta_{\text{Bytes}}$).

---

[7] https://www.nvidia.com/en-au/data-center/v100/
[8] https://www.nvidia.com/en-us/data-center/a100/
[9] https://resources.nvidia.com/en-us-tensor-core/nvidia-tensor-core-gpu-datasheet
[10] https://resources.nvidia.com/en-us-blackwell-architecture/datasheet

**Theoretical Runtime over Chunk Size (Figure 21).** We show the theoretical runtime of our mLSTMsig kernel for 7B model size (i.e. $d_{hv}$=512, $p_{qk}$=0.5, $N_{\text{head}}$=8) for $N_{\text{batch}}$=8, $T$=8192 and bytes$_{Cmn}$=4 on different NVIDIA A100, H100 and B200 GPUs (according to Table 19) in Figure 21. We observe that newer GPUs (e.g. H100 or B200) are faster and have a higher memory bandwidth, which results in a lower runtime. Moreover, for newer GPUs, the runtime becomes less sensitive to the chunk size $L$, as the curve becomes flatter. Moreover, there exists an optimal chunk size that minimizes the runtime, which is determined by the physical constraints of the hardware. This optimal chunk size increases on more recent GPUs (e.g. B200). We explore the runtime-optimal chunk size more in depth in Appendix G.4.

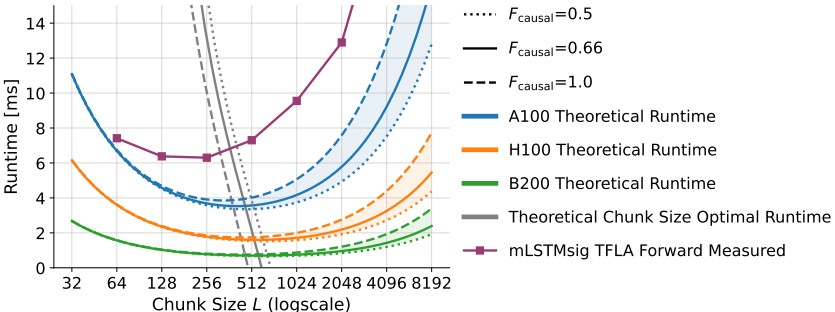

Figure 21: Theoretical Runtime of TFLA mLSTMsig Forward Pass with varying Chunk Size $L$ on different accelerators. We plot the theoretical runtime of mLSTMsig with 7B size (i.e. $d_{hv}$=512, $p_{qk}$=0.5, $N_{\text{head}}$=8) for $N_{\text{batch}}$=8, $T$=8192 and bytes$_{Cmn}$=4 on NVIDIA A100, H100 and B200 GPUs (according to Tab. 19). We also plot the measured runtime of the mLSTMsig kernel on NVIDIA H100. Newer GPUs (e.g. H100, B200) are faster and have a higher memory bandwidth, which results in a lower theoretical runtime.

**Discrepancy between Measured and Theoretical Runtime (Figure 21)** In Figure 21, we additionally compare the measured runtime of mLSTMsig kernels on NVIDIA H100 with the theoretical runtime. While the measured runtime is higher than the theoretical runtime, the qualitative trend of the runtime over the chunk size is similar.

There are several reasons for the discrepancy between the measured and theoretical runtime. First, the model of the runtime of our mLSTMsig kernels in Equation (108) has approximation errors. While we account for the uncertainty in the causal factor $F_{\text{causal}}$, when counting the FLOPs and memory operations, we still make approximations and simplifications. For example, we do not account for the recomputation and reloading of data within the kernels (see Appendix F). Addditionally, we do not model the fact that our computation consists of multiple kernels (e.g. the recurrent and parallel kernel (see Figure 1)). Hence, we do not include the delay when multiple kernels are launched in sequence.

Second, in addition to model errors our kernels are not optimized to reach peak performance. For example, we do not use advanced features of the H100 GPUs, such as asynchronous memory loading, which would allow to overlap memory operations with computation.

Finally, the hardware specifications (e.g. $\alpha_{\text{FLOPs}}$) specify the peak performance of the accelerator for matrix multiplications. Our computation for the mLSTM (see Section 2.2) includes several pointwise and vector operations, which are not performed on tensor cores and are therefore slower.

Therefore, we do not expect our theoretical runtime model to perfectly match the measured runtime, but it provides a good approximation of the overall qualitative runtime behavior of our mLSTM kernels, that could guide further optimization efforts.

### G.2 Arithmetic Intensity

For a specific kernel we can compute the time to compute the FLOPs $\tau_{\text{FLOPs}}$ and the time to load and store from and to the GPU memory $\tau_{\text{Bytes}}$. We can then distinguish two cases: (1) $\tau_{\text{FLOPs}} > \tau_{\text{Bytes}}$: The runtime is dominated by the computation time. We call this being *compute-bound*. (2) $\tau_{\text{FLOPs}} < \tau_{\text{Bytes}}$: The runtime is dominated by memory loading. We call this being *memory-bound*.

Instead of comparing the times, we can also compare the *arithmetic intesity* or *operational intensity* of our algorithm and our hardware to determine if the kernel is compute-bound or memory-bound.

The arithmetic intensity directly relates the number of FLOPs to the number of bytes of GPU memory traffic (Williams et al., 2009). We can compute the arithmetic intensity for our accelerator $\mathcal{I}_{\text{acc}}$ or our algorithm $\mathcal{I}_{\text{algo}}$ by computing the ratio between the accelerator speed and the memory bandwidth or the number of FLOPs and the number of bytes loaded and stored, i.e.

$$\mathcal{I}_{\text{acc}} = \frac{\alpha_{\text{FLOPs}}}{\beta_{\text{Bytes}}} \quad \text{and} \quad \mathcal{I}_{\text{algo}} = \frac{\text{FLOPs}_{\text{algo}}}{\text{Bytes}_{\text{algo}}}. \tag{109}$$

Then, (1) if $\mathcal{I}_{\text{algo}} > \mathcal{I}_{\text{acc}}$, the kernel is likely to be compute-bound and (2) if $\mathcal{I}_{\text{algo}} < \mathcal{I}_{\text{acc}}$, the kernel is likely memory-bound. The accelerator arithmetic intensity $\mathcal{I}_{\text{acc}}$ is the minimum arithmetic intensity required to achieve maximum performance of the accelerator.

**Arithmetic Intensity of TFLA mLSTMsig Forward Pass.** We compute the arithmetic intensity of the TFLA mLSTMsig forward pass $\mathcal{I}_{\text{mLSTMsig}}$ by computing the ratio between the total FLOP count $\text{FLOPs}_{\text{mLSTMsig}}$ and the memory operation count in bytes as $\text{Bytes}_{\text{mLSTMsig}}$. This gives

$$\mathcal{I}_{\text{mLSTMsig}} = \frac{L^2 F_{\text{causal}} \left( 2d_{hv}p_{qk} + 2d_{hv} + 6 \right) + L^2 + L \left( 4d_{hv}^2 p_{qk} + 2d_{hv}p_{qk} + d_{hv} + 11 \right) + 2d_{hv}^2 p_{qk} + 5}{6Ld_{hv}\left( 1 + p_{qk} \right) + 8L + 2\text{bytes}_{Cmn} d_{hv}^2 p_{qk}}. \tag{110}$$

The arithmetic intensity depends on the model architecture (e.g. head dimension $d_{hv}$ or query-key projection factor $p_{qk}$) and the kernel parameters (e.g. chunk size $L$, $F_{\text{causal}}$ or $\text{bytes}_{Cmn}$).

**Arithmetic Intensity over Chunk Size (Figure 22).** We plot the arithmetic intensity of the TFLA mLSTMsig forward pass over the chunk size $L$ in Figure 22 and vary the head dimension $d_{hv}$ and the precision of the memory cell states $\text{bytes}_{Cmn}$. Additionally, we indicate the arithmetic intensity $\mathcal{I}_{\text{acc}}$ for different NVIDIA GPUs (according to Table 19). Values above the accelerator arithmetic intensity indicate that the kernel is likely to be compute-bound for the corresponding chunk sizes, while values below indicate that the kernel is likely memory-bound.

We observe that the arithmetic intensity increases for larger chunk sizes and that the kernel arithmetic intensity curve crosses the accelerator arithmetic intensity at larger chunk sizes. This means, we can move from the memory-bound regime to the compute-bound regime by increasing the chunk size parameter $L$.

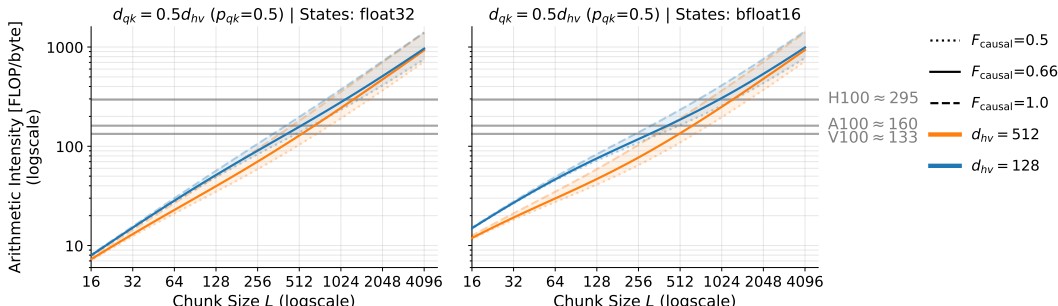

Figure 22: Arithmetic Intensity of TFLA mLSTMsig Forward Pass with varying Chunk Size $L$. **Left:** States in `float32`, i.e. $\text{bytes}_{Cmn}$=4. **Right:** States in `bfloat16`, i.e. $\text{bytes}_{Cmn}$=2. The arithmetic intensity increases for larger chunk sizes.

### G.3 Roofline Analysis

The roofline model is a performance model that combines the arithmetic intensity and the accelerator specifications to determine the upper bound on the peak performance of a kernel (Williams et al., 2009). To do so, the roofline model plots the arithmetic intensity of the algorithm on the x-axis and the attainable performance in FLOPs per second (FLOPs/s) on the y-axis. For each hardware accelerator we plot the roofline, which is the maximum performance of the accelerator for a given arithmetic intensity as

$$\text{Roofline}(\alpha_{\text{FLOPs}}, \beta_{\text{Bytes}}) = \min\left( \beta_{\text{Bytes}} \cdot \mathcal{I}_{\text{algo}}, \alpha_{\text{FLOPs}} \right). \tag{111}$$

If we then plot the arithmetic intensity of a kernel as a column that hits the roof, either it hits the flat part of the roof, meaning performnace is compute-bound or performance is ultimately memory bound Williams et al. (2009).

**Roofline Model for TFLA mLSTMsig Forward Pass (Figure 23).** We perform a roofline analysis for our TFLA mLSTMsig forward kernels in Figure 23 for different chunk sizes $L$ and plot the rooflines for NVIDIA V100, A100, and H100 GPUs (according to Table 19).

We observe that smaller chunk sizes are memory-bound, while larger chunk sizes are compute-bound (similar to Figure 22). Moreover, we find that our TFLA Triton kernels for the mLSTMsig, which we benchmark on NVIDIA H100, are still far from the attainable peak performance (intersection with H100 roofline). This highlights the potential for further optimization.

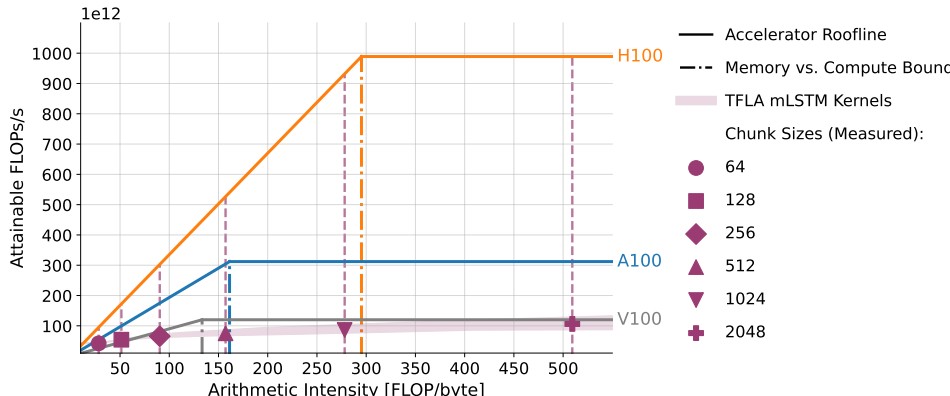

Figure 23: Roofline Model for TFLA mLSTMsig Forward Pass with varying Chunk Size $L$. We measure the performance of mLSTMsig with $d_{hv}$=512, $p_{qk}$=0.5, $N_{\text{head}}$=8, and batch size 8 at sequence length 8192 and compare the performance with the roofline of NVIDIA V100, A100 and H100 GPUs (according to Table 19). Smaller chunk sizes are memory-bound, while larger chunk sizes are compute-bound. Our TFLA Triton kernels for the mLSTMsig are still far from the attainable peak performance (intersection with H100 roofline).

**FLOPs/s is not the optimal Performance Metric for TFLA (Figure 24).** In Figure 23 we observe and the roofline model also suggests that we can increase the performance of our TFLA kernels by increasing the chunk size $L$. However, while this might increase the performance in FLOPs/s, it might not decrease the overall runtime, as number of FLOPs increase with the chunk size $L$ due to the increased quadratic term in the FLOP count (see Figure 19).

We confirm this in Figure 24, where we plot the FLOPs/s and the total FLOPs over the chunk size $L$ and compare the values with the actual runtime. We observe that the FLOPs/s continuously increase with the chunk size $L$, but the total FLOPs also increase. This means that while we could reach peak performance in FLOPs/s by increasing the chunk size $L$, the actual runtime is not necessarily minimized by doing so. Therefore, for our TFLA kernels we should use the actual runtime as final performance metric and to determine the runtime-optimal chunk size, which we do next.

## G.4  Runtime-Optimal Chunk Size

The main advantage of TFLA is that we can choose arbitrary chunk sizes $L$ to trade off between FLOPs and memory IO and minimize the runtime (see Figure 6 and 24). The reason for this tradeoff is that the chunk size $L$ affects the number of FLOPs and the memory IO (e.g. for loading and storing the memory cell states) (see Appendix F). In this section, we use this insight and our theoretical runtime model from Appendix G.1 to determine the theoretical runtime-optimal chunk size for the TFLA mLSTMsig forward kernel.

**Runtime-Optimal Chunk Size for TFLA mLSTMsig Forward Pass.**    In order to compute the runtime-optimal chunk size $L_{\mathrm{opt,Runtime}}$ for the TFLA mLSTMsig forward pass, we use the theoretical runtime $\tau_{\mathrm{mLSTMsig}}$ from Equation (108). We first differentiate the theoretical runtime with respect to the chunk size $L$ and set the derivative to zero to find the minimum runtime, i.e.

$$\frac{\partial \tau_{\mathrm{mLSTMsig}}}{\partial L} = 0. \tag{112}$$

We then solve the equation for the chunk size $L$ to find the runtime-optimal chunk size $L_{\mathrm{opt,Runtime}}$. This yields

$$L_{\mathrm{opt,Runtime}} = \sqrt{\frac{2\,d_{hv}^2 p_{qk} + 5 + 2\,\mathcal{I}_{\mathrm{acc}}\,d_{hv}^2 p_{qk}\,\mathrm{bytes}_{Cmn}}{2F_{\mathrm{causal}}\left(d_{hv}\left(1 + p_{qk}\right) + 3\right) + 1}}, \tag{113}$$

where $\mathcal{I}_{\mathrm{acc}} = \frac{\alpha_{\mathrm{FLOPs}}}{\beta_{\mathrm{Bytes}}}$ is the accelerator arithmetic intensity from Equation (109). Compared to the FLOP-optimal chunk size $L_{\mathrm{opt,FLOP}}$ (see Equation (103)), the runtime-optimal chunk size $L_{\mathrm{opt,Runtime}}$ additionally depends on the arithmetic intensity $\mathcal{I}_{\mathrm{acc}}$ of our hardware and the precision of our states $\mathrm{bytes}_{Cmn}$. The runtime-optimal chunk size grows proportional to the square root of the head dimension $d_{hv}$ (i.e. $\mathcal{O}(\sqrt{d_{hv}})$) and the accelerator arithmetic intensity $\mathcal{I}_{\mathrm{acc}}$ (i.e. $\mathcal{O}(\sqrt{\mathcal{I}_{\mathrm{acc}}})$). We visualize these trends in Figure 25 and 26.

**Runtime-Optimal Chunk Size depends on Model Architecture (Figure 25).**    We plot the runtime-optimal chunksize for the TFLA mLSTMsig forward pass $L_{\mathrm{opt,Runtime}}$ over the head dimension $d_{hv}$ in Figure 25 for memory cell states in `float32` and `bfloat16` and NVIDIA A100 and H100 arithmetic intensities.

The runtime-optimal chunk size grows proportional to the square root of the head dimension $d_{hv}$ (i.e. $\mathcal{O}(\sqrt{d_{hv}})$) and is much larger than the FLOP-optimal chunk size.

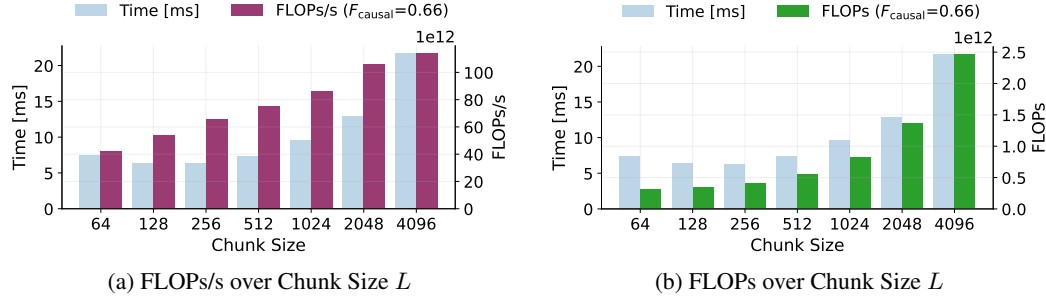

(a) FLOPs/s over Chunk Size $L$  (b) FLOPs over Chunk Size $L$

Figure 24: FLOPs/s and FLOPs over Chunk Size $L$ for TFLA mLSTMsig Forward Pass. We compare the FLOPs/s and the total FLOPs with the actual runtime at different chunk sizes. We measure the performance of mLSTMsig with $d_{hv}$=512, $p_{qk}$=0.5, $N_{\mathrm{head}}$=8, and batch size 8 at sequence length 8192. The FLOPs/s increase with the chunk size $L$, but the total FLOPs also increase. The actual runtime is not necessarily minimized by increasing the chunk size $L$.

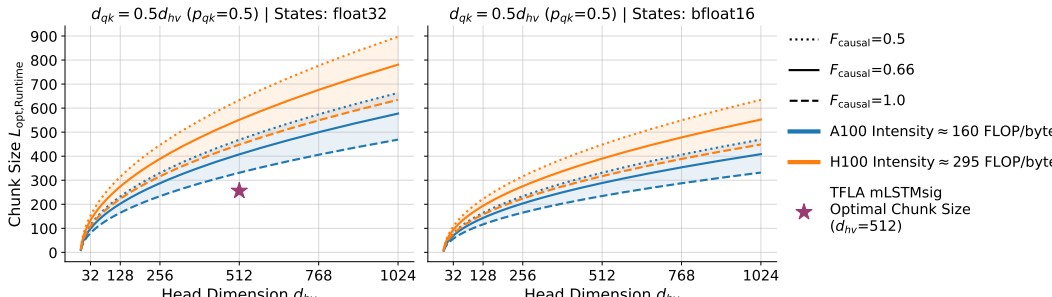

Figure 25: Runtime-Optimal Chunk Size $L_{\text{opt,Runtime}}$ over head dimension $d_{hv}$ for mLSTMsig. **Left:** Memory cell states in `float32`. **Right:** Memory cell states in `bfloat16`. We plot the runtime-optimal chunk size over head dimension $d_{hv}$ with $d_{qk} = 0.5d_{hv}$ ($p_{qk}$=0.5). Similar to the FLOP-optimal chunk size (Fig. 20), the runtime-optimal chunk size grows proportional to the square root of $d_{hv}$ (i.e. $\mathcal{O}(\sqrt{d_{hv}})$), but is much larger than the FLOP-optimal chunk size.

Our measured runtime-optimal chunk size for mLSTMsig on NVIDIA H100 is around 256, which is smaller than the theoretical runtime-optimal chunk size. This discrepancy is due to the approximations in our theoretical runtime model and the fact that our kernels are not yet optimized to reach peak performance on NVIDIA H100 GPUs (see Appendix G.1), but are already faster than almost all all other baseline kernels and than Flash Attention 3, which is optimized for NVIDIA H100 GPUs (see Section 5.2).

**Runtime-Optimal Chunk Size depends on Hardware Accelerator (Figure 26).** We plot the runtime-optimal chunk size for the TFLA mLSTMsig forward pass $L_{\text{opt,Runtime}}$ over the hardware accelerator intensity in Figure 26 for different head dimensions $d_{hv}$. We highlight NVIDIA GPU accelerator intensities for common GPUs (e.g. V100, A100, H100).

The runtime-optimal chunk size grows proportional to the square root of the accelerator intensity $\mathcal{I}_{\text{acc}}$ (i.e. $\mathcal{O}(\sqrt{\mathcal{I}_{\text{acc}}})$). More recent GPUs (like e.g. H100) have higher accelerator intensities, which results in a larger runtime-optimal chunk size. If the trend of increasing accelerator intensities continues, TFLA that enables arbitrary large chunk sizes will become increasingly important.

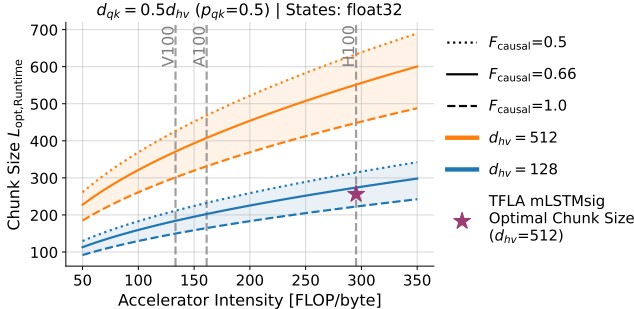

Figure 26: Runtime-Optimal Chunk Size $L_{\text{opt,Runtime}}$ over Hardware Accelerator Intensity for mLSTMsig. We plot the runtime-optimal chunk size over the hardware accelerator intensity for different head dimensions $d_{hv}$. We highlight NVIDIA GPU accelerator intensities for commong GPUs (e.g. V100, A100, H100). The runtime-optimal chunk size grows with the accelerator intensity (i.e. $L_{\text{opt,Runtime}} \propto \mathcal{O}(\sqrt{\mathcal{I}_{\text{acc}}})$).

### G.5 FLOP and Theoretical Runtime Analysis Summary

The configurable chunk size $L$ is the core advantage of Tiled Flash Linear Attention. We summarize the statements about the chunk size:

1. The chunk size $L$ mediates a trade-off between runtime and GPU memory usage. [Figure 6]
2. $L$ determines the total compute in FLOPs: $L = 1$ matches the recurrent formulation, while $L = T$ matches the parallel one. [Figure 19]
3. There exists an optimal chunk size $L \in [1, T]$ that minimized the total FLOP count. [Equation (103), Figure 19, Figure 20]
4. Increasing $L$ raises the arithmetic intensity of TFLA kernels. [Equation (109), Figure 22]
5. The chunk size determines whether the kernel is memory-bound or compute-bound on a given hardware.
   [Figure 23, Figure 22]
6. FLOPs/s alone can be misleading; the optimal chunk size should be chosen based on total runtime.
   [Figure 24, Figure 21]
7. he runtime-optimal chunk size scales proportionally with the square root of the head dimension and the accelerator's computational intensity.
   [Figure 25, Figure 26]
8. Newer hardware generations require larger chunk sizes to approach peak performance.
   [Figure 26, Figure 21]

