# OpenReview forum: "Tiled Flash Linear Attention: More Efficient Linear RNN and xLSTM Kernels"
_NeurIPS.cc/2025/Conference — NeurIPS 2025 poster_

### Official Review · Reviewer_5h29 · 2025-07-01

**Clarity:** 3
**Significance:** 3
**Originality:** 3
**Rating:** 4
**Confidence:** 2

**Summary:**

This paper introduces Tiled Flash Linear Attention (TFLA), a new kernel algorithm for linear RNNs designed for high efficiency on modern GPUs. TFLA extends prior work like Flash Linear Attention by adding a second level of sequence parallelism via tiling within each chunk. This technique enables arbitrarily large chunk sizes, improving arithmetic intensity and reducing memory I/O. The authors apply TFLA to xLSTM (mLSTM) and propose a simplified, faster variant, mLSTMsig. Extensive benchmarks on H100 GPUs demonstrate that TFLA kernels outperform highly optimized baselines, including FlashAttention 3 and Mamba 2 in training speed.

**Questions:**

Please see the weakness.

**Ethical Concerns:**

["NO or VERY MINOR ethics concerns only"]

**Final Justification:**

This paper proposes a well-motivated and clever kernel optimization technique.

**Limitations:**

yes

**Paper Formatting Concerns:**

No major formatting issues.

**Quality:**

3

**Strengths And Weaknesses:**

Strengths:

1. The paper proposes a well-motivated and clever kernel optimization technique, TFLA. By combining the chunking parallelism of linear RNNs with the intra-chunk tiling from attention mechanisms, it effectively addresses the SRAM size limitation of previous methods. This enables a flexible and powerful trade-off between computation and memory usage.

2. The experimental results are extensive and compelling. The proposed TFLA kernels demonstrate significant speedups over state-of-the-art and highly optimized baselines, including FlashAttention 3 and Mamba 2, on modern hardware. This establishes a new state of the art for efficient long context sequence modeling primitives.

3. Beyond kernel optimization, the paper provides valuable insights into the model architecture itself. The introduction and analysis of mLSTMsig, along with the empirical study of gate initialization and its effect on training stability, adds significant depth and practical value to the work.

Weaknesses:

1. The core idea of TFLA, while effective, is a combination of two existing techniques: chunk-wise parallelism from Flash Linear Attention and intra-chunk tiling from FlashAttention. The novelty lies in the synthesis and implementation of these ideas rather than a fundamentally new algorithmic paradigm.

2. The benchmark comparison in Figure 5 uses different head configurations for the proposed model versus key baselines. For example, TFLA uses 16 heads while FlashAttention uses 32 heads. While the total embedding dimension is constant, this difference can influence hardware performance and make a direct, apples-to-apples speed comparison less clear.

3. The paper reports training instabilities like gradient spikes for certain configurations, particularly for the xl_chunk kernels shown in Table 2 and Table 5. While the authors identify a mitigation strategy through bias initialization, the root cause of the instability is not fully diagnosed. It is unclear if it is an inherent model property or a numerical precision issue in the custom kernel.

---

> ### Author Rebuttal · Authors · 2025-07-28
>
> We thank the reviewer for the thorough and constructive feedback and appreciate the recognition of the TFLA kernel as a well-motivated and effective solution to SRAM limitations, along with the acknowledgement of the strong experimental results and architectural contributions—particularly the analysis of mLSTMsig and gate initialization.
>
> We would like to address the reviewers concerns in the following:
>
> We explain the benchmark setup of Figure 5 in detail in Section 5.2 and Appendix A.3. The main motivation for the benchmark setup is to assess the kernel runtime in a setting, which would be used for training a model of size ~7B parameters with embedding dimension typically 4096.
>
> We now aimed to select also typical head dimensions that would be used for Llama or Mamba2 models. Specifically, we used configs from huggingface, where available. For Llama for example we used the Meta-Llama-8B, which has 32 heads and for Mamba2 we used Codestral-Mamba2 which has a head dim of 64 as reference.
>
> For our model, we preferred slightly larger head dimensions (16 heads, d_hv = 256, d_qk = 128) as this leads to a larger memory state and similar perplexity.
> To analyse the effect of the head dimension on the runtime we perform a benchmark with varying head dimensions in Figure 18 in the Appendix. We observe that smaller head dimensions (i.e. more heads) lead to even faster runtimes. This means that the runtimes of mLSTM are expected to be lower if we choose the same (small) head dim as Mamba2, but in practice one would choose larger head dimensions for mLSTM, which is why we made this decision.
>
> Regarding the numerical issues, we verified correctness of the kernel implementations in float32 and confirmed that training with different implementations JAX vs. our Triton kernels yield the same results (see Table 3).
>
> We sincerely thank the reviewer for their thoughtful feedback, and hope our responses have addressed the reviewers questions.

---

> > ### Comment · Reviewer_5h29 · 2025-08-05
> > **Thank you for your rebuttal.**
> >
> > Thank you for your rebuttal. Most of concerns have been solved.

---

### Official Review · Reviewer_uJbu · 2025-07-01

**Clarity:** 4
**Significance:** 3
**Originality:** 3
**Rating:** 5
**Confidence:** 4

**Summary:**

The paper addresses the problem of accelerating training and inference of LLM on GPUs. It proposes an implementation applicable to any linear RNNs types of LLMs, called Tiled Flash Linear Attention (TFLA). TFLA builds upon FlashLinearAttention (the version of Flash attention for linear RNN), which is based on a chunkwise-parallel implementation (recurrent inter-chunk, parallel intra-chunk). Inspired by FlashAttention2, TFLA adds another level of intra-chunk parallelism enabling arbitrary chunk size. This flexibility allows to chose the chunk size to balance the number of operations and the number of costly memory accesses to the GPU High Bandwith Memory, depending on model architecture and hardware characteristics, to achieve optimal runtime. The method is applied on the mLSTM architecture, on which some modifications are proposed to enable even faster inference. The resulting mLSTM architecture with the chunkwise-parallel acceleration yields faster training than efficient implementations of FlashAttention and linear RNN (Mamba 2), while the inference time is comparable to other linear RNN implementations.

**Questions:**

1)	Improvement of the inference runtime. Why is there a bigger difference in inference and training time compared to other linear RNN methods (e.g. Mamba 2) ? How could inference speed be improved ? Indeed, training time is a major limitation to AI development, but inference time is crucial for reducing the costs of AI deployment.
2)	Improvement of the method for modern GPUs. As the FLOPs/s in hardware increases faster than the memory bandwidth, the authors observe that the optimal chunk size increases in modern GPUs. Moreover, the proposed method yields lower benefits in more recent GPUs than older ones. Knowing this trend, do the authors see limitations of their method as GPUs improve ? Could the authors comment on the benefit of their method on future hardware ?
3)	Compatibility with other linear RNNs. Knowing that Mamba 2 is compatible with such acceleration and is currently the fastest in inference runtime, it would be interesting to know how this acceleration would performed compared to the one originally implemented for Mamba2.
4)	Extension of the method to other hardware. The proposed method is intended for cloud-type GPUs. Would this method be relevant for embedded GPUs (e.g. Jetson), although they are targeted for inference loads rather than training loads ?

**Ethical Concerns:**

["NO or VERY MINOR ethics concerns only"]

**Final Justification:**

After reading the other reviewers’ comments and the authors’ response, I want to maintain my acceptance recommendation. My concerns regarded the inference runtime compared to other efficient linear RNN (Mamba2) and the authors clarified that it depends on the head dimension that was chosen for each of the models. Another concern was about the relevance of the method for modern GPU. The authors clarified that, by enabling arbitrary large chunk sizes, the method is well suited to modern GPUs that benefits from larger chunk size. The authors did a very good work in analysing their method, facilitating its practical use. The remaining limitation concerns the limited evaluation of the proposed mLSTM variant. In particular, I suggest to evaluate accuracy in downstream tasks, rather than evaluating only perplexity. This may further motivate the adoption of the model. I believe this can be addressed in future work.

**Limitations:**

yes

**Quality:**

3

**Strengths And Weaknesses:**

Strengths :
- Quality : The paper seems technically correct and complete, with analysis allowing to understand the challenges and the proposed method. Limitations and future works are properly aknowledge, consisting mainly in further optimizations to get closer to the theoretical peak performance.
- Clarity : The paper provides sufficient details to reproduce the experiments. There are many analysis (theoretical and experimental) to understand the proposed method and how to use it.
- Significance : The applicability of the proposed method is large. Indeed, the paper shows that the proposed acceleration is not only applicable to the mLSTM architecture but also to most of linear RNNs.
- Originality : The paper provides an interesting analysis of the method, with theoretical and experimental results on key performance indicators (FLOPs/s, runtime...) and especially how they should be used.

Weaknesses
- Quality : The authors propose modifications on the mLSTM layer to be implemented more efficiently (for instance replacing an exponential activation function by a sigmoid, and removing the normalization state), therefore they need to evaluate the impact of the proposed modifications on the accuracy of the model. This evaluation is relatively limited (only 1 dataset, with only 1 run per experiment).
- Signifiance : While the method yields state-of-the-art training time, it is not the case for inference time. Moreover, the authros observed that the proposed method yields lower benefits in more recent GPUs than older ones.

---

> ### Author Rebuttal · Authors · 2025-07-28
>
> We thank the reviewer for the helpful feedback which helped to improve our paper.
> We appreciate that the reviewer highlights the general applicability of our method and sees strengths in our extensive analysis in the paper.
>
> We would like to address the reviewers questions as follows:
>
> 1.)
> In Figure 5a (Forward pass only / Inference runtime measurements) the runtime of Mamba2 forward pass is lower than for the mLSTM TFLA kernels. This can be explained by the fact that Mamba2 uses smaller head dimensions (or the respective equivalent) of 64 by default. However, for mLSTM and other linear RNNs the head dimension is typically larger (which we confirm in our experiments). Therefore, we kept larger head dimensions for the mLSTM in Fig. 5 and added an ablation on the effect of the head dimension at fixed batch size in Fig. 18. We find that TFLA runtimes are faster for smaller head dimensions. (A side note: The broad range of possible head dimensions for TFLA is another advantage compared to Mamba2.)
>
> In the current form we optimized the kernels for training stability and kept the precision dtype of the C states in float32 in forward and backward. Future optimizations for maximal inference performance could explore lower precisions (e.g. bfloat16 for the memory states) for example.
>
> 2.)
> In our theoretical runtime analysis in Fig. 21 and 22 we show that newer hardware generations tend to have a higher arithmetic intensity and that the (theoretical) runtime optimal chunk size increases with newer hardware generations due to the higher arithmetic intensity.
>
> Since our TFLA algorithm enables arbitrary large chunk sizes it is well suited for this trend, as our analysis shows that large chunk sizes will be necessary for optimal performance on future hardware generations.
>
> Finally, as the region of optimal runtime in Fig. 21 becomes wider for newer hardware generations, we suspect that we could increase the chunk size even further to reduce GPU memory consumption and potentially use larger batch sizes to increase efficiency.
>
> 3.)
> Please see 1.)
>
> 4.)
> We believe that the general principles (arithmetic intensity, memory vs. compute trade-off) from our theoretical runtime analysis of TFLA apply also to other GPU models, such as NVIDIA Jetson GPUs.
> However, at the time of writing the paper we had only H100 GPUs available. Therefore, we could not test TFLA on other GPUs, such as Jetson GPUs for embedded systems.
> In the future we plan to test TFLA also on other hardware, which might also require rewriting the kernels in other frameworks than Triton.
>
>
> We sincerely thank the reviewer for their thoughtful feedback, and hope our responses have addressed all remaining questions.

---

> > ### Comment · Reviewer_uJbu · 2025-08-05
> >
> > Thank you for your answer, this clarifies my concerns. After reading the other reviewers’ comments and the authors’ response, I want to maintain my acceptance recommendation. You did a significant work in analysing the efficiency of the proposed method. In future work, I suggest you to address the main limitation of this paper in my opinion – the limited evaluation in terms of accuracy of the proposed mLSTM. In particular, I suggest you to evaluate accuracy in downstream tasks, rather than evaluating only perplexity. This may further motivates the adoption of the model.

---

### Official Review · Reviewer_i6Dq · 2025-07-01

**Clarity:** 4
**Significance:** 4
**Originality:** 3
**Rating:** 5
**Confidence:** 4

**Summary:**

This paper introduces Tiled Flash Linear Attention (TFLA), a novel kernel for linear RNNs that improves existing Flash Linear Attention (FLA) kernels. The key innovation is the introduction of two levels of sequence parallelism: first over chunks of the input sequence, and second within each chunk through tiling of matrix computations. This modification is key, as it enables larger chunk sizes while maintaining high arithmetic intensity.
The authors apply TFLA to the mLSTM architecture and also propose mLSTMsig, a new variant that uses sigmoid input gates instead of exponential gates to accelerate computation.
The paper shows a comprehensive theoretical analysis including FLOP counts, memory operation analysis, and theoretical runtime modeling.
Experimental results demonstrate that TFLA kernels outperform highly optimized baselines including FlashAttention 3 and Mamba 2 kernels on H100 GPUs, particularly for longer sequences during training.

- Small note: Use \citep instead of \citet in Line 103.

**Questions:**

- Can you provide more detailed analysis of numerical stability implications when using larger chunk sizes?

- What specific guidance can you offer practitioners for choosing between TFLA and existing methods based on problem characteristics?

- How sensitive are the results to the chunk size selection strategy, and can this process be automated?

- Can you expand the experimental validation to other linear RNN architectures beyond mLSTM to better support the generalizability claims?

**Ethical Concerns:**

["NO or VERY MINOR ethics concerns only"]

**Final Justification:**

I have decided to keep my score unchanged. This is paper represents a solid work, and the authors have addressed my concerns convincingly during the rebuttal.

**Limitations:**

The authors briefly discuss the limitations in the conclusion and theoretical analysis sections. They acknowledge that their Triton implementations may not achieve peak performance and that the theoretical model has approximation errors. However, the paper could better address the numerical stability issues observed in some experiments and provide clearer guidance on when these might occur.

**Quality:**

3

**Strengths And Weaknesses:**

**Strong Points**


- The paper clearly identifies and addresses the constraint on chunk sizes due to GPU SRAM limitations, which leads to excessive memory consumption and IO costs for long sequences and is a real limitation of existing FLA implementations.

- The two-level sequence parallelism design (chunking + intra-chunk tiling) is technically sound and follows established GPU optimization principles. The mathematical formulations are consistent and, as far as I could check, the implementation details seem correct to me

- The paper provides both theoretical analysis (runtime complexity, arithmetic intensity, FLOP counts) and empirical validation through language modeling experiments and detailed kernel benchmarks against strong baselines

- The paper shows concrete speedups over highly optimized baselines including FlashAttention-3 and Mamba-2 kernels, with particularly strong results for longer sequences.

- Sufficient implementation details are provided, with clear algorithmic descriptions.

- Sections 5.2 and 5.3 are great. Figure 6/21 (and the analysis there-in) is very insightful. Well done!

- Overall, the paper is really well-written and the technical details are introduced in a clear manner.




**Weak Points**

- The core contribution essentially combines existing techniques (chunking from FLA + tiling from FlashAttention). While the specific combination is novel, the individual components are well-established, somewhat limiting the novelty of this work.

- The primary application is to mLSTM, which is a recent architecture with limited adoption. While the paper claims broader applicability to other linear RNNs, this is mostly theoretical with limited empirical validation.

- The comparison between mLSTMexp and mLSTMsig shows minimal performance differences, questioning the novelty value of the sigmoid variant. Also, the motivation behind seems a bit related to [Dynamic Tanh (DyT)](https://arxiv.org/abs/2503.10622), which I think deserves a citation here.

In a more concrete manner, there are some minor mathematical details that needs some rethinking to improve reading:

- The jump from equations 1-5 to equation 6-7 are unclear.
- The way $\tilde{D}$ is defined is confusing in a first glance.

Being a bit picky, although I really liked Figure 3, a more descriptive caption would significantly improve the readability and understanding of the main idea of the paper: intra chunk tilling.

Finally, the related work section in the main paper is very shallow. In contrast, the related work section in the appendix is comprehensive and elucidative. I suggest the author to pull it to the main paper in order to give a broader view of the field of Linear Attention for non-familiar readers.

---

> ### Author Rebuttal · Authors · 2025-07-28
>
> We thank reviewer i6Dq for the helpful feedback, which helps to improve our paper.
> We appreciate that the reviewer finds our paper clearly identifies and addresses limitations in existing FLA implementations, that our paper is well-written with the technical details introduced in a clear manner.
>
> Suggestions:
>
> We thank the reviewer for the helpful suggestions regarding the related work and figure captions, citations and some minor mathematical details. We will rework the particular sections  and update our final version of the paper.
>
> Weaknesses:
>
> We would like to highlight that the main motivation of mLSTMsig is its reduced computation (no normalizer and max state) and hence faster runtimes at equal modeling performance, with the main differences visible in the faster runtimes of the kernels in Fig. 5.
>
> In Section 2.2 we explain that the transition from equations 1-5 to 6-7 happens due to rearranging the sequence dimension into N_c chunks of chunk size L. Therefore, we change the time index from t to chunk index k.
> As highlighted by the reviewer, this transition is not trivial and therefore we tried to make it more understandable by adding illustrative Figures 2 and Figure 9 (in the Appendix), which make the transition explicit for a small problem of chunk size L=4.
> We hope that especially Figure 9 will help to follow the formulas in the main paper.
> Furthermore due to space limitations the equations in Section 2.2 are a compact form of the detailed chunk-wise formulation in Appendix B.2.
>
> Questions:
>
> Investigating numerical stability implications is a common problem in many optimized deep learning frameworks and training setups, especially with mixed precision settings.
> We investigated the mean absolute errors for our mLSTMexp XL chunk kernels and did not find major sensitivity regarding the chunk size.
>
> We show the mean absolute errors for mLSTMexp XL chunk kernels forward compared to the pytorch reference implementation for context length 8192 at the same head dimension as in our kernel benchmark in Figure 5 (d_qk=128, d_hv=256) for Gaussian normal distributed random inputs in float32 and bfloat16:
> | Chunk size | Float32              | Bfloat16              |
> |------------|----------------------|-----------------------|
> | 64         | 7.8761462e-04        | 2.9082941e-03          |
> | 128        | 7.8597685e-04        | 2.9058390e-03          |
> | 256        | 7.8570450e-04        | 2.9046188e-03          |
> | 512        | 7.8560099e-04        | 2.9047025e-03          |
> | 1024       | 7.8498963e-04        | 2.9049833e-03          |
> | 2048       | 7.8578707e-04        | 2.9037838e-03          |
>
> **Chunksize selection and TFLA in practice**
>
> The runtime optimal chunk size depends on the underlying hardware accelerator and the model head dimensions (see Fig. 21, 25 and Eq. 113). This means that typically you have to tune the chunksize once for each head dimension configuration when switching to a specific accelerator.
> This process can be automated and in our source code we provide a kernel benchmarking module which could help with this.
>
> However, in some scenarios you might want to reduce peak memory usage, e.g. for long contexts and large models, where you trade-off runtime for GPU memory (see Fig. 6).
> In that case you want to maximize the batch size that fits in one GPU, which is typically limited by GPU memory size.
> In this case the chunk size L would be another hyperparameter (in addition to gradient checkpointing or gradient accumulation for example).
>
> **Extending Empirical Validation to other Architectures**
>
> In our paper we implemented TFLA for two different mLSTM variants with and without a normalizer state and different gate activation functions.
> We would like to highlight that RetNet (fixed forget gate, no input gate) and Simple GLA (no input gate) are a subset of mLSTMsig and are hence already implemented. This already demonstrates its generalizability. Since reimplementing and applying TFLA to other advanced linear RNNs requires a significant amount of resources, we leave this for future work, but provide guidance on how it might be applied in Appendix A.3.
>
>
> We thank the reviewer for their valuable feedback and hope that our clarifications help reinforce the merits of the submission.

---

> > ### Comment · Reviewer_i6Dq · 2025-08-03
> >
> > Thank you for the comprehensive response, especially for answering my questions in a direct way.
> >
> > I maintain my acceptance score.
> >
> > Great work!

---

### Official Review · Reviewer_Qinz · 2025-07-07

**Clarity:** 3
**Significance:** 3
**Originality:** 3
**Rating:** 4
**Confidence:** 5

**Summary:**

This paper proposes a two-level tiling algorithm for efficiently training linear attention models. In particular, it focuses on optimizing the mLSTM model with new sigmoid-based gating. Results show that the training algorithm can greatly accelerate training while mLSTM achieves comparable performance with other linear attention models.

**Questions:**

Can you explain the intuition behind achieving high arithmetic intensity in the main paper? What are the computational trade-offs involved?
Providing training speedup results at long context lengths (e.g., 65k tokens) would make the efficiency claims more convincing and practically relevant.

**Ethical Concerns:**

["NO or VERY MINOR ethics concerns only"]

**Paper Formatting Concerns:**

No.

**Quality:**

3

**Strengths And Weaknesses:**

Strengths:

A highly optimized mLSTM variant with strong empirical performance that demonstrates the effectiveness of the combined algorithmic and architectural improvements.
The ablation study on gate design is insightful and provides valuable understanding of the component contributions.


Weaknesses:
From a presentation standpoint, the paper would benefit from directly presenting mLSTM-sig + the new training algorithm as a unified contribution, which would be much clearer than the current flow. The current presentation suggests mLSTM and the algorithm are more tightly coupled than they actually are.

---

> ### Author Rebuttal · Authors · 2025-07-28
>
> We thank the reviewer Qinz for the helpful feedback. We appreciate that the reviewer highlights the strong empirical performance and finds our ablation study insightful.
>
> Weaknesses:
>
> We did not present mLSTMsig + TFLA as a unified contribution as TFLA could be applied to other Linear RNN variants as we outline in Section A.3.
> We agree that mLSTM and TFLA are not tightly coupled and in our updated version of the paper we will emphasize that mLSTM is one possible application of TFLA.
>
> Questions:
>
> Unfortunately, at this time we are not able to run new large scale experiments at extremely long contexts due to compute constraints. However, we extracted the training times for our largest scale experiments at 1.4B parameters for 8k context lengths (i.e. Table 5) from our experiment logs. We confirm that the speedups in kernel runtime from Figure 5 transfer to speedups in training step time. More specifically, we confirm that our TFLA kernels yield more than 30% end-to-end speed up over the Llama baseline.
> We will add this table to the final version of the paper.
>
> Training step times for the 1.4B models at 8192 context length with the same head dimensions as for our kernel benchmark in Figure 5. All runs use the same training setup with global batch size of 256 sharded across 32 GPUs, which corresponds to a local batch size of 8:
> | Model/Kernel                                 | Time (s) |
> |----------------------------------------------|----------|
> | Llama                                        | 2.39     |
> | mLSTMexp limit_chunk kernel (chunk size 64)  | 1.67     |
> | mLSTMexp xl_chunk kernel (chunk size 128)    | 1.66     |
> | mLSTMsig xl_chunk kernel (chunk size 128)    | 1.64     |
>
> **Arithmetic Intensity and computational trade-offs**
>
> TFLA trades-off memory vs. compute (See Figure 6).
> Due to the enhanced intra chunk tiling strategy (Figure 3), the chunk size can be chosen arbitrarily, overcoming the limitation by physical SRAM on the GPU.
>
> The chunk size L determines the number of chunks and hence number of materialized memory states $N_C = T / L$ (see Sec. 2.2) necessary to compute the forward and backward pass.
> The higher the chunk size, the fewer memory states (less memory) are materialized (i.e. stored to HBM). However, the number of FLOPs increases (see Fig. 19).
>
> The arithmetic intensity of an algorithm is defined as its FLOP per byte loaded or stored ratio. The arithmetic intensity for GPUs or TPUs is defined similarly by their FLOPS/s per bytes/s ratio (see Sec. G.1 Eq. (109)).
>
>
> Since a larger chunk size leads to more FLOPs and less bytes stored and loaded, the arithmetic intensity increases with larger chunk size. We show this relation in Fig. 22.
> At the point where the algorithm arithmetic intensity matches or exceeds the accelerators arithmetic intensity the workload can reach peak performance (see Fig. 23). We analyze this in Appendix G.
>
> We will add the intuition and explanations to the final version of the main paper.
>
> We thank the reviewer again for their helpful feedback, and hope they find their questions addressed.

---

### Decision · Program_Chairs · 2025-09-17

**Decision:**

Accept (poster)

**Comment:**

The paper proposes Tiled Flash Linear Attention (TFLA), a new GPU kernel for Linear RNNs and mLSTMs. Builds on Flash Linear Attention (FLA) but removes chunk-size limitations by adding two-level sequence parallelism: 1) Chunkwise parallelism (as in FLA), 2) Intra-chunk tiling (new contribution, inspired by FlashAttention2). Benchmarks show state-of-the-art training efficiency over FlashAttention 3, Linear Attention, and Mamba 2, especially for long-context training.
It is not a wholly new paradigm, but a good synthesis of existing GPU optimization techniques with a practical implementation. The paper demonstrates good efficiency gains for long-context training, accompanied by a strong theoretical and empirical analysis, as well as practical improvements to mLSTM.
The weaknesses of this paper include limited novelty, a narrow evaluation scope (mostly mLSTM), weaker inference improvements, and some clarity issues.
Overall, reviewers generally lean towards accept this paper.